# Increased new particle yields with largely decreased probability of survival to CCN size at the summit of Mt. Tai under reduced SO₂ emissions

Yujiao Zhu[1], Likun Xue[1,2*], Jian Gao[3], Jianmin Chen[4], Hongyong Li[1], Yong Zhao[5], Zhaoxin Guo[5], Tianshu Chen[1], Liang Wen[1], Penggang Zheng[1], Ye Shan[1], Xinfeng Wang[1], Tao Wang[6], Xiaohong Yao[7*], Wenxing Wang[1]

[1]Environment Research Institute, Shandong University, Qingdao 266237, China

[2]Collaborative innovation Center for climate Change, Jiangsu Province, Nanjing, 210023, China

[3]Chinese Research Academy of Environmental Sciences, Beijing 100012, China

[4]Shanghai Key Laboratory of Atmospheric Particle Pollution and Prevention (LAP3), Department of Environmental Science & Engineering, Fudan University, Shanghai 200433, China

[5]Taishan National Reference Climatological Station, Tai'an 271000, China

[6]Department of Civil and Environmental Engineering, Hong Kong Polytechnic University, Hong Kong, China

[7]Key Lab of Marine Environmental Science and Ecology, Ministry of Education, Ocean University of China, Qingdao 266100, China

*Correspondence to*: Likun Xue (xuelikun@sdu.edu.cn) and Xiaohong Yao (xhyao@ouc.edu.cn)

**Abstract.** Because anthropogenic sulfur dioxide (SO₂) emissions have decreased considerably in the last decade, PM₂.₅ pollution in China has been alleviated to some extent. However, the effects of reduced SO₂ on the particle number concentrations and subsequent contributions of grown new particles to cloud condensation nuclei (CCN) populations, particularly at high altitudes with low aerosol number loadings, are poorly understood. In contrast, the increase in provincial forest areas with the rapid afforestation in China over the last decades expectedly increases the biogenic emissions of volatile organic compounds and their oxidized products as nucleating precursors therein. In this study, we evaluated the campaign-based measurements made at the summit of Mt. Tai (1534 m a.s.l.) from 2007 to 2018. With the decrease in the SO₂ mixing ratios from $15 \pm 13$ ppb in 2007 to $1.6 \pm 1.6$ ppb in 2018, the apparent formation rate of new particles (FR) and the net maximum increase in the nucleation-mode particle number concentration (NMINP) in the spring campaign of 2018 was 2–3 fold higher than those in the spring campaign of 2007, with almost the same occurrence frequency of new particle formation (NPF) events. In contrast, the campaign-based comparison showed that the occurrence frequency, in which the maximum geometric median diameter of the grown new particles ($D_{pgmax}$) was >50 nm, decreased considerably, from 43%–78% of the NPF events before 2015 to <12% in 2017–2018. Assuming >50 nm as a CCN threshold size at high supersaturations, the observed net CCN production decreased from $3.7 \times 10^3$ cm⁻³ (on average) in the five campaigns before 2015 to $1.0 \times 10^3$ cm⁻³ (on average) in the two campaigns in 2017–2018. We argue that the increases in the apparent FR and NMINP are mainly determined by the availability of organic precursors that participate in nucleation and initial growth, whereas the decrease in

the growth probability is caused by the reduced emissions of anthropogenic precursors. However, large uncertainties still exist because of a lack of data on the chemical composition of these smaller particles.

## 1. Introduction

Atmospheric new particle formation (NPF) is regarded as an important source of aerosol particles in terms of number concentrations, and the newly formed particles can grow into a variety of sizes, with different health and climate effects. For example, particles larger than 50–80 nm may act as cloud condensation nuclei (CCN), whereas those larger than 100 nm may directly affect solar radiation (Kulmala and Kerminen, 2008; Kerminen et al., 2012; Seinfeld and Pandis, 2012). Sulfuric acid ($H_2SO_4$) is considered as the key nucleating precursor for NPF, and other species, such as ammonia ($NH_3$), amines, and highly oxygenated molecules [HOMs—oxidation products of volatile organic compounds (VOCs)], can also participate and enhance nucleation in the continental troposphere (Ehn et al., 2014; Tröstl et al., 2016; Yao et al., 2018; Kerminen et al, 2018; Chu et al., 2019; Lee et al., 2019). The subsequent growth of new particles is affected by not only the abovementioned precursors but also the semivolatile compounds (Riipinen et al., 2012; Ehn et al., 2014; Tröstl et al., 2016).

NPF events have been reported widely throughout the world, including in severely polluted urban and rural areas in China that experience high sulfur dioxide ($SO_2$) concentrations and high aerosol loading (Kulmala et al., 2004; Gao et al., 2009; Guo et al., 2012; Nie et al., 2014; Kerminen et al., 2018; Chu et al., 2019). In the last few decades, anthropogenic emissions of gaseous and particulate air pollutants in China have been reduced substantially due to rigorous emission control policies. Between 2007 and 2018 (the observation period in this study), the national total $SO_2$ emissions decreased by 67% (from 24.7 million tons to 8.2 million tons), and the national average ambient $SO_2$ concentrations decreased by 73% (from 17.9 ppb to 4.9 ppb; see Fig. S1). The North China Plain (NCP) region experiences the most severe $SO_2$ pollution, which has visibly decreased trend since 2011 (Krotkov et al., 2016; Fan et al., 2020). Such huge reductions in $SO_2$ emissions may alter the frequency and intensity of NPF events and the subsequent growth of new particles. The changes in the mixing ratios of VOC components, ambient oxidants, aerosol loading, and meteorological factors may also influence NPF events, yielding more complex and uncertain feedback (Kulmala and Kerminen, 2008; Zhang et al., 2012).

The long-term changes in NPF events under lower $SO_2$ conditions have been studied in several cities in Europe and the U.S. For example, decreased NPF frequency and reduced new particle yields were associated with decreases in $SO_2$ concentrations in Pittsburgh (U.S.), Rochester (U.S.), and Melpitz (Germany) (Hamed et al., 2010; Wang et al., 2011, 2017; Saha et al., 2018). In contrast, long-term studies in Pallas (Finland), Hyytiälä (Finland), and Crete (Greece) observed no trends in NPF frequencies despite considerable decreases in the ambient $SO_2$ concentrations all over Europe (Asmi et al., 2011; Nieminen et al., 2014; Kalivitis, et al., 2019). Moreover, a slight upward trend in the particle formation and growth rates was observed in Pallas and Hyytiälä, attributable to the increased biogenic VOC (BVOC) emissions (Asmi et al., 2011; Nieminen et al., 2014).

In China, the earliest observation of NPF events started in approximately around 2004 in Beijing (Wu et al., 2007). Comparison of tens of independent experiments showed that the NPF frequency has remained relatively constant until recent years, possibly due to the reduced production and reduced loss rate of $H_2SO_4$; these phenomena may have canceled each other out to some extent in Beijing (Chu et al., 2019; Li et al., 2020). In addition, a recent study reported that China has experienced rapid afforestation by a net increase of ~18% in leaf area from 2000 to 2017, with provincial forest areas increasing by between 0.04 million and 0.44 million hectares per year over the past 10 to 15 years, leading to an increase in the $CO_2$ sink based on long-term observations over large spatial scales (Chen et al., 2019; Wang et a., 2020). BVOC emissions in China are reasonably expected to increase over the past decades (Zhang, et al., 2016; Chen et al., 2017). Considering the key roles of oxidized BVOCs in NPF events (Riipinen et al., 2012; Ehn et al., 2014; Tröstl et al., 2016; Chu et a., 2019), the responses of NPF in China, in terms of their occurrence frequency, intensity and potential impacts on the climate, to reduced anthropogenic $SO_2$ emissions may not be the same as those observed in Europe and North America. However, the long-term changes in NPF intensities and the subsequent growth of new particles have not been studied in China, where the anthropogenic emissions of various air pollutants and biogenic emissions of VOCs have been changing in opposite directions in the past decade.

In this study, we analyzed the measurement data of particle number concentrations, chemical compositions, trace gases, and meteorological parameters collected at the summit of Mt. Tai (36.25°N, 117.1°E, 1534 m a.s.l.) during seven observational campaigns from 2007 to 2018. Mt. Tai is the highest mountain in the NCP, located at the region's center, and the observation station has been widely deployed to investigate regional air pollution as well as transport and chemical processes in the NCP (Gao et al., 2005; Li et al., 2011; Sun et al., 2016; Wen et al., 2018). Moreover, the summit is close to the top of the planetary boundary layer or even in the free troposphere sometimes, and is characterized by relatively few pre-existing particles, strong UV solar radiation, and low ambient temperature, which favor NPF events (Li et al., 2011; Shen et al., 2016a; Lv et al., 2018;). The tree coverage areas around the sampling site evidently increased from 2003 to 2016 based on MODIS satellite data (Fig. S2). The contribution of new particles, compared with that of primary particles, to the CCN population reportedly increases above the boundary layer, indicating a critical role of high-altitude NPF in cloud formation and the related climate impacts (Merikanto et al., 2009). The main purpose of this study included: 1) to examine the effects of reduced $SO_2$ emissions on regional NPF events at a high altitude (from the upper boundary layer to the lower free troposphere), i.e., the changes in the NPF frequency, intensity and the subsequent growth of new particles; 2) to quantify the potential contribution of new particles to the CCN population and its changes under decreasing $SO_2$ emission; and 3) to rationalize the variation patterns of the NPF characteristics and CCN parameters in terms of observational concentrations of gaseous precursors and their origins and atmospheric behaviors. Note that all these changes in the study area should occur under the background of an increase in BVOCs and their oxidized products as nucleating precursors over the decades in China, although no studies can confirm the decadal increase in nucleating precursors from biogenic VOCs because of the lack of related analytic technologies in the past.

## 2. Methods

### 2.1 Experimental

This study comprised seven intensive campaigns from 2007 to 2018, and the details are summarized in Table 1. The duration of each campaign varied from 18 days to 71 days. The measurement data obtained in the four campaigns in 2007, 2014, and 2015 have been reported by Gao et al. (2008) and Lv et al. (2018), and here, all of the available data were combined to examine the effects of reduced $SO_2$ emissions on regional NPF events.

All measurements were obtained using commercial instruments, which were housed in a container and have been described in previous studies (e.g., Zhou et al., 2009; Zhu et al., 2017; Lv et al., 2018). During the seven campaigns, the particle number size distributions (PNSDs) were monitored using a wide-range particle spectrometer (*WPS; Model 1000XP, MSP Corporation, USA*) at ambient relative humidity (RH). Conductive tubes (TSI 1/4 in.) were used for the WPS sampling, and the length of the tube was kept at approximately 2 m in each campaign. The WPS combines a differential mobility analyzer (DMA), a condensation particle counter (CPC), and a laser particle spectrometer (LPS). The DMA and CPC can measure particles in the 10–500 nm (or 5–500 nm in the advance mode) size range and were set up to 48 channels. The SWS mode (DMA operating in the voltage-scanning mode) was selected. The LPS covers the size range of 350 nm–10 μm and is divided into 24 channels. In this study, the detection limit of the DMA was 10 nm in 2007 and 2009, while it was 5 nm in 2014, 2015, 2017, and 2018. For consistency, the particles sized 10–300 nm were used for the calculations in all campaigns except for that in spring of 2007, when the data of >153 nm particles were missing, so we used the data of 10–153 nm particles instead for calculation. The net increases in the number concentrations of 10-25 nm particles during the initial several hours of NPF events were over 1~2 orders of magnitude in this study, e.g., in Fig. 1b, d. The uncertainties of the measured particle number concentrations at approximately 10 nm in 2007 and 2009 and those after had a negligible influence on the net increases. In addition, the use of the 10-153 nm particles in 2007 may lead to underestimation of the particle number concentration, as detailed in the supplementary materials

The WPS instrument was calibrated and/or repaired every 1-2 years by its vendor. The regular maintenance allowed the WPS to perform well, based on the recent comparison results of the WPS and a new scanning mobility particle sizer (SMPS, Grimm) in the summer of 2020, as shown in Fig. S3. The regular calibration parameters included the DMA sample/sheath flow, LPS sample/sheath flow, DMA/CPC pressure, DMA voltage, and DMA/ambient temperature. Polystyrene latex (PSL) spheres (NIST) with mean diameters of 100.7 nm and 269 nm were used for calibration. At the beginning of each campaign, the zero-points of the DMA, CPC, and LPS were checked using a purge filter at the inlet. Sometimes, the WPS operated improperly, and the data were excluded from the analysis (see the Fig. S4 for the occasional unexpected errors in three channels around 213 nm). In addition, we reproduced the $PM_{2.3}$ mass concentration from the WPS data and found that it was reasonably comparable to the measured $PM_{2.5}$, further supporting the accuracy of the WPS data. Details can be found in Fig. S5 and the supplementary materials.

The trace gases were monitored during each campaign. $SO_2$ was measured using an ultraviolet fluorescence analyzer (*Model 43C, Thermo Electron Corporation, USA*); $O_3$ was monitored using two ultraviolet absorption analyzers (*Model 49C, Thermo Electron Corporation, USA*, or *Model 400U, Advanced Pollution Instrumentation Inc., USA*); NO and $NO_2$ were

monitored using a chemiluminescence analyzer (*Model 42C or 42i, Thermo Electron Corporation, USA*) equipped with a blue light converter before August 2014 and subsequently using a chemiluminescence analyzer (*Model T200U, API, USA*) and a cavity-attenuated phase-shift spectroscopy instrument (*Model T500U, API, USA*), respectively. For the analysers of $SO_2$, $O_3$, NO and $NO_2$, we performed multipoint calibrations every month and changed the filter every two weeks. The detection limits of $SO_2$, $O_3$, NO and $NO_2$ were 1 ppb, 0.4 ppb, 0.04 ppb, and 0.4 ppb, respectively. $PM_{2.5}$ was measured using a TEOM 1400a

in 2007 and a Thermo 5030 SHARP after 2014. This device was calibrated by mass foil calibration according to the instrument manual, and the detection limit was 0.5 $\mu g/m^3$. The inorganic water-soluble ions in $PM_{2.5}$ together with the acid and alkaline gases were measured using an online Ambient Ion Monitor (*URG-AIM 9000B, URG Corporation, USA*; only for water-soluble ions in $PM_{2.5}$) in 2007 and using a Monitor for Aerosols and Gases (*MARGA; ADI20801, Applikon-ECN, Netherlands*) in the five campaigns from 2014 onward. A multipoint calibration was performed for the online MARGA before and after the field

campaigns to examine the sensitivity of the detectors. The detection limits were determined to be 0.05, 0.05, 0.04 and 0.05 $\mu g$ $m^{-3}$ for $Cl^-$, $NO_3^-$, $SO_4^{2-}$ and $NH_4^+$, respectively. More details about the instrument calibration can be found in Wen et al. (2018) and Li et al. (2020). Data on the meteorological parameters including temperature (T), RH, wind speed, wind direction, and precipitation were provided by the Mt. Tai Meteorological Station.

The air mass back trajectories were calculated using the Hybrid Single Particle Lagrangian Integrated Trajectory

(HYSPLIT) model. The input meteorological data [Global Data Analysis System (GDAS) data] were used with a 1° latitude–longitude resolution. A trajectory ending height of 1400 m a.g.l. was selected because the terrain height on Mt. Tai was approximately 150 m in the GDAS data.

**2.2 Calculation methods**

**2.2.1 Definition of NPF events and relevant parameters**

In this study, particles with diameters smaller than 25 nm were defined as nucleation mode particles (Kulmala et al., 2012). Following the criteria proposed by Dal Maso et al. (2005) and Kulmala et al. (2012), three features had to be met for an event to qualify as NPF: 1) a distinctly new nucleation mode particles must appear in the size distribution; 2) the new mode should prevail over a time span of hours; and 3) the new mode should show signs of growth. All three features are required for a day (00:00-23:59 LT) to be classified as an NPF day. Otherwise, the day is classified as a non-NPF day.

The initial time of an NPF event was defined as when new nucleation mode particles started to be observed (e.g., $t_0$ in Fig. 1b, d). The end time of an NPF event was defined as the new particle signal dropping to a negligible level and the total particle number concentrations approaching the background levels before the NPF event. In cases with the invasion of other plumes, the end time was determined to be when the new particle signals were suddenly overwhelmed by plumes and could

no longer be identified (e.g., the end times in Fig. 1b, d). The NPF event duration was defined as the time duration between the initial time and end time of an NPF event. Note that the detection limit of WPS was 10 nm, but the particles were nucleated at critical cluster sizes of approximately 1-1.5 nm. Therefore, the NPF actually occurred at some time prior to our observation, and the actual duration should be longer than our calculation.

Three parameters are commonly used to evaluate NPF characteristics, viz., apparent formation rate (FR), growth rate (GR), and condensation sink (CS) (Sihto et al., 2006; Kulmala et al., 2012). The apparent FR of new particles is calculated based on nucleation mode particles with sizes of 10-25 nm. The GR is quantified by fitting the geometric median diameter of new particles ($D_{pg}$) during the whole particle growth period. The size range of $D_{pg}$ varies from event to event. Details of the calculation equations can be found in the supplementary materials. Note that the lack of measurements of >153 nm particles in the spring campaign of 2007 may lead to an underestimation of the CS. We tested the possible underestimation using the data measured in 2018 by comparing the CS values calculated for the measured number concentrations of particles in the 10-153 nm, 10-300 nm and 10-2500 nm size ranges. The use of the 10-153 nm particles may lead to ~50% underestimation of the CS compared to that using the 10-2500 nm particles. Thus, the CS value from the 10-153 nm particles in the spring campaign of 2018 was compared with that obtained in the spring campaign of 2007. However, the CS values from the 10-300 nm particles accounted for 94% of those from the 10-2500 nm particles. Thus, the CS values from the 10-300 nm particles were used throughout the study, except in 2007.

Another two metrics were applied to characterize the NPF events, i.e., the net maximum increase in the nucleation-mode particle number concentration (NMINP), and the maximum size of $D_{pg}$ ($D_{pgmax}$). The two metrics were proposed in our previous studies (Zhu et al., 2017, 2019). The NMINP indicated the intensity of an NPF event, which was calculated as:

$$\text{NMINP} = N_{10-25\,\text{nm}}(t_1) - N_{10-25\,\text{nm}}(t_0) \tag{1}$$

where $N_{10-25\,\text{nm}}$ is the sum of the nucleation mode particle number concentrations, and $t_0$ and $t_1$ represent the time when an NPF event is initially observed and the time when $N_{10-25\,\text{nm}}$ reaches the maximum value, respectively. Fig. 1b, d shows the schematic diagram of the NMINP.

Note that a few spikes were occasionally observed with a broader particle number size distribution during the NPF period. These spikes were excluded in the calculation of the FR, GR, $D_{pg}$, NMINP and CCN parameters (described in 2.2.2) because they may reflect primary particles from localized sources (Liu et al., 2014; Man et al., 2015; Zhu et al., 2017).

According to the different sizes of $D_{pgmax}$, the NPF events were classified into three categories (as shown in Fig. 1). In Category 1 events (e.g., December 25, 2017, Fig. 1a), the new particles grow to a $D_{pgmax}$ of <50 nm and are too small to serve as CCN. In Category 2 events (e.g., April 7, 2018, Fig. 1b), the new particles grow to a $D_{pgmax}$ of 50–80 nm. In Category 3 events (e.g., December 24, 2017, Fig. 1a), the new particles grow to $D_{pgmax}$ of >80 nm. The NPF events in Categories 2 and 3 can be regarded as climate-relevant events.

## 2.2.2 CCN parameters

In the absence of direct CCN measurements, the potential contribution of new particles to the CCN population can be estimated from the particle number size distribution (Lihavainen et al., 2003; Rose et al., 2017). Theoretically, particles larger than 50 nm (i.e., 80 nm) can be activated as CCN under quite high (moderate) supersaturation (Dusek et al., 2006; Petters and Kreidenweis, 2007; Ma et al., 2016), and particles larger than 100 nm can directly impact the climate by scattering and

195 absorbing solar radiation (Charlson et al., 1992; Seinfeld and Pandis, 2012). In this study, we introduced three terms: the net increase in the NPF-derived CCN number concentration ($\Delta N_{CCN}$), the apparent survival probability of new particles to the CCN sizes (SP), and the relative increase ratio of the CCN population ($R_{CCN}$). Three sizes, viz., 50 nm, 80 nm, and 100 nm, were defined as the CCN threshold sizes. $\Delta N_{CCN}$ was calculated following the method of Rose et al. (2017):

$$\Delta N_{CCN} = N_{CCN}(t'_1) - N_{CCN}(t_0) \tag{2}$$

where the $N_{CCN}$ terms represent the potential CCN number concentrations and were estimated from the number concentrations of particles larger than 50 nm, 80 nm, and 100 nm ($N_{CCN50}$, $N_{CCN80}$, and $N_{CCN100}$, respectively); $t_0$ is the time when an NPF event is initially observed, the same as that in equation (1); and $t'_1$ is the time when $N_{CCN}$ reaches the maximum value during the new particle growth periods. Each concentration was taken as a 1-h average. The $\Delta N_{CCN}$ term eliminates the influence of pre-existing particles. A schematic diagram of $N_{CCN100}$ and $N_{CCN50}$ can be found in Fig. 1b, d.

The SP was calculated as described by Zhu et al. (2019):

$$SP = \Delta N_{CCN}/NMINP \tag{3}$$

Note that the spatial-temporal heterogeneity during NPF events may result in high SPs. If the observed $\Delta N_{CCN}$ exceeded NMINP and the calculated SPs exceeded 100%, suggesting equation (3) was not applicable in these cases, and SP was therefore not calculated.

The $R_{CCN}$ values were the ratios of the NPF-derived CCN to the pre-existing CCN and were calculated as follows:

$$R_{CCN} = \Delta N_{CCN}/N_{CCN}(t_0) \tag{4}$$

Moreover, Fig. 1b shows that the choice of $t_0$ may lead to underestimation of $\Delta N_{CCN}$ to some extent in the presence of spatial-temporal heterogeneity of pre-existing particles with diameters larger than 50 nm. In these cases, the mean value of $N_{CCN}$ in the percentiles smaller than 5th during the whole NPF event may be more accurate representing the background (see

the grey dashed line in Fig. 1 b). However, this method may also introduce more subjective factors and therefore was not adopted in this study.

In addition, the maximum geometric median diameter of the grown new particles never exceeded 89 nm in spring 2007. Considering the log-normal distribution of the grown new particles, the number concentration of grown new particles with

diameters >153 nm was less than 15% (Fig. S6). Thus, it is safe to say that the lack of data for >153 nm particles had a negligible effect on the calculated $\Delta N_{CCN50}$, $\Delta N_{CCN80}$ and $\Delta N_{CCN100}$ in 2007.

### 2.2.3 Sulfuric acid proxy

The proxy for the $H_2SO_4$ concentration could be roughly estimated based on the solar radiation (SR), $SO_2$ concentration, CS, and RH as follows (Mikkonen et al., 2011; Lv et al., 2018):

$$[H_2SO_4] = 8.21 \times 10^{-3} \cdot k \cdot SR \cdot [SO_2]^{0.62} \cdot (CS \cdot RH)^{-0.13} \tag{5}$$

where $k$ is a temperature-dependent reaction rate constant, and SR was estimated from the HYSPLIT model.

The contribution of $H_2SO_4$ vapor to the particle growth from $D_{p0}$ to $D_{p1}$ can be expressed by the following equation (Kulmala et al., 2001):

$$R = ([H_2SO_4]_{avg}/C) \times 100\% \tag{6}$$

where $[H_2SO_4]_{avg}$ is the average concentration of $H_2SO_4$ during the particle growth period, and C is the total concentration of condensable vapor for the particle growth from $D_{p0}$ to $D_{p1}$, which can be calculated as described by Kulmala et al. (2001). Notably, uncertainty may exist in the estimated contribution of the $SO_2$ concentrations and radiation intensity to $H_2SO_4$, as well as in the contribution of $H_2SO_4$ to the particle growth.

## 3. Results

### 3.1 Variation in the NPF frequency

During the seven campaigns, NPF events were observed on 106 of the 265 sampling days. As shown in Fig. 2, the NPF frequencies in the three seasons of different years were surprisingly almost the same, i.e., 50% in the spring of 2007, 50% in the summer of 2009, 49% in the winter of 2017, and 51% in the spring of 2018. However, the NPF frequency decreased to 42% in the summer of 2014, 33% in the fall of 2014, and 20% in the summer of 2015. The low NPF frequencies were likely caused by perturbations from meteorological conditions. For example, there were 15 rainy days out of the 40 sampling days during the 2015 summer campaign, but only 3 rainy days out of the total 18 sampling days in the 2009 summer campaign. Moreover, the solar radiation averaged over the 2009 summer campaign was 1.4 times that of the 2015 summer campaign (Fig. S7). These factors may have caused the NPF frequency in the 2009 summer campaign to be close to that in the other season campaigns but that of the 2015 summer campaign to be lower.

When Categories 1, 2, and 3 of the NPF events were examined separately, the Category 1 NPF frequencies in the winter of 2017 (43%) and the spring of 2018 (49%) were significantly higher than those before (5%–21%; $p < 0.05$). Category 2 events were absent in the winter of 2017, whereas Category 3 events were absent in the spring of 2018. The sums of the Category 2 and 3 NPF frequencies in 2017 (6%) and 2018 (3%) were significantly lower than those before (14%–39%; $p < 0.05$), even in comparison with the relatively low NPF frequencies in the summer of 2015 (15%) and the fall of 2014 (14%). When the sums of the Category 2 and 3 NPF frequencies in each campaign were normalized by the corresponding total NPF frequency, the boundary was clearer, i.e., the normalized sum values were as high as 43%–78% before 2015 and <12% in 2017–2018. Clearly, the newly formed particles observed at Mt. Tai in 2017–2018 were less climatically relevant than those before 2015 (64-78% in the three summer campaigns versus 43% and 59% in the fall and spring campaigns), despite the comparable NPF frequencies.

## 3.2 Variations in the apparent FR, NMINP, GR, and D$_{pgmax}$

We used four metrics, i.e., the apparent FR, NMINP, GR, and D$_{pgmax}$, to characterize the NPF events and evaluate the potential climate impacts of the grown new particles (Fig. 3). During the four campaigns in 2007, 2009, and 2014, the calculated apparent FR varied narrowly in each campaign and the campaign average narrowed to 0.8–1.2 cm$^{-3}$ s$^{-1}$. The apparent FR increased in the three subsequent campaigns, i.e., 2.6 ± 1.3 cm$^{-3}$ s$^{-1}$ in 2015, 2.0 ± 1.7 cm$^{-3}$ s$^{-1}$ in 2017, and 3.0 ± 2.7 cm$^{-3}$ s$^{-1}$ in 2018. The apparent FRs were 3–7 times lower than those obtained from the measurements with a lower limit of 3 nm of the twin differential mobility particle sizer (TDMPS) and neutral cluster and air ion spectrometer (NAIS) at the same site during previous campaigns (Shen et al., 2016a,b; Lv et al., 2018).

The NMINP showed a temporal variation pattern similar to that of the apparent FR (Fig. 3b). The campaign average NMINP varied in a narrow range of 3.8–5.1 × 10$^3$ cm$^{-3}$ in 2007–2014, but then increased to 6.3 ± 2.6 × 10$^3$ cm$^{-3}$ in 2015, 9.4 ± 7.9 × 10$^3$ cm$^{-3}$ in 2017, and 1.1 ± 1.0 × 10$^4$ cm$^{-3}$ in 2018. The increase in the NMINP should enhance the contribution of NPF events to the ambient particle number concentration, but the NMINP at Mt. Tai before 2015 was only approximately ¼–½ of those of our previous observations in urban and marine atmospheres (Zhu et al., 2017, 2019).

The variations in GR were strongly seasonally dependent (Fig. 3c). Higher GRs were generally observed in the summer campaigns, with the three campaign averages in the range of 7.3–9.6 nm h$^{-1}$. The higher GRs in summer were due to the higher photochemical reactions and biological activities, which is consistent with those reported in the literature (Kulmala et al., 2004; Chu et al., 2019). The reverse was true in winter, and a lower GR was expectedly observed in the winter of 2017, i.e., 2.3 ± 1.3 nm h$^{-1}$. The GRs in the fall and spring campaigns ranked between those of the summer and winter campaigns. For example, the average GR was 4.9 ± 2.7 nm h$^{-1}$ in the fall of 2014. The average GR in the spring of 2007 (4.4 ± 2.3 nm h$^{-1}$) was slightly higher than that in the spring of 2018 (3.3 ± 2.3 nm h$^{-1}$), although the apparent FR and NMINP increased considerably in the latter spring campaign.

The $D_{pgmax}$ is partially determined by the GR. The largest campaign average $D_{pgmax}$ of $84 \pm 39$ nm appeared as expected in the summer of 2009, followed by $71 \pm 24$ nm in the summer of 2015. However, the campaign average $D_{pgmax}$ was only $60 \pm 18$ nm in the summer of 2014, followed by $57 \pm 16$ nm in the spring of 2007, $53 \pm 22$ nm in the fall of 2014, and $40 \pm 40$ nm in the winter of 2017. The campaign average $D_{pgmax}$ in the spring of 2018 was the lowest, i.e., $29 \pm 13$ nm, although the campaign average GR was even larger than that in the winter of 2017. These findings indicate that $D_{pgmax}$ is clearly determined not only by the GR but also by unidentified factors, which is addressed in Section 4.2.

In summary, we found that the apparent FR and NMINP in the spring campaign of 2018 were higher than those of 2007. The GR showed strong seasonal dependence. The $D_{pgmax}$ was significantly lower in 2018, but the GR alone could not explain the lower values.

## 3.3 Potential contribution to CCN production from the NPF events

Direct measurements of the CCN were not available; therefore, the potential contributions of the grown new particles to the CCN population were estimated using equations (2)–(4). The contributions varied considerably between campaigns (Fig. 4). In general, the NPF-derived CCNs were seasonally dependent, i.e., the highest number concentrations occurred in summer, followed by spring, fall, and winter. With an increase in the threshold diameters, roughly corresponding to a decrease in supersaturation from >0.6% to <0.1% (Li et al., 2015), the estimated contributions decreased because new particles were continuously removed either by coagulation or atmospheric deposition during the particle growth. During the three summer campaigns in 2009, 2014, and 2015, larger NPF-derived CCNs were estimated with average $\Delta N_{CCN50}$, $\Delta N_{CCN80}$, and $\Delta N_{CCN100}$ values of $4.4 \pm 2.5 \times 10^3$ cm$^{-3}$, $1.9 \pm 1.5 \times 10^3$ cm$^{-3}$, and $1.0 \pm 0.9 \times 10^3$ cm$^{-3}$, respectively. Overall, the values decreased by approximately 50% in the spring of 2007 and the fall of 2014. The NPF-derived CCNs in these five campaigns were larger than those reported in previous studies for the same season at Mt. Chacaltaya (5240 m, Bolivia) and Botsalano (1424 m, South-Africa) (Kerminen et al., 2012; Rose et al., 2017). In comparison, extremely low NPF-derived CCNs were observed in 2017 and 2018, i.e., $\Delta N_{CCN50}$ of only $1.1 \pm 1.7 \times 10^3$ cm$^{-3}$, $\Delta N_{CCN80}$ of $0.5 \pm 0.9 \times 10^3$ cm$^{-3}$, and $\Delta N_{CCN100}$ of $0.2 \pm 0.5 \times 10^3$ cm$^{-3}$.

High SPs were found during the three summer campaigns in 2009, 2014, and 2015, with average $SP_{CCN50}$, $SP_{CCN80}$, and $SP_{CCN100}$ values of 61%, 23%, and 14%, respectively (Fig. 4b). The SPs decreased by approximately 30% in the spring of 2007 and the fall of 2014. In 2017 and 2018, the average $SP_{CCN50}$, $SP_{CCN80}$, and $SP_{CCN100}$ were only 10%, 4%, and 1%, respectively, indicating that only a minor fraction of new particles could grow to CCN sizes before being scavenged.

Figure 4c shows the percentage increase in the NPF-derived CCN relative to the pre-existing CCN. The percentages were the highest in the summer of 2014, e.g., $6.8 \times 10^2$%, $6.0 \times 10^2$%, and $4.8 \times 10^2$% for $R_{CCN50}$, $R_{CCN80}$, and $R_{CCN100}$, respectively. This finding could be attributed to the combination of high $\Delta N_{CCN}$ and low number concentrations of pre-existing particles in that campaign (Fig. S8-9). In the remaining four campaigns during 2007–2015, the percentages still exceeded 100%, i.e., $2.5 \times 10^2$%–$3.8 \times 10^2$% for $R_{CCN50}$, $1.5 \times 10^2$%–$2.9 \times 10^2$% for $R_{CCN80}$, and $1.1 \times 10^2$%–$2.8 \times 10^2$% for $R_{CCN100}$. These ratios are within the range reported in the literature (50%–$1.1 \times 10^3$%), although the calculation methods of the studies were slightly different

(Lee et al., 2019). However, in 2017 and 2018, the percentages decreased considerably, e.g., <40% for $R_{CCN50}$ and <20% for $R_{CCN80}$ and $R_{CCN100}$.

## 4. Discussion

### 4.1 Question 1: What caused the unexpected responses of NPF to decreasing $SO_2$ concentrations?

$H_2SO_4$ oxidized from ambient $SO_2$ is one of the most important precursors for atmospheric nucleation. Decreases in ambient $SO_2$ mixing ratios, e.g., an annual average concentration decreases from 9 ppb to 1 ppb in Pittsburgh, 5 ppb to 3 ppb in Rochester, and 5 ppb to 2 ppb in Melpitz, have been reported to cause 40%–60% reductions in the NPF occurrence frequency and 40%–70% reductions in the NPF intensity (e.g., Hamed et al., 2010; Wang et al., 2011, 2017; Saha et al., 2018). However, this was not the case at the summit of Mt. Tai, where the NPF occurrence frequencies were almost invariant in the spring of 2007, summer of 2009, winter of 2017, and spring of 2018. The observed $SO_2$ mixing ratios in this study decreased considerably, from $15 \pm 13$ ppb in 2007 to $1.6 \pm 1.6$ ppb in 2018 (the $SO_2$ during the NPF periods decreased from $17 \pm 11$ ppb to $2.8 \pm 1.8$ ppb, Fig. 5a). In addition, the $SO_2$ emissions in China were reduced by approximately two-thirds from 2007 to 2018 (Fig. S1), where the sharpest reduction occurred in 2015–2016 owing to stringent mitigation policies.

As the calculated CSs before the NPF events in the 2017 and 2018 campaigns were higher than those in the 2007 and 2009 campaigns (Fig. 5b), CSs were unlikely to be the cause for the lack of decreases in the NPF occurrence frequency in 2017 and 2018. It has been reported that a low CS is not necessary to promote NPF occurrence at altitudes higher than 1000 m (Sellegri et al., 2019). Thus, other factors such as meteorological conditions and biogenic precursors (e.g., amines and highly oxidized organics) may overwhelm the effects of $SO_2$ and CS on the NPF occurrence frequency at Mt. Tai. Note that the increased CSs in the last decade is also found on the basis of our independent unpublished measurements made at a coastal megacity of northern China, which is out of the scope of this study.

We further conducted a few statistical tests to explore the association of the apparent FR and NMINP with $SO_2$. The correlation analysis using the campaign averages showed weak negative correlations for the apparent FRs and NMINPs with the $SO_2$ mixing ratios (r = 0.4 and 0.3, respectively; both $p > 0.05$). Again, these results implied that other factors overwhelmed the effect of the $SO_2$ mixing ratios on the apparent FRs and NMINPs. When the observations were analyzed case by case, the correlations of the apparent FRs and NMINPs with the $SO_2$ mixing ratios were even weaker, with r = −0.12 and −0.14, respectively (both $p > 0.05$). Similar results were found when the estimated $H_2SO_4$ vapor was used for correlation analysis (r = −0.12 and −0.13, respectively; both $p > 0.05$). The scatter plots are shown in Fig. S10.

Recall that the occurrence frequencies of NPF were also almost the same in the spring of 2007 and 2018, at high values of 50-51%, implying that ambient factors in both campaigns favored NPF. Table 2 provides a comprehensive comparison of the measured air pollutants of the two spring campaigns. The decrease in the $SO_2$, estimated $H_2SO_4$ concentration, and $NH_3$ did not explain the increases in the FRs and NMINP in 2018. Although amines were not measured, they are usually highly

correlated with $NH_3$ (Xie et al., 2018). Based on the unique roles of HOMs in enhancing atmospheric nucleation and promoting the growth of new particles (Paasonen et al., 2010; Ehn et al., 2014; Kerminen et al., 2018), it was speculated that HOMs played an important role in the unexpected responses of NPF to lower $SO_2$ in 2018. Increased HOMs were expected on the NPF event days on the basis of the rapid afforestation over the last decade in China (Chen et al., 2019; Wang et a., 2020) and the increase in the forest areas upwind the sampling site from the west to north. NPF events frequently took place when air mass came from the direction (Fig. S2). However, we had no measurements of HOMs. Nevertheless, the correlation between the FR and NMINP at Mt. Tai appears to support the hypothesis as presented below.

During the 106 cases of NPF events, the apparent FR and NMINP showed a good linear correlation (r = 0.84, $p < 0.01$) (Fig. 6). The fitted equation was highly consistent with those derived for urban and marine atmospheres (Man et al., 2015; Zhu et al., 2017, 2019; Ma et al., 2020). The strong linear relationship between the apparent FR and NMINP suggested that $H_2SO_4$ vapor was sufficient for nucleation, and the NPF intensity was very likely determined by the abundance of organic vapors available for participating in nucleation. Following the equation in the literature, i.e., FR = $k_{NucOrg}[H_2SO_4]^m$ [NucOrg]$^n$ (where $k_{NucOrg}$ is a constant, and m and n are integers; Zhang et al., 2012), the apparent FR is controlled by the concentrations of both $H_2SO_4$ vapor and organic vapor. We then considered two technical terms, i.e., the total concentration of $H_2SO_4$ vapor and the consumed amount of $H_2SO_4$ vapor for NPF. Unlike the apparent FR, the NMINP was always determined by the consumed amount of $H_2SO_4$ vapor, which may or may not have a positive correlation with the total concentration of $H_2SO_4$ vapor. The linear correlation between the FR and NMIMP suggests one possibility, i.e., the $H_2SO_4$ vapor was sufficient and the availability of organic vapor determined both the FR and the consumed amount of $H_2SO_4$ vapor proportional to the NMINP. Previous field measurements have shown that gaseous $H_2SO_4$ at concentrations of $10^5$ molecules/cm$^3$ is necessary for NPF (McMurry et al., 2005; Nieminen et al., 2009; Erupe et al., 2010; Lee et al., 2019). In this study, the estimated $H_2SO_4$ concentration was in the range of $10^6$–$10^7$ molecules/cm$^3$ and was theoretically sufficient for NPF (Table 2). Under other conditions, poor correlations are expected between the FR and NMIMP, e.g., with FR > 8 cm$^{-3}$ s$^{-1}$ reported in previous studies (open markers in Fig. 6).

Previous studies have reported that the BVOC emissions over the NCP have increased in the last decade because of the afforestation and accelerating global warming (Stavrakou et al., 2014; Ma et al., 2019). During our observations, the TVOC (including $C_2$–$C_{10}$) mixing ratios approached 16.1 ± 6.5 ppb in the 2018 spring campaign, which was almost double that (including $C_4$–$C_{12}$) in the June 2006 campaign (Mao et al., 2009; no data from the spring 2007 campaign were available). The difference was reasonably consistent with the large increase in forest area over the last decade across the whole China, especially that high BOVC emissions are expected in summer. Note that a discrepancy may also exist between chemical analysis results of VOCs in different labs. However, a large knowledge gap between the increase in BVOC emissions and the increase in nucleating organics still exists because of a lack of studies. Thus, the unexpected response of NPF events to reduced $SO_2$ still unexplained, and more measurements of $H_2SO_4$ and organics (e.g., HOMs) are needed. Note that the campaign average $PM_{2.5}$ mass concentration in 2018 indeed decreased. The decrease was apparently determined by the decrease in >153

nm particles, since no significant difference existed in the calculated CS based on <153 nm particles between 2007 ($0.32 \pm 0.19 \times 10^{-2}\,s^{-1}$) and 2018 ($0.40 \pm 0.15 \times 10^{-2}\,s^{-1}$) .

### 4.2 Question 2: Did the contribution of NPF events to the CCN population decrease considerably with decreasing SO₂?

Based on the observations alone, the $D_{pgmax}$ and the contribution of NPF to the CCN population decreased considerably with decreasing SO₂. However, the growth behaviors of new particles after the new particle signals disappeared from the observations were unknown. Thus, we further analyzed the $D_{pgmax}$ in terms of the correlations with the calculated particle GR, the observation duration of the NPF events on site, and the underlying atmospheric processes.

        Theoretically, the $D_{pgmax}$ should be a function of the GR and the NPF duration. The GR is determined by real-time
concentrations of condensation vapors, whereas the $D_{pgmax}$ is determined by the availability of condensation vapor over a certain long period, both of which are influenced by the concentration of oxidants (Zhang et al., 2012, Apsokardu and Johnston, 2018). In this study, a moderate correlation was observed between $D_{pgmax}$ and GR ($r = 0.58$, $p < 0.01$). The low r value suggested that the GR alone does not determine the $D_{pgmax}$. When one outlier was removed, r increased to 0.66 (Fig. 7a). In addition, the GR had a positive correlation with the total oxidant ($O_x = NO_2 + O_3$) but with an r as low as 0.38 ($p < 0.01$) (Fig. S11).
Additionally, the $D_{pgmax}$ and the duration of NPF events also showed good correlation ($r = 0.67$, $p < 0.01$) (Fig. 7e). Our results imply that both the real-time concentrations and the continuous supply of condensation vapor play dominant roles in the growth of new particles to the CCN size.

        In further analysis, we considered three situations of the new particle growth. Type A (full marker in Fig. 7) represents that new particles continuously grow to the size of $D_{pgmax}$ until the new particle signal drops to a negligible level. Type B
(empty marker) represents the NPF events in which the growth of new particles is similar to that in Type A before $D_{pgmax}$ is reached. After $D_{pgmax}$ is reached, the grown new particles in Type B can still be observed for one more hour, after which either the growth stops for over one hour or the particles start shrinking to a smaller size until the new particle signal disappears. Type C (half-full marker) represents the NPF events that are not subject to Type A or B. Multistage growth of new particles can be observed for Type C particles. A few examples of the three types are shown in Fig. S12. We also separated the
observations in 2017 and 2018 (in red) from those in 2007–2015 (in blue).

        For Type A, the average GR and $D_{pgmax}$ in 2017–2018 were only 1.5 nm h⁻¹ and 23 nm, respectively, significantly lower than the 3.5 nm h⁻¹ and 48 nm values observed in 2007–2015 ($p < 0.01$). When the regression equation of the GR and $D_{pgmax}$ is examined, i.e., $D_{pgmax} = 11.0 \times GR + 8.2$ with a moderate good $r$, newly formed particles appear to grow beyond 50 nm only when the GR exceeds 3.8 nm h⁻¹ in Type A. There was no significant difference between the duration of NPF events in 2017–
2018 and that in 2007–2015. However, based on the regression equations between the duration and $D_{pgmax}$ obtained in 2007–2015 and 2017–2018, newly formed particles could grow beyond 50 nm only when the NPF duration exceeded 9.9 h in 2007–2015, but the duration in 2017-2018 had to exceed 27.8 h. As reported in the literature, the lifetime of 50 nm particles in the

boundary layer is approximately one day, while that in the free troposphere is much longer (Williams et al., 2002). It can be argued that the new particles in Type A of 2007–2015 may still have been able to grow to the CCN size even after the new

particle signal disappeared from the observation. However, the lifetime of 20 nm particles in the boundary layer is only several hours (Williams et al., 2002). If the Type A NPF events in 2017–2018 occurred in the boundary layer, the new particles may not have been able to grow to the CCN size before being scavenged from the ambient air. If they occurred in the free troposphere, the longer lifetime may have allowed the new particles in some events to be able to grow to the CCN size. For example, the NPF event on March 21, 2018 ended with an increase in the wind speed and a change in the wind direction, and

the number concentration of new particles remained at a high level. The air mass back trajectories passed through the NCP at a high altitude (>1700 m a.g.l) at the beginning and end of the NPF event (Fig. S12a, b). In addition, the spikes of PNSDs during this NPF event indicated the vertical transport of atmospheric particles (Meng et al., 2015; Shen et al., 2020). We inferred that this NPF event seemingly occurred in the free troposphere, and a large decrease in the entrainment from the free troposphere to the boundary layer may have led to the disappearance of the new particle signal. Recent aircraft and airship

measurements in northern and eastern China suggested that NPF events sometimes occurred in the free troposphere and then mixed down to the boundary layer (Quan et al., 2017; Qi et al., 2019).

In the case of Type B, the GR and $D_{pgmax}$ in 2017–2018 (the mean values of 3.2 nm h$^{-1}$ and 29 nm, respectively) were significantly lower than those in 2007–2015 (6.1 nm h$^{-1}$ and 56 nm, respectively; $p < 0.01$). Following the regression equation of $D_{pgmax}$ against GR, newly formed particles in Type B could grow beyond 50 nm only when the GR exceeded 5.8 nm h$^{-1}$.

The number concentrations or the sizes of new particles decreased considerably at the end of Type B NPF events, and the transient time for the decrease suggested that the events occurred in the boundary layer. For example, the air mass back trajectories at the end of the NPF on April 7, 2018 originated from low altitude, and the height varied greatly over time (Fig. S12c, d). Most of the Type B NPF events in 2017–2018 may have had less opportunity to contribute to the CCN population, if they indeed occurred in the boundary layer. However, aircraft measurements are needed to confirm the altitude at which the

NPF events occur. In addition, the changed boundary layer height had no detectable influence on $D_{pgmax}$ as shown in Fig S13. However, the change in the late afternoon may largely decrease the observed number concentrations of grown new particles.

Type C was characterized by the largest GR, duration, and $D_{pgmax}$, with mean values of 7.7 nm h$^{-1}$, 22 h, and 92 nm, respectively. These particles underwent multiple growth processes, complicating the correlation between GR and $D_{pgmax}$, and that between duration and $D_{pgmax}$. The air mass back trajectories at the end of the NPF event on September 30, 2014 were local

and originated at a low altitude (Fig. S12e, f), implying that these new particles experienced sufficient growth within the boundary layer. There were 16 Type C NPF events during 2007–2015 and only two in 2017. The lack of Type C NPF events in 2017–2018 implies a significant decrease in the contribution of new particles to the CCN population.

The factors influencing the lower $D_{pgmax}$ and NPF-derived CCN population in 2017–2018 were further explored. In the literature, the growth of newly formed particles is mainly attributed to sulfuric acid, ammonium nitrate, and secondary organic

compounds (Wiedensohler et al., 2009; Riipinen et al., 2011; Zhang et al., 2012; Ehn et al., 2014; Man et al., 2015; Wang et

al., 2015; Burkart et al., 2017; Lee et al., 2019; Wang et al., 2020). As listed in Table 2, the contribution of $H_2SO_4$ vapor to particle growth decreased from 36% in the spring of 2007 to 11% in the spring of 2018, indicating an inevitable consequence of decreasing $SO_2$ emission on particle growth. However, this percentage is likely not high enough to explain the 50% decrease in the GR and $D_{pgmax}$ of Types A and B. On the other hand, the reduction in $SO_2$ and sulfate may reduce the aerosol acidity,

subsequently affecting the acid-enhanced uptake of semivolatile organic species (Ding et al., 2011; Stangl et al., 2019). This hypothesis is supported by the lower level of organic carbon (OC) in $PM_{2.5}$ found in the spring of 2018 ($5.5 \pm 2.0$ µg m$^{-3}$) than that in the spring of 2007 ($6.1 \pm 3.0$ µg m$^{-3}$), although the BVOC emissions over the NCP have reportedly increased in recent years (Table 2, Wang et al., 2011; Stavrakou et al., 2014; Ma et al., 2019; Dong et al., 2020). Furthermore, the mass concentration of nitrate in $PM_{2.5}$ was $7.4 \pm 4.8$ µg m$^{-3}$ in 2007 during the new particle growth period, and it slightly decreased

to $6.7\pm 5.5$ µg m$^{-3}$ in 2018. The reduced nitrate may also be partially responsible for the lack of Type C in 2018. In summary, we argued that the reduced $H_2SO_4$ vapor, nitrate and OC formation (most likely because of reduced anthropogenic emissions) may have led to the SP of new particles in the spring campaign of 2018 being lower than that of 2007. Unfortunately, chemical data about size-segregated molecular constituents are not available to confirm this finding, and therefore, more refined observations are urgently needed in the future.

However, uncertainties still exist, e.g., 1) the data were obtained in seven independent campaigns, each lasting in 18~71 days, and the data size did not allow us to extend the conclusion to all the years from 2007 to 2018; 2) the observations were conducted only at one site, alternating between the boundary layer and the free troposphere, and the generality of the conclusions on NPF events needs to be examined at more sites.

## 5. Conclusions

With an order of magnitude reduction in $SO_2$ emissions, the NPF frequency observed at the summit of Mt. Tai remained relatively constant during the seven campaigns of 2007–2018. The calculated campaign-based FR and NMINP were 0.8–1.2 cm$^{-3}$ s$^{-1}$ and $3.8$–$5.1 \times 10^3$ cm$^{-3}$ in 2007–2014 and then unexpectedly increased by a factor of 2–3 in 2017–2018 to 2.0–3.0 cm$^{-3}$ s$^{-1}$ and $0.9$–$1.1 \times 10^4$ cm$^{-3}$, respectively. However, the large increase in the NPF intensity was accompanied by a smaller probability of the particles growing to the CCN size. The number concentrations of NPF-derived CCN with the three threshold

sizes of 50, 80, and 100 nm were estimated to be $3.7\times10^3$, $1.6\times10^3$, and $8.6\times10^2$ cm$^{-3}$ in the 2007 - 2015 campaigns, which then decreased to $1.0\times10^3$, $4.6\times10^2$, and $1.8\times10^2$ cm$^{-3}$, respectively, in the 2017 - 2018 campaigns. When the three types of NPF events are separately considered, it remains uncertain whether the new particles in Type A can grow to the CCN size after the disappearance of the new particle signals from observations. No conclusion can be drawn on this issue based on the current limited chemical observations. However, the new particles in Type B may have less opportunity to grow to the CCN size

before they are scavenged from the ambient air. The lack of Type C NPF events in the campaigns of 2017–2018 indicates a large decrease in the probability of new particles growing to the CCN size with the reduction in ambient air pollutants.

Moreover, the shorter durations of the NPF events in the campaigns of 2017–2018 imply that the events occurred over a smaller spatial scale.

We hypothesize that the NPF intensity increased unexpectedly with the reduction in $SO_2$ concentration, as the net production of NPF seems to be determined mainly by the availability of organic precursors for participating in nucleation and initial growth. This is reasonably consistent with the increase in forest areas over the last decade across China through rapid afforestation. The strong correlation between the FR and NMINP strongly supports this hypothesis, which needs to be further confirmed by direct observations of molecular organic vapors. The decrease in the percentage of new particles growing to the CCN size with increasing NPF intensity in 2017–2018 implies the complexity of the growth of new particles with reduced emissions of anthropogenic precursors under a large-scale increase in BVOC emissions. Overall, this study provides unique observational results regarding NPF at a regional mountain-top site in the NCP from reasonably large datasets. Based on the unique results, we comprehensively analyzed the possible causes, and proposed new challenges in determining the underlying mechanisms of the contributions of new particles to ambient particle number loading and CCN populations with reduced anthropogenic emissions.

**Data availability.** The datasets related to this work can be accessed via https://data.mendeley.com/datasets/wf3wjvpfw7/draft?a=adc7fab3-93b4-4f23-a075-904fe9729cf6.

**Author contributions**. LX designed the research. JC and JG conducted the field observations in 2007, 2014 and 2015. XW, HL, YZ, ZG, TC, LW, PZ, and YS carried out the field measurements in 2009, 2017 and 2018. YZ analyzed the data and wrote the paper. XY, TW and WW helped to interpretation of the results. XY and LX revised the original manuscript. All authors contributed toward improving the paper.

**Competing interests.** The authors declare that they have no conflict of interest.

**Acknowledgements**

This work was funded by the National Key Research and Development Program of China (2016YFC0200500), the National Natural Science Foundation of China (41922051, 42075104, 41706122), Shandong Provincial Science Foundation for Distinguished Young Scholars (ZR2019JQ09), and the Jiangsu Collaborative Innovation Center for Climate Change. We appreciate the NOAA Air Resource Laboratory for providing the HYSPLIT model, and thank the staff of the Mt. Tai Meteorological Station for the help during the measurement campaigns.

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

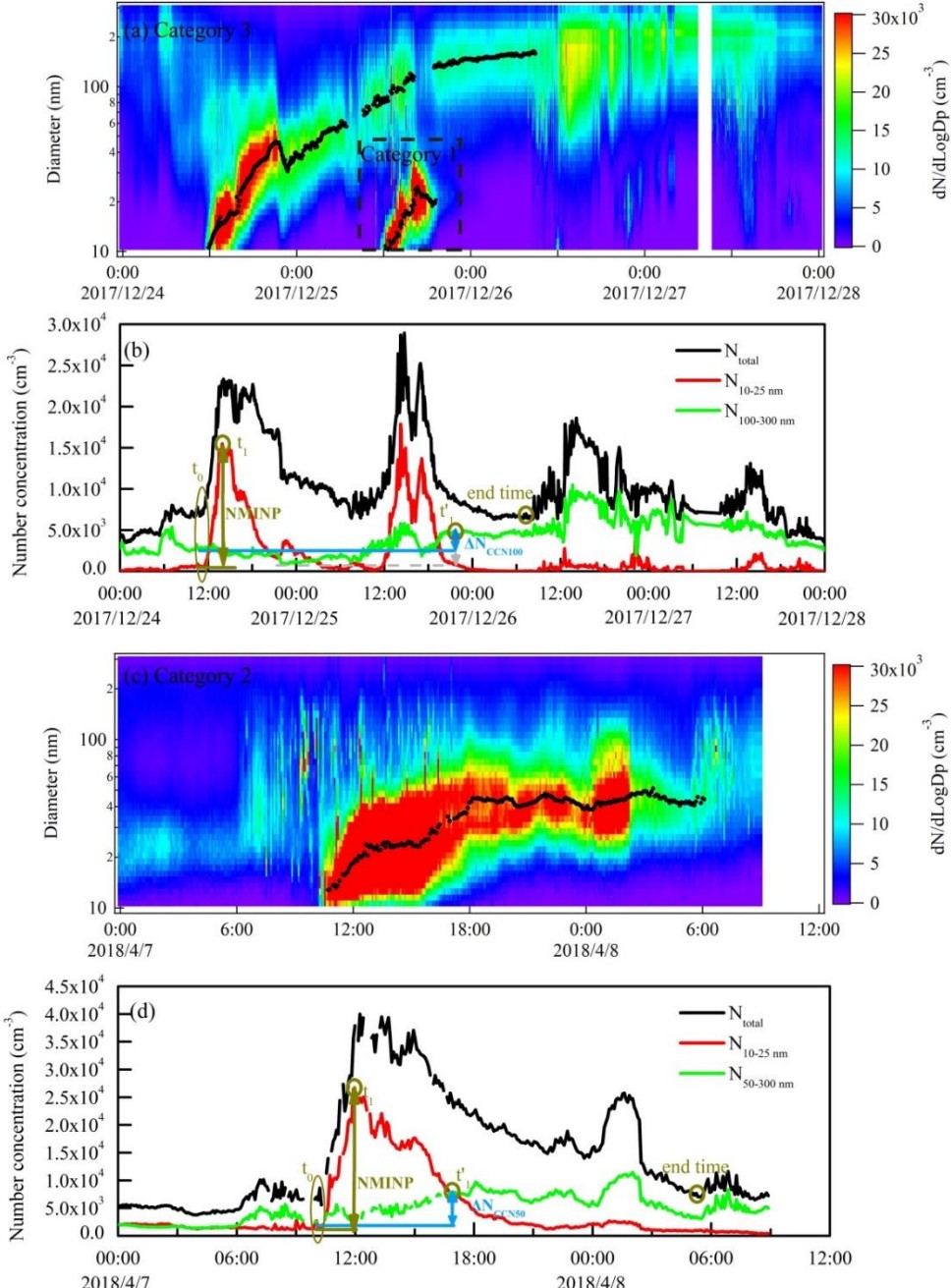

**Figure 1: Examples of NPF events in three categories. Black dots in the figures are the fitted $D_{pg}$. (a) Category 1 on December 25, 2017, in which $D_{pgmax}$ was 24 nm (<50 nm), and Category 3 on December 24, 2017, in which $D_{pgmax}$ grew to 163 nm (>80 nm); (c) Category 2 on April 7, 2018, in which $D_{pgmax}$ grew to 53 nm (50–80 nm). (b, d) Schematic diagram of $t_0$, $t_1$, $t'_1$, NMINP and $\Delta N_{CCN100}$/ $\Delta N_{CCN50}$ on December 24, 2017 and April 7, 2018 NPF events (a few spikes have been removed from Figure d).**

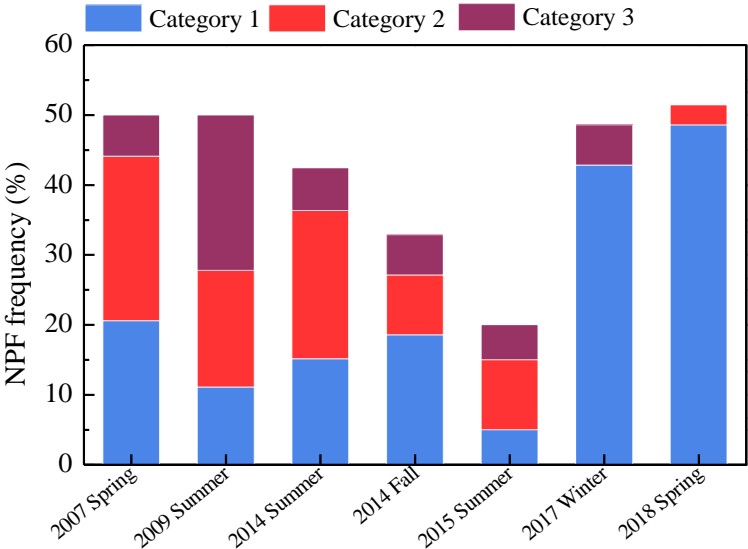

**Figure 2: Occurrence frequencies of NPF events in different categories at Mt. Tai during the seven observation campaigns.**


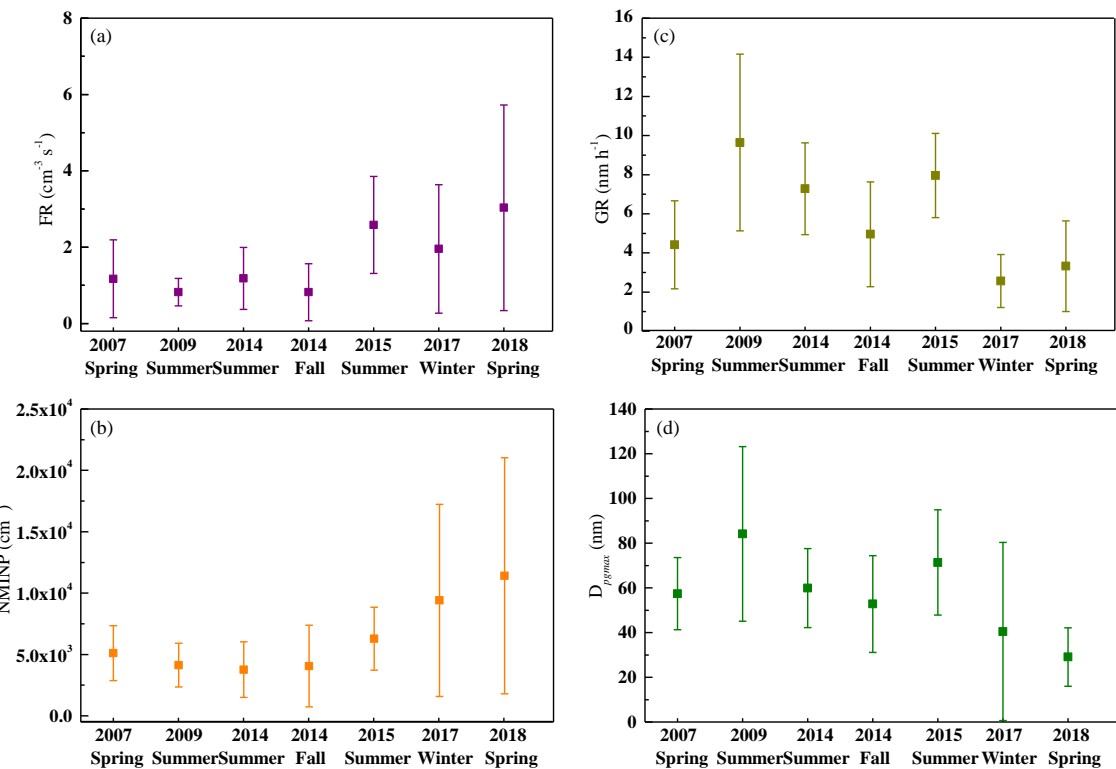

Figure 3: Campaign average of the new particle formation rate (FR, a), the net maximum increase in the nucleation-mode particle number concentration (NMINP, b), the new particle growth rate (GR, c), and the maximum geometric median diameter of the grown new particles (D$_{pgmax}$, d) during the seven observation campaigns. The error bars are the standard deviations.

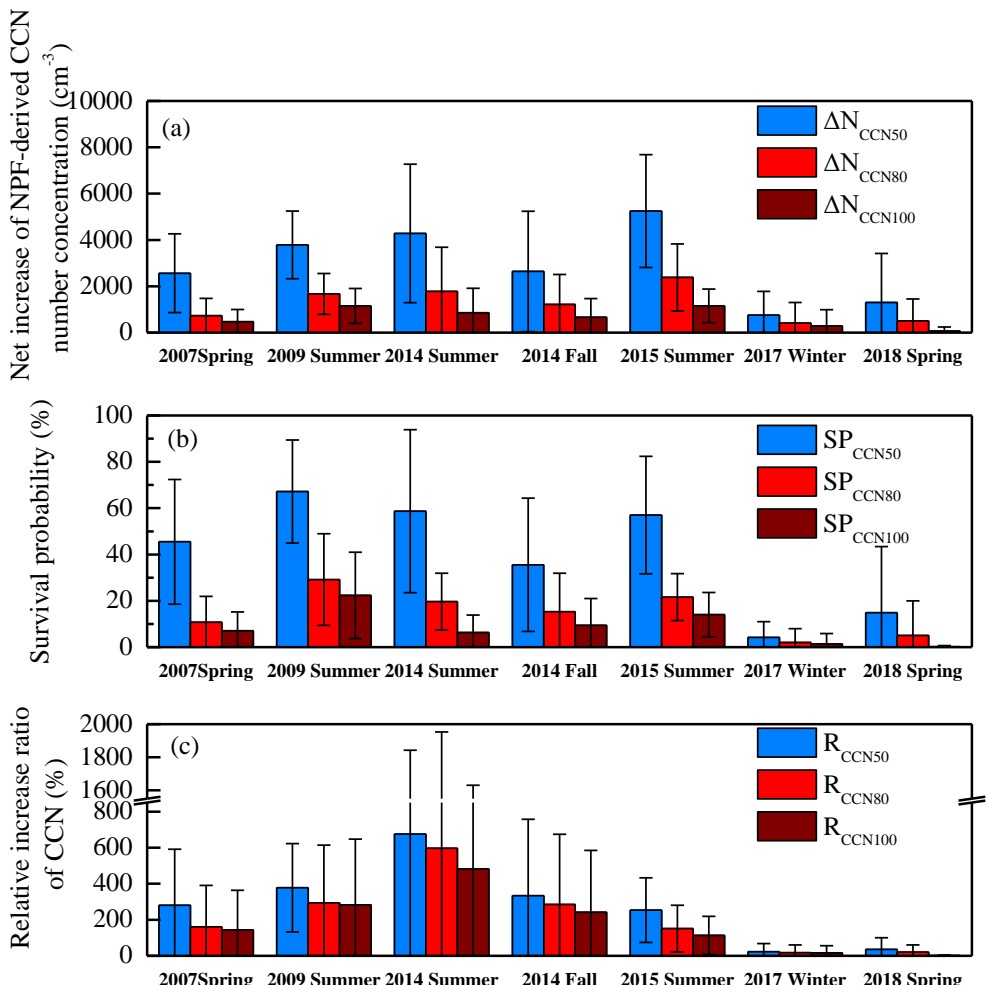

**Figure 4: Campaign average of the net increase in the NPF-derived CCN number concentration (ΔN_CCN, a), the survival probability of new particles growing to CCN sizes (SP_CCN, b), and the relative increase ratio of the CCN population (R_CCN, c) during the seven observation campaigns. The error bars are the standard deviations.**

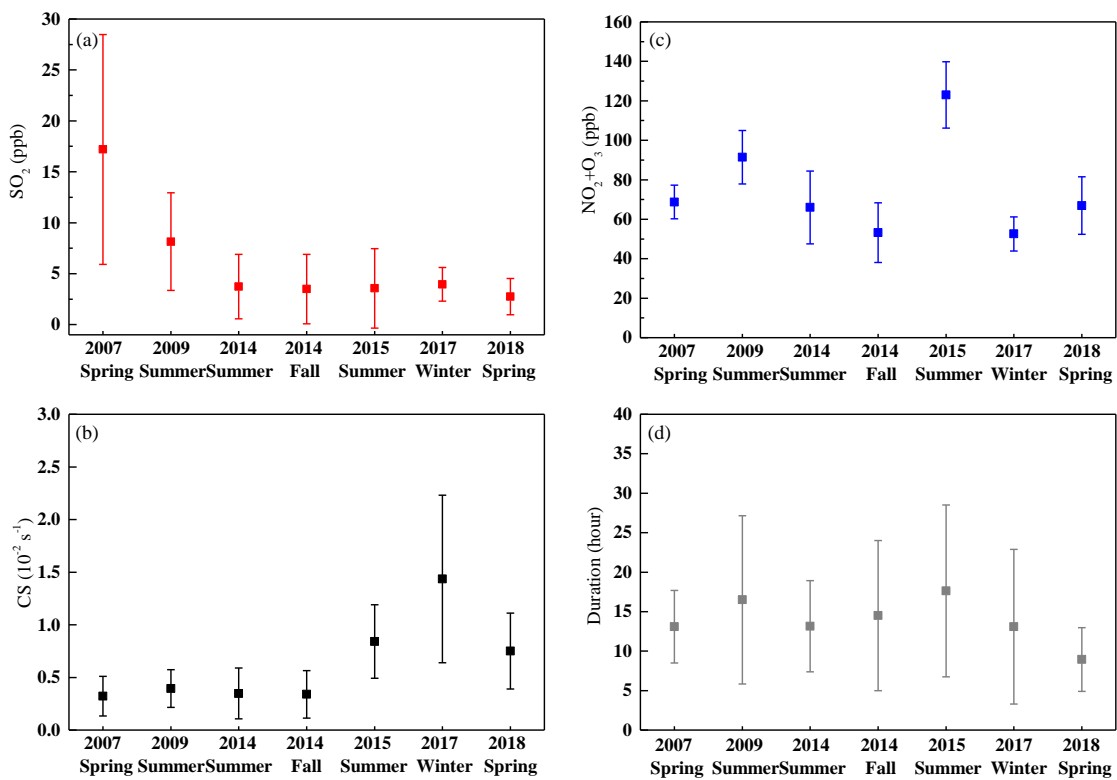


**Figure 5: Campaign average of SO₂ mixing ratios (average during NPF periods, a), CS (one hour prior to NPF events, b), NO₂ + O₃ (average during new particle growth periods, c), and NPF event durations (d) during the seven observation campaigns. The error bars are the standard deviations.**

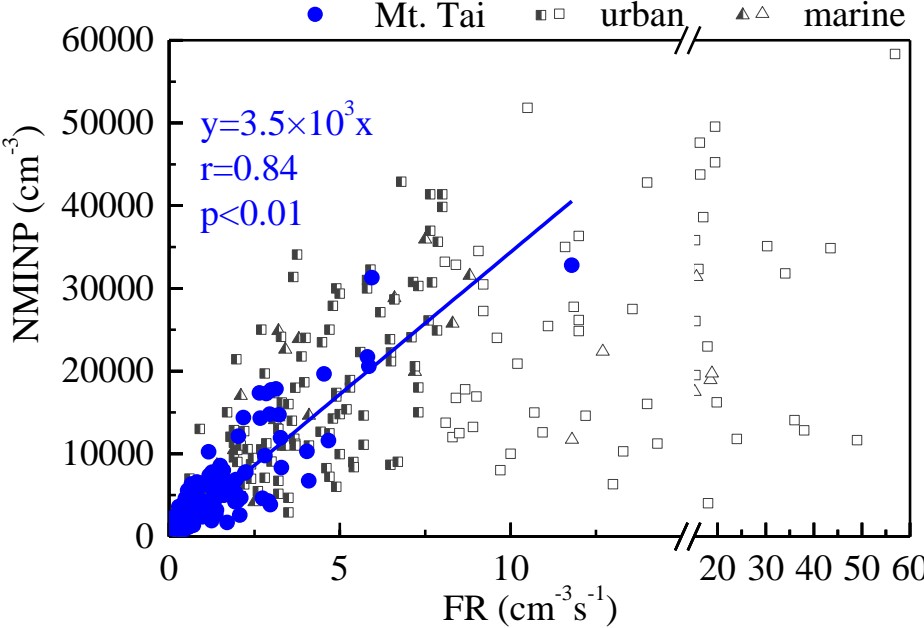

**Figure 6: Relationship between the FR and NMINP in 106 cases of NPF events at Mt. Tai in this study and in urban and marine atmospheres in previous studies (Man et al., 2015; Zhu et al., 2017, 2019; Ma et al., 2020). The half-solid markers can be fitted linearly in previous studies. The open markers show poor correlations.**

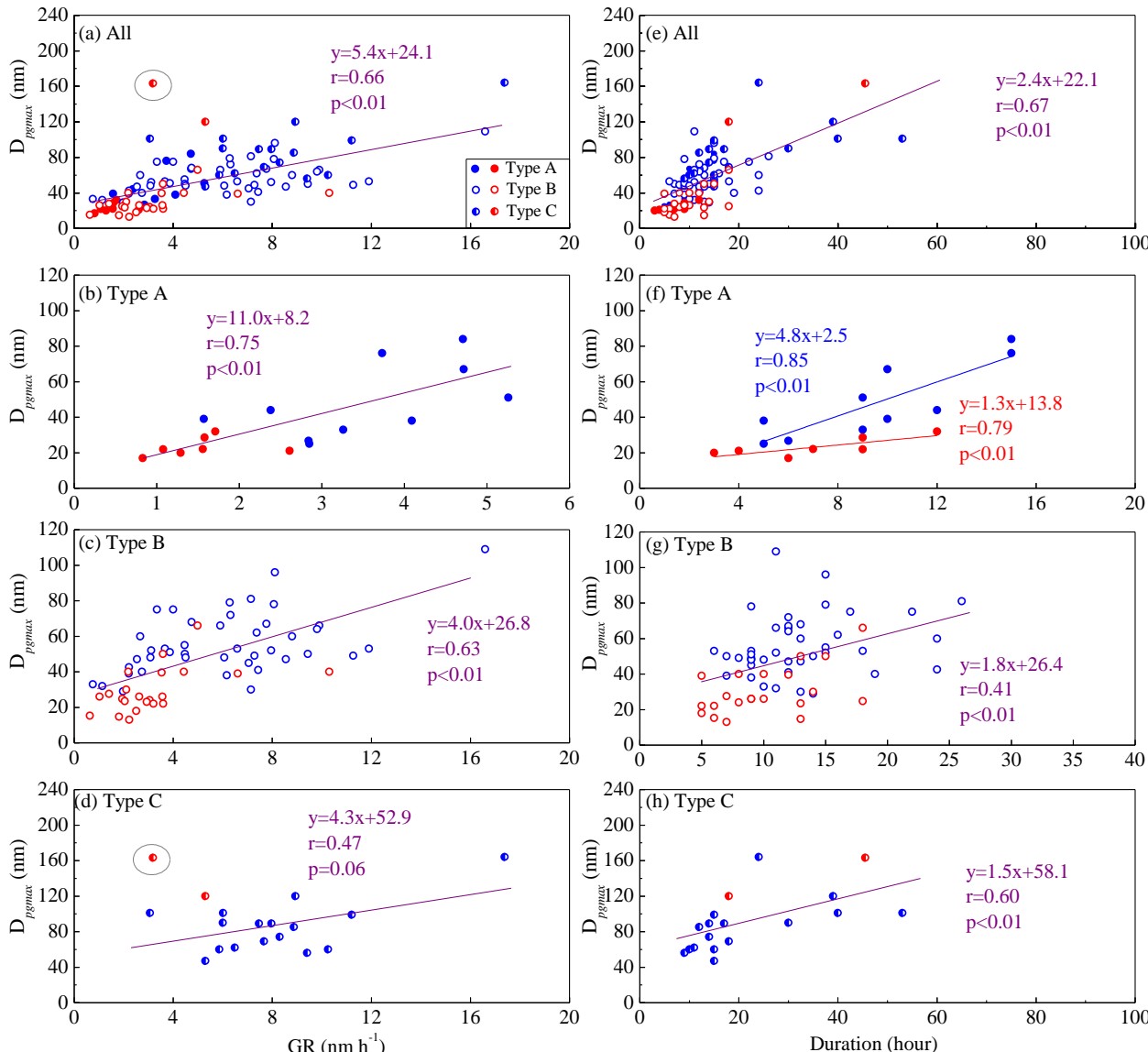


**Figure 7: Relationship between the GR and $D_{pgmax}$ (a–d) and between the duration of NPF events and $D_{pgmax}$ (e–h). Solid marker represents Type A; empty marker represents Type B; and half-solid marker represents Type C. Red markers and equations represent 2017 and 2018, and blue markers and equations represent 2007–2015. The purple equation represents fitting of all data, and the circled marker represents an outlier from the equation fit.**


**Table 1. Summary of the seven observation campaigns at Mt. Tai.**

| Campaign | Species | Instruments | Resolution |
|---|---|---|---|
| Spring 2007<br>3/22–4/24, 2007 | PNSD in 10 nm–10 μm | WPS, MSP 1000XP | 8 min |
| | $SO_2$, $O_3$, NO, and $NO_2$ | Thermo 43C, 49C, 42C | 1 min |
| | $PM_{2.5}$ | TEOM 1400a | 1 min |
| | Water-soluble ions in $PM_{2.5}$ | URG-AIM 9000B | 1 h |
| Summer 2009<br>6/12–6/29, 2009 | PNSD in 10 nm–10 μm | WPS, MSP 1000XP | 8 min |
| | $SO_2$, $O_3$, NO, and $NO_2$ | Thermo 43C, 49C, 42C | 1 min |
| Summer 2014<br>7/26–8/27, 2014 | PNSD in 5 nm–10 μm | WPS, MSP 1000XP | 5 min |
| | $SO_2$, $O_3$, NO, and $NO_2$ | Thermo 43C, 49C,42i | 1 min |
| | $PM_{2.5}$ | Thermo 5030 SHARP | 1 min |
| | Ions in $PM_{2.5}$, acid, and alkaline gases | MARGA, ADI20801 | 1 h |
| Fall 2014<br>9/21–11/30, 2014 | PNSD in 5 nm–10 μm | WPS, MSP 1000XP | 5 min |
| | $SO_2$, $O_3$ | Thermo 43C, 49C | 1 min |
| | NO, $NO_2$ | API T200U, T500U | 1 min |
| | $PM_{2.5}$ | Thermo 5030 SHARP | 1 min |
| | Ions in $PM_{2.5}$, acid, and alkaline gases | MARGA, ADI20801 | 1 h |
| Summer 2015<br>6/16–7/25, 2015 | PNSD in 5 nm–10 μm | WPS, MSP 1000XP | 5 min |
| | $SO_2$, $O_3$ | Thermo 43C,49C | 1 min |
| | NO, $NO_2$ | API T200U, T500U | 1 min |
| | $PM_{2.5}$ | Thermo 5030 SHARP | 1 min |
| | Ions in $PM_{2.5}$, acid, and alkaline gases | MARGA, ADI20801 | 1 h |
| Winter 2017<br>11/26–12/30, 2017 | PNSD in 5 nm–10 μm | WPS, MSP 1000XP | 5 min |
| | $SO_2$, $O_3$ | Thermo 43C, API 400U | 1 min |
| | NO, $NO_2$ | API T200U, T500U | 1 min |
| | $PM_{2.5}$ | Thermo 5030 SHARP | 1 min |
| | Ions in $PM_{2.5}$, acid, and alkaline gases | MARGA, ADI20801 | 1 h |
| Spring 2018<br>3/5–4/8, 2018 | PNSD in 5 nm–10 μm | WPS, MSP 1000XP | 5 min |
| | $SO_2$, $O_3$ | Thermo 43C, API 400U | 1 min |
| | NO, $NO_2$ | API T200U, T500U | 1 min |
| | $PM_{2.5}$ | Thermo 5030 SHARP | 1 min |
| | Ions in $PM_{2.5}$, acid, and alkaline gases | MARGA, ADI20801 | 1 h |

**Table 2. Meteorological conditions and air pollutants during the formation and growth periods of new particles in the spring campaigns in 2007 and 2018.**

| Parameters | 2007 spring | | 2018 spring | |
|---|---|---|---|---|
| | Formation | Growth | Formation | Growth |
| T (℃) | $5.8 \pm 3.2$ | $7.1 \pm 3.3$ | $3.5 \pm 5.8$ | $5.3 \pm 5.9$ |
| RH (%) | $54 \pm 22$ | $52 \pm 18$ | $45 \pm 17$ | $46 \pm 17$ |
| $SO_2$ (ppb) | $16.7 \pm 10.9$ | $20.2 \pm 13.0$ | $2.6 \pm 1.8$ | $2.5 \pm 1.5$ |
| $NH_3$ (ppb) | $12.6 \pm 18.0$ | $11.2 \pm 17.0$ | $6.5 \pm 9.5$ | $6.6 \pm 7.2$ |
| $NO_2 + O_3$ (ppb) | $63.7 \pm 8.4$ | $70.1 \pm 9.7$ | $61.3 \pm 14.0$ | $63.8 \pm 14.3$ |
| $PM_{2.5}$ ($\mu g\ m^{-3}$) | $56.5 \pm 33.0$ | $71.1 \pm 49.0$ | $30.3 \pm 21.8$ | $29.2 \pm 20.4$ |
| $SO_4^{2-}$ ($\mu g\ m^{-3}$) | $16.4 \pm 11.0$ | $18.5 \pm 9.7$ | $3.3 \pm 2.4$ | $3.6 \pm 2.7$ |
| $NO_3^-$ ($\mu g\ m^{-3}$) | $7.4 \pm 5.7$ | $7.4 \pm 4.8$ | $6.3 \pm 5.1$ | $6.7 \pm 5.5$ |
| $NH_4^+$ ($\mu g\ m^{-3}$) | $5.5 \pm 4.2$ | $6.1 \pm 3.5$ | $2.3 \pm 1.7$ | $2.2 \pm 1.6$ |
| Calculated $H_2SO_4$ ($10^7$ molecules·$cm^{-3}$) | $8.8 \pm 4.9$ | $9.4 \pm 4.5$ | $2.2 \pm 1.1$ | $2.4 \pm 1.0$ |
| $[H_2SO_4]_{avg}$/C (%) | $59 \pm 23$ | $36 \pm 18$ | $23 \pm 10$ | $11 \pm 7$ |
| TVOC (ppb) | $7.0 \pm 5.7$[a] | | $16.1 \pm 6.5$ | |
| OC ($\mu g\ m^{-3}$) | $6.1 \pm 3.0$[b] | | $5.5 \pm 2.0$[c] | |
| EC ($\mu g\ m^{-3}$) | $1.8 \pm 1.6$[b] | | $1.3 \pm 0.6$[c] | |

[a](Mao et al., 2009)

[b](Wang et al., 2011)

[c](Dong et al., 2020)