# Peer review of "Increased new particle yields with largely decreased probability of survival to CCN size at the summit of Mt. Tai under reduced SO2 emissions"

_Atmospheric Chemistry and Physics, 2020_

## Referee Comment (RC1) · Anonymous Referee #3 · 15 Aug 2020

The manuscript analyzes seven field campaigns where particle number size distributions (PNSD) and sulfur dioxide were measured at the summit of a mountain site in the North China plain. Supporting measurements of time-resolved PM2.5, O3, and oxides of nitrogen were taken. And each campaign included 1-h time resolution ions in PM2.5 using water extractive methods (URG-AIM or MARGA). The most recent campaign was in 2018. Across the 7 campaigns, a little over 100 particle formation and growth events were detected, with the analysis focused on the size range of 10-300 nm size range. From the earliest to most recent campaign, SO2 emissions and concentrations have dropped dramatically, and the paper tries to analyze whether the particle formation and growth activity has changed in ways that are expected from the sulfur dioxide

decrease. A large number of metrics are computed and then analyzed for each particle formation and growth event (PFGE). The metrics include, but are not limited to, the apparent formation rate of 10-25 nm particles (FR), the growth rate, the absolute increase in N10-25 particle concentration from the start to the peak of the PFGE (this is the NMINP variable), the PFGE duration, the PFGE frequency, the size to which the growth event reaches (Dpgmax variable in the manuscript), and particle counts which are used as surrogates for the change in CCN concentrations at low, medium, and high supersaturations (N100-300, N80-300, and N50-300). The paper includes values for and discussion of total VOC during the campaigns.

Complicating the analysis is that the field campaigns were in different months of the year: April 2007 ($\sim$30 d), June 2009 ($\sim$20 d), Aug 2014 ($\sim$30 d), Oct/Nov 2014 ($\sim$70 d), Jul 2014 ($\sim$40 d), Dec 2017 ($\sim$35 d), and Mar 2018 ($\sim$30 d).

The paper's abstract makes five claims: a. The formation rate in 2018 is 2-3 times higher than the formation rate in 2007. b. Net maximum increase in nucleation mode number concentration is 2-3 times higher in 2018 than in 2007. c. The occurrence of events where the mode of the growth event goes above 50 nm is lower in 2018 than it was in 2007. d. A surrogate for CCN production at high supersaturation (N50-300 at its peak during each growth event minus N50-300 before the event) decreased from 3703 per cm3 (before 2015) to 1026 (2017-2018). e. The authors argue availability of organic precursors has increased in the most recent campaigns, allowing more particle production and initial growth; furthermore, they argue that the lack of later growth is from reduction of "anthropogenic precursors" (presumably SO2).

The paper requires substantial revision before it is suitable for publication. The key issue, to this reviewer, is that making accurate claims about year-on-year trends and variability in PFGE is difficult. The requirements to make the claims defensible are: (1) take a sufficient number of samples to reduce random variability and give sufficient statistical power; (2) take steps to minimize, test for, and quantify campaign-specific systematic instrument bias (also known as "instrument drift"); (3) take steps to enforce

consistency in any subjective data interpretation steps, such as classification of PFGE into "types" and the determination of the start and end times of events; (4) use statistical methods designed for trend analysis, time series analysis, and combined analysis of seasonal and interannual variability.

Each requirement needs to be met in order for the claims about trends to be defensible. And for peer review and reproducibility purposes, things need to be documented for the peer-review and scientific communities.

I think the current work fails to meet all four of the requirements. While some of the conclusions are likely accurate (in that they would not change if all the requirements were met) – others would change, or require extensive qualification.

1. Statistical power:

PFGE exhibit substantial seasonal variation, due to changes in temperature, relative humidity, biogenic activity, atmospheric chemistry, soil moisture, preexisting aerosol concentration and chemistry, radiation, cloudiness, boundary layer structure, land cover/vegetation canopy structure, synoptic meteorology, anthropogenic emissions, and atmospheric ion levels. Local meteorological features (i.e. orographic meteorology) and local sources may also have month-to-month variability. And at the 20-30 d time scale, large scale persistent geophysical features can cause a whole campaign of measurements to be atypically high or low for a number of PFGE variables. To accommodate all these sources of variability, large sample sizes are required for analysis of seasonal variation and interannual trends. In the absence of large sample sizes, careful pairing of events and analysis of alternate sources of variability / alternate hypotheses are needed isolate cause-effect relationships on specific PFGE variables.

With each campaign at a slightly different time of the year, some campaigns as short as ∼20 d, and no discussion of whether air pollution levels, air pollution meteorology, and climate variables were at climatologically representative levels, the reader has to apply great skepticism to any claims of interannual trends and cause-effect relationships for

those interannual trends. See for example (Birmili and Wiedensohler 2000) who do take into account air mass characteristics.

The size distributions shown in Figure S3 are suggestive of insufficient number of days sampled in the dataset. Telling whether the system shifted from unimodal to bimodal behavior between 2015 and 2017 (all unimodal for 2015 and prior) vs. this occurring through some instrument drift vs. this occurring through sampling non-climatological conditions due to small samples sizes is difficult.

Given the decrease (Table 2) in PM2.5, sulfate, and SO2 between spring 2007 and 2018 (PM2.5 60 vs. 30 ug/m3; sulfate 17 vs. 4 ug/m3, SO2 18 vs. 3 ppb), more discussion is needed of the large increase in condensation sink in 2018 (Figure S3) and in the large increase in the height of the size distribution function at 100 and 150 nm between 2007 and 2015.

The discontinuity in the slope of the size distribution function at 200 nm also indicates there may be some drift in the size-specific performance of the WPS (Figure S3). The discontinuity in slope is not really evident until 2017, but then appears in 2017 and 2018.

2. Minimize, test for, and quantify campaign-specific instrument drift:

Achieving consistency in PNSD in long-term measurements is difficult. And it is not sufficient to state that each individual campaign had sufficient quality assurance, referring the reader to the campaign specific papers. There needs to be a presentation of data and discussion of how comparable the instrument responses are from campaign to campaign. What steps were taken to make sure instruments were not drifting. Aging of components can cause variation in flows, sizing accuracy, counting accuracy, particle losses, CPC supersaturations, and in the effective lower size limit of the instrumentation of the particle number spectrometer system. The detection efficiency as a function of size at the lower range of the instrument (5-25 nm), at the upper range of the mobility analyzer, at the lower end of the optical particle devices, and at the upper end

of the optical particle analyzer – these are all difficult to maintain at stable levels over long periods of time. The total particle counts, height of the size distribution function, sensitivity at the lower and upper ranges of size distributions – these vary from year to year and require careful intercomparison, quality assurance, and maintenance procedures to deal with. See for example the results of intercomparison studies (Pfeifer, Müller et al. 2016) and papers focusing on quality assurance, calibration, and harmonization (Pitz, Birmili et al. 2008, Wiedensohler, Birmili et al. 2012, Wiedensohler, Wiesner et al. 2018, Gaie-Levrel, Bau et al. 2020). Comparison to other instruments for total particle counts, size distribution functions in overlapping regions, checks with monodisperse particles are some of the techniques that can be used to establish more confidence and quantify campaign-to-campaign comparability.

Consistency in inlet dimensions, inversion algorithms (including multiple charge correction), use of impactors to manage multiple charge issues, corrections for inlet transmission efficiency, – these can all be issues in campaign-to-campaign comparability. They need to be discussed.

While being able to reproduce time-resolved PM2.5 measurements from the WPS size distribution is not sufficient to show accuracy in the nucleation and Aitken ranges – it is probably necessary. At least showing consistency from campaign to campaign in the volume of particles measured by the WPS and the mass of particles by time-resolved mass measurements can help to demonstrate stability in instrumentation and data processing algorithms.

The fact that the authors are using an instrument with nominal lower cutoff of 5 nm, but discarding data between 5-10 nm indicates that there may be a problem with sensitivity at the lower size limit, or (more likely) variability in the sensitivity at the lower size limit. There is further evidence in Figures 1 and S6 – of a problem. In all the bursts shown save one, the particle size distribution function slopes down from a peak at about 13 nm to a lower value at 10 nm. If the instrument is biased low in the 10-13 nm range, then the statistics developed in the work will also be biased. If that bias varies from

campaign to campaign, then that creates additional interpretation difficulties.

At line 180, it is implied that at times the WPS was collocated with instruments with lower limit of 3 nm. Therefore, the actual performance in the 5-15 nm range could (and should) be determined though comparison to such collocated instruments.

3. Consistency in subjective data interpretation/classification steps:

It is not clear which of the variables used for analysis involve human classification. Sometimes, human classification is used for PFGE types (often using how smooth the growth event is in time); human classification is used sometimes for establishing times (start of event, end of the event). The end time is described. From line 113 of manuscript, "The end time of an NPF event was defined as the time when the particle number concentrations approached the background levels observed before the NPF event. The NPF event duration was defined as the time duration between the start time and end time of an NPF event." This seems like the end of event was a subjective determination of when background was approached. Thus the end time, duration, and any rate that has the duration in the denominator may be subjective.

For subjective (human) event classifications, were the events uniformly reclassified for this paper, or were prior classifications adopted from 2007 and 2009 and mixed with new classifications done for the more recent campaigns. See (Dal Maso, Kulmala et al. 2005) for best practices on human classification.

4. Statistical methods appropriate to analysis of combined seasonal and interannual variability

Statistical procedures for evaluating trends in seasonally varying time series need to be followed in order to state claims that trends exist. These can be found in a number of textbooks, papers, and government reports. See for example Statistical Methods for Environmental Pollution Monitoring by Gilbert https://www.osti.gov/servlets/purl/7037501/. And (Asmi, Coen et al. 2013, Collaud

Coen, Andrews et al. 2013, Squizzato, Masiol et al. 2019). Many other good models for seasonally adjusted trend detection can be found in the O3, NOx, PM2.5, and hydrology/climatology literature. Squizzato et al. (2019) for example have the statistical procedures necessary to detect turning points (see line 236 where manuscript discusses turning points)

See for example line 290 "the CS still increased in 2018 compared with that in 2007." That implies annual average condensational sink increased, and this is a season or month specific result – and it is not clear there is enough statistical confidence to state this. Many other locations in the paper have broad statements about PFGE behavior in one year vs. another, or imply a long term trend where it has not really been shown.

Interpretation of PFGE data from this site seems more complicated than most, because of two issues: (1) it is sometimes influenced by boundary layer and other times by free troposphere; (2) very long PFGE events (see for example Figure 1a, where a 3-d long event is shown) are being compared with shorter (midday + afternoon) growth events. See Figure 7 which has events ranging from 3-h duration to 85-h duration. The flow patterns and chemistry required to sustain a 3-h event and an 80+ h event are likely very different, and would require more thoughtful comparison metrics than used in the paper.

The paper acknowledges this difficulty in interpretation (line 295) but more needs to be done than just acknowledge the difficulty. See analysis papers from PFGE studies at other high altitude sites. They do attempt to determine the degree of FT influence and the impact of polluted boundary layer air. And there are many papers that factor in air mass characteristics and/or back trajectory in analysis of PFGE.

See for example Figure 1a where on 25-Dec 2017 there were simultaneously occurring a short PFGE (category 1) and evolution of the category 3 event that started on 24-Dec 2017. This raises a number of questions on how such a dataset can be analyzed to determine trends.

How much of the variability in data is that some campaigns had more free tropospheric influence and others have less. How much of the conclusions of the paper are driven by switches (during PFGE) in air mass influence to/from FT influence. In other words, PFGE events that have their evolution dynamics controlled by airflows, and not by chemistry – hence the authors observed lack of influence or counterintuitive effects of SO2.

As for statistical procedures, I think it would be much more appropriate to put 95% confidence intervals on means rather than standard deviations on the plots. (Most figures have standard deviations)

Some of the variables appear to NOT be normally distributed (see figure S4) and thus use of statistical tests designed for normally distributed data are inappropriate.

Another weakness of the approaches used are that changes in boundary layer height are not accounted for. This weakness cannot really be addressed without additional measurements, but it should be noted.

Other issues:

5. The abstract overstates the conclusions of the work. The conclusions have significant caveats, are based on limited number of sampled days, but the abstract makes it seem like the trends are well established, statistically significant, and based on a complete multi-year time series.

6. There are a number problems with Figure 6. It is not appropriate to grey out datasets that are not correlated. Data are data, and data points should not be deemphasized visually just because they do not fit a linear correlation. The datasets should be clearly labeled so that each symbol type can be connected back to its underlying study and land cover type. Having a linear correlation shown and then a change in the tick mark spacing is not a fair way of graphing in my opinion. The size ranges in question should be included in the axis labels and/or the caption. I believe this is the formation rate at

10 nm, and the NMINP at 10-25 nm? Is that consistent for all the datasets? If not, then I don't think this is a fair plot to put in. I don't think having regression equations and correlation coefficients on graphs is effective or appropriate (see additional comments on this later).

7. If a p-value appears in a figure or in the paper, then the statistical test needs to be discussed. What are the null and alternative hypothesis. And why is each hypothesis test implied by each p value important, scientifically interesting, novel, or useful?

8. If a regression equation (e.g., y=12.5x+5.6) appears in a figure or in the paper, then its use – either for scientific or engineering purposes – needs to be discussed. The paper has 9 regression equations in it. Are they of any use?

9. I believe all r values can be deleted from the paper without any loss.

10. Is the size range covered sufficient for calculating the condensational sink? Or stated differently, how much of the condensational sink is being missed by focusing on 10 to 150 or 250 nm.

11. Line 138 "can be calculated" or was calculated?

12. Are variables that are sensitive to the upper size limit (CCN concentrations that are based on the number of particles greater than size X, condensation sink) consistent given the change in the upper size limit shown in Figure S3, from campaign to campaign.

13. Line 282 – climate change typically requires 30-y averaging. Interannual variability may be much more likely at the time scales studied here.

14. Line 293 – "data size was small" is vague. A more detailed description of what aspects of the dataset are too small is needed.

15. Line 294 – there are two issues: spatial representativeness, and sparsity of the record in time. In my opinion these create two different problems for the work. "the

data size was small, and we should be cautious in extending the conclusion to a large spatiotemporal scale"

16. Line 299 – this shows the authors are thinking of these events as perfect Lagrangian experiments, where sampling at the mountain site is equivalent to sampling along a 0-D Lagrangian air mass trajectory. Vertical and horizontal mixing are not accounted for in this conceptual model. And the possibility that back trajectories evolve over the course of the PFGE is neglected. In reality, as the event evolves, winds will bring air with a variety of histories (chemical, emissions, radiation, accumulation mode particles, interaction with precipitation and clouds, etc.). The survival probabilities over 100% (Figure 4) are likely a symptom of the fact that reality has complex flows and spatial heterogeneity and does not fit the idealized box model concept.

17. Figure 7 is of low resolution. Difficult to see some of the symbols, and symbols are of different sizes in different plots.

18. The discussions of biogenic and total VOCs throughout the paper are problematic. What species are these? How were they measured? Were the measurements collocated with the PFGE measurements and matched in time? The amount of oxidation needed to grow from 3 to 10 nm or 10 to 20 nm, is quite small, so making broad generalizations about seginficant changes in entire classes of VOCs or in specific compounds, and then connecting them to PFGE is not scientifically valid.

19. Rather than making the data available "on request", the data should be publicly posted in machine readable formats at the time of publication in order to allow replication.

Birmili, W. and A. Wiedensohler (2000). "New particle formation in the continental boundary layer: Meteorological and gas phase parameter influence." Geophysical Research Letters 27(20): 3325-3328.

Dal Maso, M., M. Kulmala, I. Riipinen, R. Wagner, T. Hussein, P. P. Aalto and K. E. J.

Lehtinen (2005). "Formation and growth of fresh atmospheric aerosols: eight years of aerosol size distribution data from SMEAR II, Hyytiala, Finland." Boreal Environment Research 10(5): 323-336. Gaie-Levrel, F., S. Bau, L. Bregonzio-Rozier, R. Payet, S. Artous, S. Jacquinot, A. Guiot, F. X. Ouf, S.

Bourrous, A. Marpillat, C. Foulquier, G. Smith, V. Crenn and N. Feltin (2020). "An inter-comparison exercise of good laboratory practices for nano-aerosol size measurements by mobility spectrometers." Journal of Nanoparticle Research 22(5): 13.

Pfeifer, S., T. Müller, K. Weinhold, N. Zikova, S. Martins dos Santos, A. Marinoni, O. F. Bischof, C. Kykal, L. Ries, F. Meinhardt, P. Aalto, N. Mihalopoulos and A. Wiedensohler (2016). "Intercomparison of 15 aerodynamic particle size spectrometers (APS 3321): uncertainties in particle sizing and number size distribution." Atmospheric Measurement Techniques 9(4): 1545-1551.

Pitz, M., W. Birmili, O. Schmid, A. Peters, H. E. Wichmann and J. Cyrys (2008). "Quality control and quality assurance for particle size distribution measurements at an urban monitoring station in Augsburg, Germany." Journal of Environmental Monitoring 10(9): 1017-1024.

Wiedensohler, A., W. Birmili, A. Nowak, A. Sonntag, K. Weinhold, M. Merkel, B. Wehner, T. Tuch, S. Pfeifer, M. Fiebig, A. M. Fjaraa, E. Asmi, K. Sellegri, R. Depuy, H. Venzac, P. Villani, P. Laj, P. Aalto, J. A. Ogren, E. Swietlicki, P. Williams, P. Roldin, P. Quincey, C. Huglin, R. Fierz-Schmidhauser, M. Gysel, E. Weingartner, F. Riccobono, S. Santos, C. Gruning, K. Faloon, D. Beddows, R. M. Harrison, C. Monahan, S. G. Jennings, C. D. O'Dowd, A. Marinoni, H. G. Horn, L. Keck, J. Jiang, J. Scheckman, P. H. McMurry, Z. Deng, C. S. Zhao, M. Moerman, B. Henzing, G. de Leeuw, G. Loschau and S. Bastian (2012). "Mobility particle size spectrometers: harmonization of technical standards and data structure to facilitate high quality long-term observations of atmospheric particle number size distributions." Atmospheric Measurement Techniques 5(3): 657-685.

Wiedensohler, A., A. Wiesner, K. Weinhold, W. Birmili, M. Hermann, M. Merkel, T. Muller, S. Pfeifer, A. Schmidt, T. Tuch, F. Velarde, P. Quincey, S. Seeger and A. Nowak (2018). "Mobility particle size spectrometers: Calibration procedures and measurement uncertainties." Aerosol Science and Technology 52(2): 146-164.

---

## Referee Comment (RC2) · Anonymous Referee #4 · 28 Aug 2020

This papers investigates long-term behavior of new particle formation (NPF) and associated particle growth at an elevated site. This is an important and scientifically very interesting topic, since there are quite limited number of studies about the response of NPF to SO2 emission reductions, and since the obtained results are somewhat mixed between different environments. The fact that the study is based on relatively short-term measurement campaigns made in different seasons, rather than continuous measurements over full years, limits the reliability of the obtained results, and this should be properly acknowledged in the paper. Anyway, I would support publication of this paper, provided that the authors will address the issues raised below.

[Figure]

The introduction of this paper is generally well written. However, it would benefit from having a more concrete list of scientific questions aimed to addressed (in addition the aim mentioned on lines 75-76) in this paper. Two other, minor issues in this section: 1) the term "functions" on line 34 does not sound correct, perhaps "mechanisms"?, 2) the statement on line 74-75 is unclear. What altitudes are authors referring to here, above the boundary layer in general or upper free troposphere? One should be more careful with this, as elevated NPF can be associated with many different things, including convective uplift, presence of clouds, mixing of different air masses etc.

Experimental methods have been described very shortly, and should be expanded a bit. How were the measurement data used in the current paper quality checked, are these data undergoing any quality assurance procedures? Did detection limits etc. cause any issues for data interpretations? Were there any serious gaps in the data during the periods chosen for the current study?

Concerning the calculation methods, the authors should explicitly mention in main text (section 2.2.1) at which size particle formation rates were calculated, and what size range the calculated particle growth rates refer to (or if the applied size range for this calculation varied from event to event). Also, definition of "NPF duration" referred to e.g. on line 298 should explicitly described. Is it the time period over which new particles are observed to appear at the smallest sizes, or the time period over which the growth of new particle to larger sizes can be followed.

Categorizing NPF event based on the maximum size that the formed particle are able to reach by growth is in principle fine. However, doing that has one important issue that should be at least mentioned, and preferably shortly discussed, in the text. Following particle growth over several days, or event over the night, from observations is often difficult because of the typically large diurnal variation of boundary layer properties (e.g. mixed layer height), and because of changes in measured air masses. This can be seen, for example, on Dec 24 NPF event shown in Figure 1a: there are at least two major discontinuities in the particle number size distribution data (apparent in sudden

huge changes in particle number concentration in certain size ranges). As a result, it is highly questionable whether the particles observed to reach 217 nm actually initiated from the NPF event that took place much earlier on Dec 24. The same issues concerns the use of the term SP (survival probability). SP works fine when following the particle growth for a few hours, but becomes questionable for larger time periods. The authors should replace the term "survival probability" with something like "apparent survival probability" and discuss shortly this issue in the paper, including when interpreting the results.

The authors should be a bit more careful when using the term "trend". On line 191, for example, should there read "pattern" rather than "trend"? Multi-year trends can be season-dependent, but I suppose this not what the authors mean here. Please check out that "trend" is correctly used throughout the paper.

---

## Author Comment (AC1) · 18 Oct 2020

This paper investigates long-term behavior of new particle formation (NPF) and associated particle growth at an elevated site. This is an important and scientifically very interesting topic, since there are quite limited number of studies about the response of NPF to SO2 emission reductions, and since the obtained results are somewhat mixed between different environments. The fact that the study is based on relatively short-term measurement campaigns made in different seasons, rather than continuous measurements over full years, limits the reliability of the obtained results, and this

should be properly acknowledged in the paper. Anyway, I would support publication of this paper, provided that the authors will address the issues raised below.

Response: Thanks for the reviewer's comments. We agree that long-term continuous measurements would allow better investigating NPF trends under changing ambient conditions including air pollutants and meteorological factors. Due to practical difficulties, we tried short-term measurement campaigns made in different seasons of multiple years to characterize the NPF, with particular attention to the response of NPF to SO2 emission. The limitations of the non-continuous measures and the uncertainties to explain the results have been added in the revised discussion. We also try our best to respond the comments and revise our manuscript accordingly.

The introduction of this paper is generally well written. However, it would benefit from having a more concrete list of scientific questions aimed to addressed (in addition the aim mentioned on lines 75-76) in this paper. Two other, minor issues in this section: 1) the term "functions" on line 34 does not sound correct, perhaps "mechanisms"?, 2) the statement on line 74-75 is unclear. What altitudes are authors referring to here, above the boundary layer in general or upper free troposphere? One should be more careful with this, as elevated NPF can be associated with many different things, including convective uplift, presence of clouds, mixing of different air masses etc.

Response: Thanks. Lines 75-76 was revised as follow: "The main purposes of this study included: 1) to examine the effects of reduced SO2 emissions on the regional NPF events at high altitude (from the upper boundary layer to the lower free troposphere), i.e., changes in NPF frequency, intensity and the subsequent growth of new particles; 2) to quantify the potential contribution of new particles to the CCN population, and its changes under SO2 emission reduction; 3) to rationalize variation patterns of NPF characteristics and CCN parameters in terms of observational concentrations of gaseous precursors, their origins and atmospheric behaviors."

In the revision, we change ""functions" to "then raise up with different health and climate

effects". On line 74, we change "at higher elevations" to "above the boundary layer in general".

We agree that convective uplift, presence of clouds, mixing of different air masses, etc., may affect NPF events in clean remote atmospheres. We revise our discussion on the light of the issue.

Experimental methods have been described very shortly, and should be expanded a bit. How were the measurement data used in the current paper quality checked, are these data undergoing any quality assurance procedures? Did detection limits etc. cause any issues for data interpretations? Were there any serious gaps in the data during the periods chosen for the current study?

Response: Agree. The information will be added in the revised manuscript as following:

The WPS instrument was calibrated and/or repaired every 1-2 years by its vendor. The calibration parameter including the DMA sample/sheath flow, LPS sample/sheath flow, DMA/CPC pressure, DMA voltage, and DMA/ambient temperature. Polystyrene Latex (PSL) spheres (NIST) with the mean diameter of 100.7 nm and 269 nm were used for calibration. The detection limit of DMA was 10 nm when the DMA sample flow and sheath flow were 0.3L/min and 3 L/min, respectively. The detection limit of DMA could shift down to 5 nm when the DMA sheath flow increased to 4L/min (advanced mode). However, the pump consumption was faster. In this study, the detection limit of DMA was 10 nm in 2007 and 2009, while it shifted down to 5 nm in 2014, 2015, 2017, and 2018. To be consistent, only concentrations of particles >10 nm were used for the analysis. At the beginning of each campaign, the zero-points of the DMA, CPC, and LPS were checked using a purge filter at the inlet. The WPS sometimes operated improperly and the data had been excluded in the analysis.

SO2, O3, NO and NO2 were continuously measured at the Mt. Tai station since 2007. We performed multi-point calibrations every month and changed the filter every two weeks. The detection limits of SO2, O3, NO and NO2 were 1 ppb, 0.4 ppb, 0.04 ppb

and 0.4 ppb, respectively. PM2.5 was calibrated every three months by the method of mass foil calibration according to the instrument manual with the detection limit of 0.5 $\mu$g/m3. A multipoint calibration was performed for the online MARGA before and after the field campaigns in order to examine the sensitivity of the detectors. The detection limits were evaluated as 0.05, 0.05, 0.04 and 0.05 $\mu$g m$-3$ for Cl-, NO3-, SO42- and NH4+, respectively. More details about the instrument calibration can be found in Wen et al., (2018) and Li et al., (2020).

Concerning the calculation methods, the authors should explicitly mention in main text (section 2.2.1) at which size particle formation rates were calculated, and what size range the calculated particle growth rates refer to (or if the applied size range for this calculation varied from event to event). Also, definition of "NPF duration" referred to e.g. on line 298 should explicitly described. Is it the time period over which new particles are observed to appear at the smallest sizes, or the time period over which the growth of new particle to larger sizes can be followed.

Response: Thanks. In the revised manuscript, we added "The apparent formation rate of new particles is calculated based on the nucleation mode particles in sizes of 10-25 nm. The apparent growth rate is quantified by fitting the geometric median diameter of new particles (Dpg) during the whole particle growth period. The size range of Dpg varies from event to event. Details of the calculation equations can be found in the supplementary materials."

The definition of "NPF duration" has been added and reads as "The initial time of an NPF event was defined as the new nucleation mode particles started to be observed. The end time of an NPF event was normally determined by the new particle signal dropping to a negligible level and the total particle number concentrations approaching the background levels before the NPF event. In cases with the invasion of other plumes, the end time was determined by the new particle signals being suddenly overwhelmed by plumes and can't be identified since then. The NPF event duration was defined as the time duration between the initial time and end time of an NPF event. Noticed that

the detection limit of WPS was 5 nm or 10 nm, but the particles were nucleated at the critical cluster sizes around 1-1.5 nm. Thence, the NPF should occur for some times prior to our observation, and the actual duration should be longer than our calculation."

Categorizing NPF event based on the maximum size that the formed particle are able to reach by growth is in principle fine. However, doing that has one important issue that should be at least mentioned, and preferably shortly discussed, in the text. Following particle growth over several days, or event over the night, from observations is often difficult because of the typically large diurnal variation of boundary layer properties (e.g. mixed layer height), and because of changes in measured air masses. This can be seen, for example, on Dec 24 NPF event shown in Figure 1a: there are at least two major discontinuities in the particle number size distribution data (apparent in sudden huge changes in particle number concentration in certain size ranges). As a result, it is highly questionable whether the particles observed to reach 217 nm actually initiated from the NPF event that took place much earlier on Dec 24. The same issues concerns the use of the term SP (survival probability). SP works fine when following the particle growth for a few hours, but becomes questionable for larger time periods. The authors should replace the term "survival probability" with something like "apparent survival probability" and discuss shortly this issue in the paper, including when interpreting the results.

Response: In this study, Dpgmax and SP were calculated within the NPF duration. The definition of NPF duration has been clarified in the response above and will be added in the revision. In addition, a few spikes were excluded in calculating Dpgmax and SP since the spikes were characterized by a sudden change in particle number size distribution (PNSD) and may reflect the intrusion of primary or aged plume signals.

As reported in numerous literatures, NPF was a regional phenomenon occurring in a spatial extent varies from tens to thousands of kilometers (Kulmala et al., 2012, Nat. Protoc.) However, it is almost impossible to occur identic NPF events over the large spatial range because of different concentration levels of nucleation precursors. In

reverse, spatial heterogeneity of regional NPF events is a common phenomenon and would cause the discontinued PNSD to some extent at a fixed observational site.

No criteria have been well-established in the literature to identify spatial heterogeneity of NPF events. In this study, spatial heterogeneity of NPF events was assumed for the discontinued PNSD if no intrusion of primary or aged plume signals were clearly identified. Even though the intrusion of primary or aged plume signals overwhelmed new particle signals for a short period, new particle signals can still be reasonably observed afterwards in the contour plotting. The discontinued PNSD was also assumed as the continuity of the NPF event. The NPF event on Dec. 24 was illustrated as example:

On the Dec. 24 event, (Figure R1), the two discontinuities appeared at 7:00-9:00 and 16:10-18:50 on Dec. 25. From 22:48 on Dec. 24 to 7:00 on Dec. 25, N10-300nm continuous decreased from $1.1\times10^4$ cm-3 to $0.8\times10^4$ cm-3, meanwhile, Dpg grew from 30 nm to 63 nm with the growth rate of 4.0 nm/h. Between 7:00 and 9:00, N10-300nm and Dpg oscillated at $0.8\pm0.1\times10^4$ cm-3 and $51\pm6$ nm. At 9:00, N10-300nm went back to $0.8\times10^4$ cm-3 and Dpg was 72 nm. In these two hours, new particles were hypothesized to experience a growth similar to the previous curve. Similarly, Dpg was 115 nm at 16:10 and 128 nm at 18:50. The assumed growth rate during the 2.7 hours was about 4.8 nm/h, close to the previous GR. Actually, when we fitted the entire Dpg between 22:48 on Dec. 24 and 18:50 on Dec. 25, the GR was 4.9 nm/h and $R^2$=0.97, suggesting that particles grew in a smooth curve.

In this case, Dpg reached the maximum of 163 nm at 9:00 on Dec. 26, then the new particle signal was overwhelmed by pollutant plumes. $\Delta$NCCN reached the maximum at 20:23 on 25 Dec. The SP was calculated as SP = $\Delta$NCCN/NMINP, and SPCCN50, SPCCN80, and SPCCN100 was calculated to be 0.2, 0.2 and 0.15, respectively.

We agree to change "survival probability" to "apparent survival probability". The judgment of spatial heterogeneity in other NPF events followed the similar approach above, and we will add the discussion of spatial heterogeneity in the revised manuscript.

The authors should be a bit more careful when using the term "trend". On line 191, for example, should there read "pattern" rather than "trend"? Multi-year trends can be season-dependent, but I suppose this not what the authors mean here. Please check out that "trend" is correctly used throughout the paper.

Response: Thanks. We agree the reviewer's comment that "trend" was inappropriate in this paragraph. Our main purpose was to examine the effects of reduced SO2 emissions on the regional NPF events characteristics and the CCN parameters. We will change "trend" to "pattern" in this sentence and go through the full text and revise the ambiguous statements.

[Figure]

**Fig. 1.** Figure R1 Contour plot of NPF events (a) and the time series of particle number concentration in 10-25 nm (N10-25nm) and 10-300 nm (N10-300nm) (b) during 24-27 Dec. 2017.

[Figure]

---

## Author Comment (AC2) · 18 Oct 2020

Response to Anonymous Referee #3

The manuscript analyzes seven field campaigns where particle number size distributions (PNSD) and sulfur dioxide were measured at the summit of a mountain site in the North China plain. Supporting measurements of time-resolved PM2.5, O3, and oxides of nitrogen were taken. And each campaign included 1-h time resolution ions in PM2.5 using water extractive methods (URG-AIM or MARGA). The most recent campaign was in 2018. Across the 7 campaigns, a little over 100 particle formation and growth events were detected, with the analysis focused on the size range of 10-300 nm size

range. From the earliest to most recent campaign, SO2 emissions and concentrations have dropped dramatically, and the paper tries to analyze whether the particle formation and growth activity has changed in ways that are expected from the sulfur dioxide decrease. A large number of metrics are computed and then analyzed for each particle formation and growth event (PFGE). The metrics include, but are not limited to, the apparent formation rate of 10-25 nm particles (FR), the growth rate, the absolute increase in N10-25 particle concentration from the start to the peak of the PFGE (this is the NMINP variable), the PFGE duration, the PFGE frequency, the size to which the growth event reaches (Dpgmax variable in the manuscript), and particle counts which are used as surrogates for the change in CCN concentrations at low, medium, and high supersaturations (N100-300, N80-300, and N50-300). The paper includes values for and discussion of total VOC during the campaigns.

Complicating the analysis is that the field campaigns were in different months of the year: April 2007 ($\sim$30 d), June 2009 ($\sim$20 d), Aug 2014 ($\sim$30 d), Oct/Nov 2014 ($\sim$70 d), Jul 2014 ($\sim$40 d), Dec 2017 ($\sim$35 d), and Mar 2018 ($\sim$30 d).

The paper's abstract makes five claims: a. The formation rate in 2018 is 2-3 times higher than the formation rate in 2007. b. Net maximum increase in nucleation mode number concentration is 2-3 times higher in 2018 than in 2007. c. The occurrence of events where the mode of the growth event goes above 50 nm is lower in 2018 than it was in 2007. d. A surrogate for CCN production at high supersaturation (N50-300 at its peak during each growth event minus N50-300 before the event) decreased from 3703 per cm3 (before 2015) to 1026 (2017-2018). e. The authors argue availability of organic precursors has increased in the most recent campaigns, allowing more particle production and initial growth; furthermore, they argue that the lack of later growth is from reduction of "anthropogenic precursors" (presumably SO2).

The paper requires substantial revision before it is suitable for publication. The key issue, to this reviewer, is that making accurate claims about year-on-year trends and variability in PFGE is difficult. The requirements to make the claims defensible are:

(1) take a sufficient number of samples to reduce random variability and give sufficient statistical power; (2) take steps to minimize, test for, and quantify campaign-specific systematic instrument bias (also known as "instrument drift"); (3) take steps to enforce consistency in any subjective data interpretation steps, such as classification of PFGE into "types" and the determination of the start and end times of events; (4) use statistical methods designed for trend analysis, time series analysis, and combined analysis of seasonal and interannual variability.

Each requirement needs to be met in order for the claims about trends to be defensible. And for peer review and reproducibility purposes, things need to be documented for the peer-review and scientific communities.

I think the current work fails to meet all four of the requirements. While some of the conclusions are likely accurate (in that they would not change if all the requirements were met) – others would change, or require extensive qualification.

Response: Thanks for the review's constructive comments. We agree that some analyses and related conclusions in our original version should be more conservative. In addition, more clarifications are needed to better defense them. Scientific community normally prefers to see the extracted trend in ambient variables using the measurement data over 20 years. Thus, the technical term "trend" used in the original version is problematic and should be removed.

Long-term continuous measurements may allow better investigating NPF trends, however, all statistical tools in literature suffer from the weakness to some extent in extracting the de-weathered trend in interested variables associated with anthropogenic perturbation, based on our previous studies. For non-continuous measurements, it is still a common challenge to address specific scientific questions using the proper statistical analysis. We agree that the weakness and challenge should be included in the revision.

A comparative analysis was conducted to study particle formation and growth events

(PFGE) in different years based on two observational facts, 1) their occurrence frequencies in 2007 and 2009 were reasonably same as those in 2017 and 2018 even in different seasons; 2) a large decrease in SO2 mixing ratio 2017 and 2018 against in 2007 and 2009. We then focused on comparative analyzing the spring PFGEs in 2007 and 2018, where uncertainties from varying ambient factors may have been largely minimized. This will be clarified in the revision. Moreover, an on-site automatic weather station has been operated continuously since 2005. Meteorological parameters such as temperature, relative humidity, wind speed, wind directions and pressure will be analyzed statistically to facilitate the discussion of PFGEs during the seven campaigns.

We also found a large variation in occurrence frequency of summer PFGEs in different years. For the summer PFGEs, it may require extremely abundant chemical information to study the effect of decreasing SO2 levels on PFGEs because of a huge perturbation from meteorological conditions and related biogenic emissions of air pollutants. On the other hand, the observations of summer PFGEs also implied that the same occurrence frequencies of PFGEs may be a critical indicator to constrain the comparative analysis.

We highly appreciate the comments on technical issues of measurements. The commercial instruments were routinely calibrated to ensure the QA/QC using the service provided by their vendors. We agree that more cautions should be paid to instrument limitations, especially in a small particle size range. In the following text, we will try our best to address the comments point by point and revise the manuscript accordingly, in order to improve our analysis more defensible and robust.

1. Statistical power:

PFGE exhibit substantial seasonal variation, due to changes in temperature, relative humidity, biogenic activity, atmospheric chemistry, soil moisture, preexisting aerosol concentration and chemistry, radiation, cloudiness, boundary layer structure, land cover/vegetation canopy structure, synoptic meteorology, anthropogenic emissions, and atmospheric ion levels. Local meteorological features (i.e. orographic meteorology) and local sources may also have month-to-month variability. And at the 20-30 d time scale, large scale persistent geophysical features can cause a whole campaign of measurements to be atypically high or low for a number of PFGE variables. To accommodate all these sources of variability, large sample sizes are required for analysis of seasonal variation and interannual trends. In the absence of large sample sizes, careful pairing of events and analysis of alternate sources of variability / alternate hypotheses are needed isolate cause-effect relationships on specific PFGE variables.

With each campaign at a slightly different time of the year, some campaigns as short as 20 d, and no discussion of whether air pollution levels, air pollution meteorology, and climate variables were at climatologically representative levels, the reader has to apply great skepticism to any claims of interannual trends and cause-effect relationships for those interannual trends. See for example (Birmili and Wiedensohler 2000) who do take into account air mass characteristics.

Response: To our best understanding from recent review papers on PFGE, eight variables, i.e., concentrations and cumulative generation amounts of sulfuric acid together with Highly Oxygenated Organic Molecules (HOM) and other secondary organics in different volatilities, the product of gaseous $HNO_3$ and gaseous $NH_3$ minus the equilibrium constant of $NH_4NO_3$ and then minus the kelvin effect term, cumulative generation amounts of condensed $NH_4NO_3$ on size-dependent particles, would directly affect apparent new particle formation rate (FR), apparent net maximum increase in the nucleation-mode particle number concentration (NMINP), new particle growth rate (GR) and the maximum geometric median diameter of grown new particles (Dpgmax). Although gaseous amines have been proposed to participate in ambient nucleation, their concentrations in China, based on the authors' work, are too high (relative to sulfuric acid) to act as the limitation factor. How aminium salts contribute the growth of newly formed particles larger than 10 nm, which is the one of focus in this study, is poorly understood in China. What the reviewer claim above may indirectly affect the eight variables to some extent. Only the information of the eight variables is not sufficient to support our analysis, those indirect factors should be cautiously utilized to facilitate the analysis. The argument will be added in the revision.

We agree that the technical term "trend" is misleading and will be removed in the revision. We will also revise our discussion on the cause-effect relationships of PFGE by considering significant changes in the eight variables. If the required data are not available, we tried to use indirect factors facilitate analysis.

In polluted and NH3-rich (we may reasonably assume that amines may also be rich relative to sulfuric acid) ambient air in China, the influence of air mass characteristics on PFGE may be totally different from that in the clean and pristine atmospheres. In clean and pristine atmospheres, air mass characteristics may provide important information on precursors' sources of PFGE. In China, precursors of PFGE are abundant in general. For example, the occurrence frequency of PFGE reached ∼50% in winter campaign as presented in this study, while the value was less than 5-10% in winter in Europe. Alternatively, air mass characteristics greatly affect moisture characteristics and may subsequently affect the cloudiness (radiation) and H2SO4 concentration through modulating OH free radical concentration. Of course, the latter is poorly studied so far.

The size distributions shown in Figure S3 are suggestive of insufficient number of days sampled in the dataset. Telling whether the system shifted from unimodal to bimodal behavior between 2015 and 2017 (all unimodal for 2015 and prior) vs. this occurring through some instrument drift vs. this occurring through sampling non-climatological conditions due to small samples sizes is difficult.

Given the decrease (Table 2) in PM2.5, sulfate, and SO2 between spring 2007 and 2018 (PM2.5 60 vs. 30 ug/m3; sulfate 17 vs. 4 ug/m3, SO2 18 vs. 3 ppb), more discussion is needed of the large increase in condensation sink in 2018 (Figure S3) and in the large increase in the height of the size distribution function at 100 and 150 nm between 2007 and 2015.

The discontinuity in the slope of the size distribution function at 200 nm also indicates there may be some drift in the size-specific performance of the WPS (Figure S3). The discontinuity in slope is not really evident until 2017, but then appears in 2017 and 2018.

Response: We agree that the analysis of PFGE in the origin version misses out the important point. On the light of the comments, we re-checked the data. We do find that three channels in WPS around 213 nm suffered from unexpected errors in reporting number concentrations in approximately 30% sampling days in 2017 and 2018. It has a minor influence on PFGE in 7 NPF days out of the total of 32 NPF days in 2017 and 2018 as shown in Fig. S3. The data in the three channels suffering from abnormal errors will be corrected by assuming a linear decrease of particle number concentration from 150 nm size bin to 300 nm size bin. Thank again.

We plotted particle number size distributions on non-NPF days in different years (Figure R1). Bimodal particle number size distributions can be observed in 2009, 2014 and 2015. We don't see any shift when the size distributions in 2014, 2015 and 2018 were compared with each other. We also find that the median geometric diameters of accumulation mode on NPF days in 2017 were consistent with those on non-NPF days in 2014, 2015 and 2018. However, the median geometric diameters of accumulation mode on non-NPF days in 2017 did shift to the large size and the same was true for those on non-NPF days in 2009. Moreover, the median geometric diameters of accumulation mode on NPF days in 2009 were consistent with those on non-NPF days in 2014, 2015 and 2018. Overall, our regular instrument maintenance appears to be effective to prevent from the instrument shift, except the occasional problem at three size bins around 213 nm.

In the origin manuscript, we calculated condensation sink based on the different size range of particle in 2007 (10-153 nm) and 2018 (10-300 nm). To be consistent, we recalculated CS based on 10-153 nm particles on 2018, and the average of $0.4\pm0.15$ s-1 was slightly larger than the average CS in 2007 ($0.32\pm0.19$). CS should be unrelated

to secondary particles via ambient nucleation since the particles cannot grow over 60 nm. Primary emissions of the accumulation mode particles were out of scape of this study, although they may scavenge precursors of PFGE to some extent.

2. Minimize, test for, and quantify campaign-specific instrument drift:

Achieving consistency in PNSD in long-term measurements is difficult. And it is not sufficient to state that each individual campaign had sufficient quality assurance, referring the reader to the campaign specific papers. There needs to be a presentation of data and discussion of how comparable the instrument responses are from campaign to campaign. What steps were taken to make sure instruments were not drifting. Aging of components can cause variation in flows, sizing accuracy, counting accuracy, particle losses, CPC supersaturations, and in the effective lower size limit of the instrumentation of the particle number spectrometer system. The detection efficiency as a function of size at the lower range of the instrument (5-25 nm), at the upper range of the mobility analyzer, at the lower end of the optical particle devices, and at the upper end of the optical particle analyzer – these are all difficult to maintain at stable levels over long periods of time. The total particle counts, height of the size distribution function, sensitivity at the lower and upper ranges of size distributions – these vary from year to year and require careful intercomparison, quality assurance, and maintenance procedures to deal with. See for example the results of intercomparison studies (Pfeifer, Müller et al. 2016) and papers focusing on quality assurance, calibration, and harmonization (Pitz, Birmili et al. 2008, Wiedensohler, Birmili et al. 2012, Wiedensohler, Wiesner et al. 2018, Gaie-Levrel, Bau et al. 2020). Comparison to other instruments for total particle counts, size distribution functions in overlapping regions, checks with monodisperse particles are some of the techniques that can be used to establish more confidence and quantify campaign-to-campaign comparability.

Consistency in inlet dimensions, inversion algorithms (including multiple charge correction), use of impactors to manage multiple charge issues, corrections for inlet transmission efficiency, – these can all be issues in campaign-to-campaign comparability.

They need to be discussed.

While being able to reproduce time-resolved PM2.5 measurements from the WPS size distribution is not sufficient to show accuracy in the nucleation and Aitken ranges – it is probably necessary. At least showing consistency from campaign to campaign in the volume of particles measured by the WPS and the mass of particles by time resolved mass measurements can help to demonstrate stability in instrumentation and data processing algorithms.

The fact that the authors are using an instrument with nominal lower cutoff of 5 nm, but discarding data between 5-10 nm indicates that there may be a problem with sensitivity at the lower size limit, or (more likely) variability in the sensitivity at the lower size limit. There is further evidence in Figures 1 and S6 – of a problem. In all the bursts shown save one, the particle size distribution function slopes down from a peak at about 13 nm to a lower value at 10 nm. If the instrument is biased low in the 10-13 nm range, then the statistics developed in the work will also be biased. If that bias varies from campaign to campaign, then that creates additional interpretation difficulties.

At line 180, it is implied that at times the WPS was collocated with instruments with lower limit of 3 nm. Therefore, the actual performance in the 5-15 nm range could (and should) be determined though comparison to such collocated instruments.

Response: Honestly, we rely on the instrument vendor on instrument maintenance and calibration every 1-2 years. Except the problem at size bins around 213 nm sometimes occurring in 2017 and 2018, the measured size distributions were reasonably consistent as mentioned above. We excluded the concentrations of particles below 10 nm for analysis in order to keep the lower limit of PNSD consistent in seven campaigns. More details are presented below.

In revision, we will add the information as following: "The WPS instrument was calibrated and/or repaired every 1-2 years by its vendor. The calibration parameter including the DMA sample/sheath flow, LPS sample/sheath flow, DMA/CPC pressure, DMA

voltage, and DMA/ambient temperature. Polystyrene Latex (PSL) spheres (NIST) with the mean diameter of 100.7 nm and 269 nm were used for calibration. The detection limit of DMA was 10 nm when the DMA sample flow and sheath flow were 0.3L/min and 3 L/min, respectively. The detection limit of DMA could shift down to 5 nm when the DMA sheath flow increased to 4L/min (advanced mode). However, the pump consumption was faster. In this study, the detection limit of DMA was 10 nm in 2007 and 2009, while it shifted down to 5 nm in 2014, 2015, 2017, and 2018. To be consistent, only concentrations of particles >10 nm were used for the analysis. At the beginning of each campaign, the zero-points of the DMA, CPC, and LPS were checked using a purge filter at the inlet. The WPS sometimes operated improperly and the data had been excluded in the analysis."

In addition, the aging of components may lower the detection efficiency of WPS. However, the increased FR and NMINP in 2017-2018 reveal that the signals of nucleation mode particles enhanced in recent years. As presented in original manuscriptïijŇit reads as"During the four campaigns in 2007, 2009, and 2014, the calculated FR varied narrowly in each campaign and the campaign average narrowed to 0.8–1.2 cm−3 s−1. The FR increased thereafter, i.e., 2.6 ± 1.3 cm−3 s−1 in 2015, 2.0 ± 1.7 cm−3 s−1 in 2017, and 3.0 ± 2.7 cm−3 s−1 in 2018."Therefore, we convince that the instrument maintenance can effectively reduce the aging impact of instrument components on observational data.

Conductive tubes (TSI 1/4 in.) were used for the WPS sampling in each campaign. The length of the tube was kept at about 2 m in each campaign (fixed position of WPS in the container). We used the SWS mode (DMA operating in the voltage-scanning mode) for measuring. The charge correction was calculated by the Boltzmann charge distribution, and the equation has been considered in the instrument algorithm.

PM2.5 were measured in 2007, 2014, 2017 and 2018. Assuming the particle density is 1.5 g cm-3, the mobility diameter can convert to aerodynamic diameter following the equation of Aerodynamic diameter=Mobility diameter$\times\sqrt{1.5}$. The particle mass

concentration in each size bin can be calculated according to the particle number concentration reported by WPS. Then we integrated the mass concentration of less than 2.3 $\mu$m in aerodynamic diameter (1.9 $\mu$m in mobility diameter) and compared with the PM2.5 mass concentration reported by TEOM 1400a (2007) or Thermo 5030 SHARP (2014-2018). The relationship of hourly average data is showed in figure R2.

In 2007, we calculated the mass concentration of PM0.18 from WPS and found it has a weak correlation with PM2.5. A slope of 0.05 indicated that the particles we observe account for a minor fraction of the total mass. In the two campaigns in 2014, the mass concentration of WPS-derived PM2.3 and SHARP measured PM2.5 showed good correlations, with slopes of 0.69-0.76. In 2017 and 2018, we removed the abnormal data in three bins around 213 nm, and found a good linear correlation between the two methods, but the slopes slightly increased to 0.86-0.9. It should be noted that our calculation method depends on the density of particles. If the actual particle density deviates from the assumed value, the integrated volume (mass) is misadjusted. The difference in slopes may be due to the difference in particle density, or other unknow factors. Nevertheless, all of the deviations were less than 30% and within a reasonable range, and the linear correlations are generally good. Thus, our result showed the WPS was generally stable during the four campaigns. These will be added in the revised supplementary.

As we mentioned above, the detection limit of DMA was 10 nm in 2007 and 2009, while it shifted down to 5 nm in 2014, 2015, 2017, and 2018. To be consistent, only concentrations of particles >10 nm were used for the analysis. On April 7 (figure 1b), the initial peak at about 13 nm. That because the initial nucleation was influence by sporadic spikes, which overwhelmed the nucleation signal. In revision, we will remove the fitted Dpg when the PNSD was influenced by spikes.

The measurements made by NAIS at Mt. Tai were reported by Lv et al., 2018. But we have no confidence on the raw NAIS data. Fortunately, we have conducted a comprehensive comparison between the WPS and a SMPS (GRIMM, Germany) at a

costal site in Qingdao, China. The SMPS consists of a DMA (55-UïijŇGRIMM) and a CPC (5416ïijŇGRIMM). It covers the particle size range of 10 nm-1000 nm, and is set up to 127 channels. The time resolution of SMPS is 4 min. The two instruments of WPS and SMPS were operated side by side during 2-7 July 2020 for intercomparison.

Figure R3 shows the comparison of particle number concentration in the range of 10-25 nm (nucleation mode) and 10-300 nm (particle size range we used for calculation in this paper) between WPS and SMPS. Particle number concentration in these two size ranges showed good linear correlations, suggesting the measurements of the two instruments are highly consistent. Furthermore, the highly correlated data indicates that the WPS is not experiencing aging problems.

3. Consistency in subjective data interpretation/classification steps:

It is not clear which of the variables used for analysis involve human classification. Sometimes, human classification is used for PFGE types (often using how smooth the growth event is in time); human classification is used sometimes for establishing times (start of event, end of the event). The end time is described. From line 113 of manuscript, "The end time of an NPF event was defined as the time when the particle number concentrations approached the background levels observed before the NPF event. The NPF event duration was defined as the time duration between the start time and end time of an NPF event." This seems like the end of event was a subjective determination of when background was approached. Thus the end time, duration, and any rate that has the duration in the denominator may be subjective.

For subjective (human) event classifications, were the events uniformly reclassified for this paper, or were prior classifications adopted from 2007 and 2009 and mixed with new classifications done for the more recent campaigns. See (Dal Maso, Kulmala et al. 2005) for best practices on human classification.

Response: In revision, we will add the details when classifying NPF events: "In this study, particles with diameter smaller than 25 nm were defined as nucleation mode

particles (Kulmala et al., 2012). Followed the criteria proposed by Dal Maso et al. (2005) and Kulmala et al. (2012), three features had to be met to classify an NPF event: 1) a distinctly new nucleation mode particles must appear in the size distribution; 2) the new mode should prevail over a time span of hours; 3) the new mode should show signs of growth. All three features are required for a day (00:00-23:59 LT) to be classified as an NPF day. Otherwise, the day is classified as a non-NPF day.

The initial time of an NPF event was defined as the new nucleation mode particles started to be observed. The end time of an NPF event was normally determined by the new particle signal dropping to a negligible level and the total particle number concentrations approaching the background levels before the NPF event. In cases with the invasion of other plumes, the end time was determined by the new particle signals being suddenly overwhelmed by plumes and can't be identified since then. The NPF event duration was defined as the time duration between the initial time and end time of an NPF event. Noticed that the detection limit of WPS was 5 nm or 10 nm, but the particles were nucleated at the critical cluster sizes around 1-1.5 nm. Thence, the NPF should occur for some times prior to our observation, and the actual duration should be longer than our calculation."

Followed the definition above, we classified the NPF event uniformly during the seven campaigns, i.e., from 2007 to 2018.

4. Statistical methods appropriate to analysis of combined seasonal and interannual variability

Statistical procedures for evaluating trends in seasonally varying time series need to be followed in order to state claims that trends exist. These can be found in a number of textbooks, papers, and government reports. See for example Statistical Methods for Environmental Pollution Monitoring by Gilbert https://www.osti.gov/servlets/purl/7037501/. And (Asmi, Coen et al. 2013, Collaud Coen, Andrews et al. 2013, Squizzato, Masiol et al. 2019). Many other good models for seasonally adjusted trend detection can be found in the O3, NOx, PM2.5, and hydrology/climatology literature. Squizzato et al. (2019) for example have the statistical procedures necessary to detect turning points (see line 236 where manuscript discusses turning points)

See for example line 290 "the CS still increased in 2018 compared with that in 2007." That implies annual average condensational sink increased, and this is a season or month specific result – and it is not clear there is enough statistical confidence to state this. Many other locations in the paper have broad statements about PFGE behavior in one year vs. another, or imply a long term trend where it has not really been shown.

Interpretation of PFGE data from this site seems more complicated than most, because of two issues: (1) it is sometimes influenced by boundary layer and other times by free troposphere; (2) very long PFGE events (see for example Figure 1a, where a 3-d long event is shown) are being compared with shorter (midday + afternoon) growth events. See Figure 7 which has events ranging from 3-h duration to 85-h duration. The flow patterns and chemistry required to sustain a 3-h event and an 80+ h event are likely very different, and would require more thoughtful comparison metrics than used in the paper.

The paper acknowledges this difficulty in interpretation (line 295) but more needs to be done than just acknowledge the difficulty. See analysis papers from PFGE studies at other high altitude sites. They do attempt to determine the degree of FT influence and the impact of polluted boundary layer air. And there are many papers that factor in air mass characteristics and/or back trajectory in analysis of PFGE.

See for example Figure 1a where on 25-Dec 2017 there were simultaneously occurring a short PFGE (category 1) and evolution of the category 3 event that started on 24-Dec 2017. This raises a number of questions on how such a dataset can be analyzed to determine trends.

How much of the variability in data is that some campaigns had more free tropospheric

influence and others have less. How much of the conclusions of the paper are driven by switches (during PFGE) in air mass influence to/from FT influence. In other words, PFGE events that have their evolution dynamics controlled by airflows, and not by chemistry – hence the authors observed lack of influence or counterintuitive effects of SO2.

As for statistical procedures, I think it would be much more appropriate to put 95% confidence intervals on means rather than standard deviations on the plots. (Most figures have standard deviations)

Some of the variables appear to NOT be normally distributed (see figure S4) and thus use of statistical tests designed for normally distributed data are inappropriate.

Another weakness of the approaches used are that changes in boundary layer height are not accounted for. This weakness cannot really be addressed without additional measurements, but it should be noted.

Response: We acknowledge that the technical term "trend" is misleading in the origin manuscript. It will be removed in the revision. The "turn point" was inappropriate and we will remove this in revision.

Line 290 will be changed to "Note that the campaign average of PM2.5 mass concentration in 2018 indeed decreased. The decrease was apparently determined by the decrease in >153 nm particles, since no significant difference existed in the CS (calculated based on <153 nm particles) between in 2007 and 2018." We didn't imply the annual trend of CS or other variables, we will go through the full text and revise the ambiguous statements.

Here we comprehensively analyze four cases of NPF events in different categories (category 1 events on 5 April 2007 and 6 April 2018, category 2 event on 8 April 2018), and category 3 event on 23 April 2007) in 2007 and 2018. The meteorological parameters, gases pollutants, PM2.5 mass

concentrations and planetary boundary layer height (PBLH, download from https://goldsmr4.gesdisc.eosdis.nasa.gov/data/MERRA2/M2T1NXFLX.5.12.4/) were showed in figure R4 and R5. PBLH shows obvious diurnal variations, and the maximum value are 4110 m, 3000 m, 2316 m, and 2224 m on 6 April 2018, 7 April 2018, 5 April 2007, and 23 April 2007. There was no significant difference in the evolution of PBLH among the three categories. We argued that PFGE events were controlled by chemistry, since the changes of airflow always associate with the changes of air pollutant, which directly influence NPF as mentioned in the response to the first comment.

As reported in numerous literatures, the growth of newly formed particles is mainly attributed to sulfuric acid, ammonium nitrate, and secondary organic compounds (Wiedensohler et al., 2009; Riipinen et al., 2011; Zhang et al., 2012; Ehn et al., 2014; Man et al., 2015; Wang et al., 2015; Burkart et al., 2017; Lee et al., 2019; Wang et al., 2020). We therefore explore their respective contributions as follows. First, we calculated the contribution of sulfuric acid to the growth based on the observed mixing ratio of SO2 and equations 2-3 in section 2.2.2. Second, we examined whether NH4NO3 freshly formed in PM2.5 during the particle growth period. In case of no NH4NO3 formation, its contribution would not be expected. This is because an even higher product of HNO3gas*NH3gas is required to overcome the kelvin effect and form NH4NO3 in nucleation mode and Aitken mode particles. Thus, the growth unexplained by sulfuric acid should be mainly contributed by organics. In case of NH4NO3 formation, we considered the net increase in NH4NO3 may contribute to the particle growth, even though the ratios of increased NH4NO3 in PM2.5 may not be the same as the ratios in nucleation mode and Aitken mode particles.

On 6 April 2018 (category 1), the NPF event was first observed at 09:10. Dpg was fitted as 13 nm at 09:45, and continuous grow to 30 nm at 18:10. Then both of the particle number concentration and particle diameter decreased, and the plume overwhelm the new particle signal at 6:00 on 7 April. During the NPF period, sulfuric acid was estimated to contribute about 16% to particle growth. The mass concentration of nitrate in PM2.5 was less than 1.0 $\mu$g m-3, implying that fresh NH4NO3 formation did not occur. Thus, the growth unexplained should be mainly contributed by organic matter.

On 7 April 2018 (category 2), Dpg increased from 13 nm at 10:00 to 43 nm at 18:00, then Dpg fluctuate at 41 nm-52 nm in the following 10 hours. Sulfuric acid was estimated to contribute about 11% to particle growth. The mass concentration of nitrate in PM2.5 continuously increased from 0.8 $\mu$g m-3 at 10:00 to 2.7 $\mu$g m-3 at 20:00, then decreased to 2 $\mu$g m-3 at 4:00 on 8 April 2018. Formation of ammonium nitrate seems to contribute to the growth of new particles in this case.

Similarly, on 23 April 2007 (category 3), sulfuric acid was estimated to contribute about 23% to particle growth. The mass concentration of nitrate in PM2.5 increased from 1 $\mu$g m-3 to 10 $\mu$g m-3 during the particle growth period, indicating its important role in the particle growth. On the contrary, the mass concentration of nitrate and sulfate decreased during the NPF period on 5 April 2007 (category 1), and new particles didn't grow to the larger size.

We summarized the mass concentration of SO42−, NO3−, NH4+ and OC during the formation and growth period of NPF events in 2007 and 2018 campaigns (added in Table 2). During the growth periods, the contribution of H2SO4 vapor to particle growth decreased from 36% in 2007 to 11% in 2018. The mass concentration of nitrate in PM2.5 also decreased from 7.4 ± 4.8 $\mu$g m-3 in 2007 to 6.7± 5.5 $\mu$g m-3 in 2018. In addition, OC in PM2.5 was lower in 2018 (5.5 ± 2.0 $\mu$g m−3) than in 2007 (6.1 ± 3.0 $\mu$g m−3). In 2018, the reduced H2SO4 vapor, nitrate and OC formation may lead to the decrease in the growth probability of new particles. However, large uncertainties still exist because of a lack of data on the chemical composition of these smaller particles.

Figures 3 and 5 will be changed to the box chart with 95% confidence intervals on means.

Two sets of data in figure S4 are not linear correlated, and we will remove the four fitting

equations.

Other issues:

5. The abstract overstates the conclusions of the work. The conclusions have significant caveats, are based on limited number of sampled days, but the abstract makes it seem like the trends are well established, statistically significant, and based on a complete multi-year time series.

Response: Thanks. We will remove the "trend" in revision and rewrite abstract.

6. There are a number problems with Figure 6. It is not appropriate to grey out datasets that are not correlated. Data are data, and data points should not be deemphasized visually just because they do not fit a linear correlation. The datasets should be clearly labeled so that each symbol type can be connected back to its underlying study and land cover type. Having a linear correlation shown and then a change in the tick mark spacing is not a fair way of graphing in my opinion. The size ranges in question should be included in the axis labels and/or the caption. I believe this is the formation rate at 10 nm, and the NMINP at 10-25 nm? Is that consistent for all the datasets? If not, then I don't think this is a fair plot to put in. I don't think having regression equations and correlation coefficients on graphs is effective or appropriate (see additional comments on this later).

Response: Thanks. We have changed the grey markers to black, as shown in figure R6. In this study at Mt. Tai, all of the FR and NMINP are linearly correlated, and FRs were less than 15 cm-3s-1 (blue markers in figure R6). Therefore, the linear relationship was the key point we would like to address, and we change the tick mark space when FR larger than 20 cm-3s-1 in order to emphasize our data at Mt. Tai.

We confirmed that the FR and NMINP were calculated based on 10-25 nm particles in all cases in this figure.

7. If a p-value appears in a figure or in the paper, then the statistical test needs to be

discussed. What are the null and alternative hypothesis. And why is each hypothesis test implied by each p value important, scientifically interesting, novel, or useful?

Response: The significance of P value will be added in revision.

8. If a regression equation (e.g., y=12.5x+5.6) appears in a figure or in the paper, then its use – either for scientific or engineering purposes – needs to be discussed. The paper has 9 regression equations in it. Are they of any use?

Response: Yes, the equation has its implication. For example, in type A, the Dpgmax and GR can be fitted by the equation: y=12.5x+5.6, with moderate good Pearson correlation coefficient. Based on the obtained equation, newly formed particles could grow beyond 50 nm only when the GR exceeded 3.55 nm h-1 in this type of NPF events. These will be added in revision.

9. I believe all r values can be deleted from the paper without any loss.

Response: Correlation coefficient is a statistical concept, which helps in establishing a relation between predicted and actual values obtained in a statistical experiment. The calculated value of the correlation coefficient explains the exactness between the predicted and actual values. The r values are used to measure the degree of correlation between two variables. A high r value (close to 1) means that the variables are highly correlated, and the fitted equation has its physical meaning. A small r value (close to 0) means the two variables are irrelevant, and the fitted equation is meaningless.

10. Is the size range covered sufficient for calculating the condensational sink? Or stated differently, how much of the condensational sink is being missed by focusing on 10 to 150 or 250 nm.

Response: We recalculated CS based on 10 nm-2.5 $\mu$m, 10 nm-300 nm and 10 nm-150 nm particles in 2018. The CS was $0.80\pm0.37\times10$-2 s-1, $0.75\pm0.34\times10$-2 s-1, $0.40\pm0.15\times10$-2 s-1 for the three ranges of particles. In our manuscript, CS was calculated in the range of 10 nm-300 nm, which account for about 94% of particles less

than 2.5 $\mu$m. We believe that this size range is sufficient to calculate the condensational sink.

11. Line 138 "can be calculated" or was calculated?

Response: It should be "was calculated".

12. Are variables that are sensitive to the upper size limit (CCN concentrations that are based on the number of particles greater than size X, condensation sink) consistent given the change in the upper size limit shown in Figure S3, from campaign to campaign.

Response: The upper limit of the size is uniformly 300 nm in 2009-2018. In 2007, the upper limit of the size is 153 nm. The Dpgmax varied from 33 nm to 90 nm in 2007, and didn't affected by upper limit. But the CCN concentrations may be underestimated in some cases due to lack of data in >153 nm particles. For example, figure R7 showed the PNSD on April 23, 2007 when we calculated $\Delta$NCCN. The lognormal fitted curve showed about 15% of the area is missing. Thus, the $\Delta$NCCN might be underestimated in 2007. It will be clarified in revision. However, this will not affect our conclusion that net CCN production largely decreased in 2017–2018.

13. Line 282 – climate change typically requires 30-y averaging. Interannual variability may be much more likely at the time scales studied here.

Response: correct.

14. Line 293 – "data size was small" is vague. A more detailed description of what aspects of the dataset are too small is needed. Response: It will revised to "the data were obtain in seven independent campaigns, each lasted in 18~70 days, and the data size was small".

15. Line 294 – there are two issues: spatial representativeness, and sparsity of the record in time. In my opinion these create two different problems for the work. "the data size was small, and we should be cautious in extending the conclusion to a large

spatiotemporal scale"

Response: It will be revised to "1) the data were obtain in seven independent campaigns, each lasted in 20~70 days, and the data size was small, and 2) the observation was conduct in situ, and it should be cautious in extending the conclusion to a large spatiotemporal scale".

16. Line 299 – this shows the authors are thinking of these events as perfect Lagrangian experiments, where sampling at the mountain site is equivalent to sampling along a 0-D Lagrangian air mass trajectory. Vertical and horizontal mixing are not accounted for in this conceptual model. And the possibility that back trajectories evolve over the course of the PFGE is neglected. In reality, as the event evolves, winds will bring air with a variety of histories (chemical, emissions, radiation, accumulation mode particles, interaction with precipitation and clouds, etc.). The survival probabilities over 100% (Figure 4) are likely a symptom of the fact that reality has complex flows and spatial heterogeneity and does not fit the idealized box model concept.

Response: We agree that the vertical and horizontal mixing play an important role in the observed NPF events. If the ambient nucleation occurs aloft and newly formed particles mixes down to the height to be observed, the observed FR may be determined mainly by the downward moving rate of newly formed particles rather than the true formation rate of newly formed particles. Thus, we change "formation rate" to "apparent formation rate" in revision.

However, the growth rate and Dpgmax of newly formed particles were determined by concentrations and cumulative amounts of the condensed vapors, respectively. The condensed vapors are commonly believed to be generated from chemical reactions in air masses regardless of the moving rates of air masses in vertical and horizontal directions.

It does not make sense to calculate the SP beyond 100% because of highly spatial-heterogeneity of NPF in those particular events. In the revision, we added "Note that
the observed number concentrations of newly grown particles with a larger size some-times exceeded those with a smaller size under the condition of spatial heterogeneity of NPF. In these cases, SP was not calculated."

17. Figure 7 is of low resolution. Difficult to see some of the symbols, and symbols are of different sizes in different plots.

Response: corrected.

18. The discussions of biogenic and total VOCs throughout the paper are problematic. What species are these? How were they measured? Were the measurements collocated with the PFGE measurements and matched in time? The amount of oxidation needed to grow from 3 to 10 nm or 10 to 20 nm, is quite small, so making broad general-izations about seginficant changes in entire classes of VOCs or in specific compounds, and then connecting them to PFGE is not scientifically valid.

Response: The total VOCs in the June 2006 campaign was cited from Mao et al. (2009), since no data from the spring 2007 campaign were available. As many as 52 VOCs (C4–C12) were measured. The analyses method and the species list can be found in Mao et al. (2009). The total VOCs in the spring 2018 campaign were measured at the laboratory of the University of California at Irvine (UCI), and a total of 78 C2–C10 non-methane hydrocarbons (NMHCs) were measured. We acknowledge that the analyses methods were different in the two campaigns, and we will remove the discussion of VOC data in revision.

19. Rather than making the data available "on request", the data should be publicly posted in machine readable formats at the time of publication in order to allow replica-tion.

Response: Thanks, we will provide the website to access the raw data in revision.

Reference:

Birmili, W. and A. Wiedensohler (2000). "New particle formation in the continental

boundary layer: Meteorological and gas phase parameter influence." Geophysical Research Letters 27(20): 3325-3328.

Burkart, J., Hodshire, A. L., Mungall, E. L., Pierce, J. R., Collins, D.B., Ladino, L. A., Lee, A. K., Irish, V., Wentzell, J. J., Liggio, J., and Papakyriakou, T. (2017). Organic condensation and particle growth to CCN sizes in the summertime marine Arctic is driven by materials more semivolatile than at continental sites, Geophys. Res. Lett., 44, 10725–10734.

Dal Maso, M., M. Kulmala, I. Riipinen, R. Wagner, T. Hussein, P. P. Aalto and K. E. J. Lehtinen (2005). "Formation and growth of fresh atmospheric aerosols: eight years of aerosol size distribution data from SMEAR II, Hyytiala, Finland." Boreal Environment Research 10(5): 323-336.

Ehn, M., Thornton, J.A., Kleist, E., Sipilä, M., Junninen, H., Pullinen, I., Springer, M., Rubach, F., Tillmann, R., Lee, B., Lopez-Hilfiker, F., Andres, S., Acir, I., Rissanen, M., Jokinen, T., Schobesberger, S., Kangasluoma, J., Kontkanen, J., Nieminen, T., Kurtén, T., Nielsen, L.B., Jørgensen, S., Kjaergaard, H.G., Canagaratna, M., Maso, M.D., Berndt, T., Petäjä, T., Wahner, A., Kerminen, V., Kulmala, M., Worsnop, D.R., Wildt, J., and Mentel, T.F. (2014). A large source of low-volatility secondary organic aerosol, Nature, 506.

Gaie-Levrel, F., S. Bau, L. Bregonzio-Rozier, R. Payet, S. Artous, S. Jacquinot, A. Guiot, F. X. Ouf, S. Bourrous, A. Marpillat, C. Foulquier, G. Smith, V. Crenn and N. Feltin (2020). "An intercomparison exercise of good laboratory practices for nano-aerosol size measurements by mobility spectrometers." Journal of Nanoparticle Research 22(5): 13.

Kulmala, M., Petäjä, T., Nieminen, T., Sipilä, M., Manninen, H. E., Lehtipalo, K., Dao Maso, M., Aalto, P. P., Junninen, H., Paasonen, P., Riipinen, I., Lehtinen, K. E. J., Laaksonen, A., and Kerminen, V-M.(2012). Measurement of the nucleation of atmospheric aerosol particles, Nat. Protoc., 7, 1651–1667.

Lee, S., Gordon, H., Yu, H., Lehtipalo, K., Haley, R., Li, Y., and Zhang, R. (2019). New Particle Formation in the Atmosphere: From Molecular Clusters to Global Climate, J. Geophys. Res. Atmos., 124, 7098-7146.

Ma, L., Zhu, Y., Zheng, M., Sun, Y., Huang, L., Liu, X., Gao, Y., Shen, Y., Gao, H., and Yao, X. (2020). Investigating three patterns of new particles growing to cloud condensation nuclei size in Beijing's urban atmosphere, Atmos. Chem. Phys. Diss.

Mao, T., Wang, Y., Xu, H., Jiang, J., Wu, F., and Xu, X. (2009). A study of the atmospheric VOCs of Mount Tai in June 2006, Atoms. Envrion., 43, 2503-2508.

Man, H., Zhu, Y., Ji, F., Yao, X., Lau, N.T., Li, Y., Lee, B.P., and Chan, C.K. (2015). Comparison of Daytime and Nighttime New Particle Growth at the HKUST Supersite in Hong Kong, Environ. Sci. Technol., 49, 7170-7178.

Pfeifer, S., T. Müller, K. Weinhold, N. Zikova, S. Martins dos Santos, A. Marinoni, O. F. Bischof, C. Kykal, L. Ries, F. Meinhardt, P. Aalto, N. Mihalopoulos and A. Wiedensohler (2016). "Intercomparison of 15 aerodynamic particle size spectrometers (APS 3321): uncertainties in particle sizing and number size distribution." Atmospheric Measurement Techniques 9(4): 1545-1551.

Pitz, M.,W. Birmili, O. Schmid, A. Peters, H. E. Wichmann and J. Cyrys (2008). "Quality control and quality assurance for particle size distribution measurements at an urban monitoring station in Augsburg, Germany." Journal of Environmental Monitoring 10(9): 1017-1024.

Riipinen, I., Pierce, J. R., Yli-Juuti, T., Nieminen, T., Häkkinen, S., Ehn, M., Junninen, H., Lehtipalo, K., Petäjä, T., Slowik, J., Chang, R., Shantz, N. C., Abbatt, J., Leaitch, W. R., Kerminen, V.-M., Worsnop, D. R., Pandis, S. N., Donahue, N. M., and Kulmala, M. (2011). Organic condensation: a vital link connecting aerosol formation to cloud condensation nuclei (CCN) concentrations, Atmos. Chem. Phys., 11, 3865–3878.

Wang, Z.B., Hu, M., Pei, X.Y., Zhang, R.Y., Paasonen, P., Zheng, J., Yue, D.L., Wu,

Z.J., Boy, M., and Wiedensohler, A. (2015). Connection of organics to atmospheric new particle formation and growth at an urban site of Beijing, Atmos. Environ., 103, 7-17.

Wang, M., Kong, W., Marten, R., He, X., Chen, D., Pfeifer, J., Heitto, A., Kontkanen, J., Dada, L., Kürten, A.,Yli-Juuti, T., Manninen, H., Amanatidis, S., Amorim, A., Baalbaki, R., Baccarini, A., Bell, D., Bertozzi, B., Bräkling, S., Brilke, S., Murillo, U. C., Chiu, R., Chu, B., De Menezes, L-P., Duplissy, J., Finkenzeller, H., Carracedo, L. G., Granzin, M., Guida, R., Hansel, A., Hofbauer, V., Krechmer, J., Lehtipalo, K., Lamkaddam, H., Lampimäki, M., Lee, C.P., Makhmutov, V., Marie, G., Mathot, S., Mauldin, R. L., Mentler, B., Müller, T., Onnela, A., Partoll, E., Petäjä, T., Philippov, M., Pospisilova, V., Ranjithkumar, A., Rissanen, M., Rörup, B., Scholz, W., Shen, J., Simon, M., Sipilä, M., Steiner, G., Stolzenburg, D., Tham, Y. J., Tomé, A., Wagner, A. C., Wang, D. S., Wang, Y., Weber, S.K., Winkler, P. M., Wlasits, P. J., Wu, Y., Xiao, M., Ye, Q., Zauner-Wieczorek, M., Zhou, X., Volkamer, R., Riipinen, I., Dommen, J., Curtius, J., Baltensperger, U., Kulmala, M., Worsnop, D. R., Kirkby, J., Seinfeld, J. H., El-Haddad, I., Flagan, R. C., and Donahue, N. M. (2020). Rapid growth of new atmospheric particles by nitric acid and ammonia condensation, Nature 581, 184–189, 2020.

Wiedensohler, A., Cheng, Y.F., Nowak, A., Wehner, B., Achtert, P., Berghof, M., Birmili, W., Wu Z.J., Hu, M., Zhu, T., Takegawa, N., Kita, K., Kondo, Y., Lou, S.R., Hofzumahaus, A., Holland, F., Wahner, A., Gunthe, S.S, Rose, D., Su, H., Pöschl, U.(2009). Rapid aerosol particle growth and increase of cloud condensation nucleus activity by secondary aerosol formation and condensation: A case study for regional air pollution in northeastern china, J. Geophys. Res., 114, D00G08.

Wiedensohler, A., W. Birmili, A. Nowak, A. Sonntag, K. Weinhold, M. Merkel, B. Wehner, T. Tuch, S. Pfeifer, M. Fiebig, A. M. Fjaraa, E. Asmi, K. Sellegri, R. Depuy, H. Venzac, P. Villani, P. Laj, P. Aalto, J. A. Ogren, E. Swietlicki, P. Williams, P. Roldin, P. Quincey, C. Huglin, R. Fierz-Schmidhauser, M. Gysel, E. Weingartner, F. Riccobono, S. Santos, C. Gruning, K. Faloon, D. Beddows, R. M. Harrison, C. Monahan, S. G.

Jennings, C. D. O'Dowd, A. Marinoni, H. G. Horn, L. Keck, J. Jiang, J. Scheckman, P. H. McMurry, Z. Deng, C. S. Zhao, M. Moerman, B. Henzing, G. de Leeuw, G. Loschau and S. Bastian (2012). "Mobility particle size spectrometers: harmonization of technical standards and data structure to facilitate high quality long-term observations of atmospheric particle number size distributions." Atmospheric Measurement Techniques 5(3): 657-685.

Wiedensohler, A., A. Wiesner, K. Weinhold, W. Birmili, M. Hermann, M. Merkel, T. Muller, S. Pfeifer, A. Schmidt, T. Tuch, F. Velarde, P. Quincey, S. Seeger and A. Nowak (2018). "Mobility particle size spectrometers: Calibration procedures and measurement uncertainties." Aerosol Science and Technology 52(2): 146-164.

Zhang, R., Khalizov, A., Wang, L., Hu, M., and Xu, W. (2012). Nucleation and Growth of Nanoparticles in the Atmosphere, Chem. Rev., 112, 1957-2011, 2012.

Zhu, Y., Li, K., Shen, Y., Gao, Y., Liu, X., Yu, Y., Gao, H., and Yao, X. (2019). New particle formation in the marine atmosphere during seven cruise campaigns, Atmos. Chem. Phys., 19, 89–113.

Zhu, Y., Yan, C., Zhang, R., Wang, Z., Zheng, M., Gao, H., Gao, Y., and Yao, X. (2017) Simultaneous measurements of new particle formation at 1 s time resolution at a street site and a rooftop site, Atmos. Chem. Phys., 17, 9469–9484.
* * *
[Figure]

**Fig. 1.** Figure R1 Particle number size distribution on non-NPF days in each campaign (shaded areas are quarter of the standard deviations).

[Figure]

**Fig. 2.** Figure R2 The relationship between the PM2.5 mass concentration reported by TEOM 1400a or Thermo 5030 SHARP (x-axis) and the PM mass concentration derived from WPS (y-axis).

[Figure]

**Fig. 3.** Figure R3 Comparison of particle number concentration in 10-25 nm (a) and 10-300 nm (b) between WPS and SMPS during 2-7 July 2020.

[Figure]

**Fig. 4.** Figure R4 Time series of NPF events on 6 April 2018 and 7 April 2018: (a) contour plot of particle number size distribution using WPS data; (b) SO2 and NO2+O3; (c) mass concentration of SO42-, NO3- an

[Figure]

**Fig. 5.** Figure R5 Time series of NPF events on 5 April 2007 and 23 April 2007: (a) contour plot of particle number size distribution using WPS data; (b) SO2 and NO2+O3; (c) mass concentration of SO42-, NO3- a

[Figure]

**Fig. 6.** Figure R6 Relationship between the FR and NMINP in 106 cases of NPF events at Mt. Tai in this study and in urban and marine atmospheres in previous studies (Man et al., 2015; Zhu et al., 2017, 2019; M

[Figure]

**Fig. 7.** Figure R7 PNSD during 19:30-20:30 on April 23, 2007, when we calculated $\Delta$NCCN.

---

## Author Response (AR1)

*The manuscript analyzes seven field campaigns where particle number size distributions (PNSD) and sulfur dioxide were measured at the summit of a mountain site in the North China plain. Supporting measurements of time-resolved PM2.5, O3, and oxides of nitrogen were taken. And each campaign included 1-h time resolution ions in PM2.5 using water extractive methods (URG-AIM or MARGA). The most recent campaign was in 2018. Across the 7 campaigns, a little over 100 particle formation and growth events were detected, with the analysis focused on the size range of 10-300 nm size range. From the earliest to most recent campaign, SO2 emissions and concentrations have dropped dramatically, and the paper tries to analyze whether the particle formation and growth activity has changed in ways that are expected from the sulfur dioxide decrease. A large number of metrics are computed and then analyzed for each particle formation and growth event (PFGE). The metrics include, but are not limited to, the apparent formation rate of 10-25 nm particles (FR), the growth rate, the absolute increase in N10-25 particle concentration from the start to the peak of the PFGE (this is the NMINP variable), the PFGE duration, the PFGE frequency, the size to which the growth event reaches (Dpgmax variable in the manuscript), and particle counts which are used as surrogates for the change in CCN concentrations at low, medium, and high supersaturations (N100-300, N80-300, and N50-300). The paper includes values for and discussion of total VOC during the campaigns.*

*Complicating the analysis is that the field campaigns were in different months of the year: April 2007 (~30 d), June 2009 (~20 d), Aug 2014 (~30 d), Oct/Nov 2014 (~70 d), Jul 2014 (~40 d), Dec 2017 (~35 d), and Mar 2018 (~30 d).*

*The paper's abstract makes five claims: a. The formation rate in 2018 is 2-3 times higher than the formation rate in 2007. b. Net maximum increase in nucleation mode number concentration is 2-3 times higher in 2018 than in 2007. c. The occurrence of events where the mode of the growth event goes above 50 nm is lower in 2018 than it was in 2007. d. A surrogate for CCN production at high supersaturation (N50-300 at its peak during each growth event minus N50-300 before the event) decreased from 3703 per cm3 (before 2015) to 1026 (2017-2018). e. The authors argue availability of organic precursors has increased in the most recent campaigns, allowing more particle production and initial growth; furthermore, they argue that the lack of later growth is from reduction of "anthropogenic precursors" (presumably SO2).*

*The paper requires substantial revision before it is suitable for publication. The key issue, to this reviewer, is that making accurate claims about year-on-year trends and variability in PFGE is difficult. The requirements to make the claims defensible are: (1) take a sufficient number of samples to reduce random variability and give sufficient statistical power; (2) take steps to minimize, test for, and quantify campaign-specific systematic instrument bias (also known as "instrument drift"); (3) take steps to enforce consistency in any subjective data interpretation steps, such as classification of PFGE into "types" and the determination of the start and end times of events; (4) use statistical methods designed for trend analysis, time series analysis, and combined analysis of seasonal and interannual variability.*

*Each requirement needs to be met in order for the claims about trends to be defensible. And for*

*peer review and reproducibility purposes, things need to be documented for the peer-review and scientific communities.*

*I think the current work fails to meet all four of the requirements. While some of the conclusions are likely accurate (in that they would not change if all the requirements were met) – others would change, or require extensive qualification.*

**Response:** Thanks for the review's constructive comments. We agree that some analyses and related conclusions in our original version should be more conservative. In addition, more clarifications are needed to better defense them. Scientific community normally prefers to see the extracted trend in ambient variables using the measurement data over 20 years. Thus, the technical term "trend" used in the original version is problematic and has be removed.

Long-term continuous measurements may allow better investigating NPF trends, however, all statistical tools in literature suffer from the weakness to some extent in extracting the de-weathered trend in interested variables associated with anthropogenic perturbation, based on our previous studies. For non-continuous measurements, it is still a common challenge to address specific scientific questions using the proper statistical analysis. We thereby remove the technical "trend" through the whole manuscript in the revision.

A comparative analysis was conducted to study particle formation and growth events (PFGE) in different years based on two observational facts, 1) their occurrence frequencies in 2007 and 2009 were reasonably same as those in 2017 and 2018 even in different seasons; 2) a large decrease in $SO_2$ mixing ratio 2017 and 2018 against in 2007 and 2009. We then focused on comparative analyzing the spring PFGEs in 2007 and 2018, when uncertainties from varying ambient factors may have been largely minimized. This has been clarified in the revision.

We also found a large variation in occurrence frequency of summer PFGEs in different years. For the summer PFGEs, it may require extremely abundant chemical information to study the effect of decreasing $SO_2$ levels on PFGEs because of a huge perturbation from meteorological conditions and related biogenic emissions of air pollutants. On the other hand, the observations of summer PFGEs also implied that the same occurrence frequencies of PFGEs may be a critical indicator to constrain the comparative analysis.

We highly appreciate the comments on technical issues of measurements. The commercial instruments were routinely calibrated to ensure the QA/QC using the service provided by their vendors. We agree that more cautions should be paid to instrument limitations, especially in a small particle size range. In the following text, we try our best to address the comments point by point and revise the manuscript accordingly, in order to improve our analysis more defensible and robust. Please note that after the major revisions of our manuscript, we have also made substantial changes in this response, and the language-editing have been processed in this version.

*1. Statistical power:*

*PFGE exhibit substantial seasonal variation, due to changes in temperature, relative humidity,*

*biogenic activity, atmospheric chemistry, soil moisture, preexisting aerosol concentration and chemistry, radiation, cloudiness, boundary layer structure, land cover/vegetation canopy structure, synoptic meteorology, anthropogenic emissions, and atmospheric ion levels. Local meteorological features (i.e. orographic meteorology) and local sources may also have month-to-month variability. And at the 20-30 d time scale, large scale persistent geophysical features can cause a whole campaign of measurements to be atypically high or low for a number of PFGE variables. To accommodate all these sources of variability, large sample sizes are required for analysis of seasonal variation and interannual trends. In the absence of large sample sizes, careful pairing of events and analysis of alternate sources of variability / alternate hypotheses are needed isolate cause-effect relationships on specific PFGE variables.*

*With each campaign at a slightly different time of the year, some campaigns as short as 20 d, and no discussion of whether air pollution levels, air pollution meteorology, and climate variables were at climatologically representative levels, the reader has to apply great skepticism to any claims of interannual trends and cause-effect relationships for those interannual trends. See for example (Birmili and Wiedensohler 2000) who do take into account air mass characteristics.*

**Response:** To our best understanding from recent review papers on PFGE, eight variables, i.e., concentrations and cumulative generation amounts of sulfuric acid together with Highly Oxygenated Organic Molecules (HOM) and other secondary organics in different volatilities, the product of gaseous $HNO_3$ and gaseous $NH_3$ minus the equilibrium constant of $NH_4NO_3$ and then minus the kelvin effect term, cumulative generation amounts of condensed $NH_4NO_3$ on size-dependent particles, would directly affect apparent new particle formation rate (FR), apparent net maximum increase in the nucleation-mode particle number concentration (NMINP), new particle growth rate (GR) and the maximum geometric median diameter of grown new particles ($D_{pgmax}$). Although gaseous amines have been proposed to participate in ambient nucleation, their concentrations in China, based on the authors' work, are too high (relative to sulfuric acid) to act as the limitation factor. How aminium salts contribute the growth of newly formed particles larger than 10 nm is poorly understood in China, which is out of scope of this study. What the reviewer claim above may indirectly affect the eight variables to some extent. Only the information of the eight variables is not sufficient to support our analysis, those indirect factors should be cautiously utilized to facilitate the analysis. The arguments have been clarified in the revision (e.g., lines 40-44, lines 242-247, lines 341-345).

We agree that the technical term "trend" is misleading and has been removed in the revision. We also revised our discussion on the cause-effect relationships of PFGE by considering significant changes in the eight variables (lines 434-449).

In polluted and $NH_3$-rich (we may reasonably assume that amines may also be rich relative to sulfuric acid) ambient air in China, the influence of air mass characteristics on PFGE may be totally different from that in the clean and pristine atmospheres. In clean and pristine atmospheres, air mass characteristics may provide important information on precursors' sources of PFGE. In China, precursors of PFGE are abundant in general. For example, the occurrence frequency of PFGE reached ~50% in winter campaign as presented in this study, while the value was less than 5-10% in winter in Europe. Alternatively, air mass characteristics greatly affect moisture

characteristics and may subsequently affect the cloudiness (radiation) and $H_2SO_4$ concentration through modulating OH free radical concentration. Of course, the latter is poorly studied so far.

*The size distributions shown in Figure S3 are suggestive of insufficient number of days sampled in the dataset. Telling whether the system shifted from unimodal to bimodal behavior between 2015 and 2017 (all unimodal for 2015 and prior) vs. this occurring through some instrument drift vs. this occurring through sampling non-climatological conditions due to small samples sizes is difficult.*

*Given the decrease (Table 2) in PM2.5, sulfate, and SO2 between spring 2007 and 2018 (PM2.5 60 vs. 30 ug/m3; sulfate 17 vs. 4 ug/m3, SO2 18 vs. 3 ppb), more discussion is needed of the large increase in condensation sink in 2018 (Figure S3) and in the large increase in the height of the size distribution function at 100 and 150 nm between 2007 and 2015.*

*The discontinuity in the slope of the size distribution function at 200 nm also indicates there may be some drift in the size-specific performance of the WPS (Figure S3). The discontinuity in slope is not really evident until 2017, but then appears in 2017 and 2018.*

**Response:** We agree that the analysis of PFGE in the origin version misses out the important point. On the light of the comments, we re-checked the data. We do find that three channels in WPS around 213 nm suffered from unexpected errors in reporting number concentrations in approximately 30% sampling days in 2017 and 2018. It has a minor influence on PFGE in 7 NPF days out of the total of 32 NPF days in 2017 and 2018 as shown in Fig. S4 in the revised supplementary materials. The data in the three channels suffering from abnormal errors will be corrected by assuming a linear decrease of particle number concentration from 150 nm size bin to 300 nm size bin. Thank again.

We plotted particle number size distributions on non-NPF days in different years (Figure R1). Bimodal particle number size distributions can be observed in 2009, 2014 and 2015. We don't see any shift when the size distributions in 2014, 2015 and 2018 were compared with each other. We also find that the median geometric diameters of accumulation mode on NPF days in 2017 were consistent with those on non-NPF days in 2014, 2015 and 2018. However, the median geometric diameters of accumulation mode on non-NPF days in 2017 did shift to the large size and the same was true for those on non-NPF days in 2009. Moreover, the median geometric diameters of accumulation mode on NPF days in 2009 were consistent with those on non-NPF days in 2014, 2015 and 2018. Overall, our regular instrument maintenance appears to be effective to prevent from the instrument shift, except the occasional problem at three size bins around 213 nm.

In the origin manuscript, we calculated condensation sink based on the different size range of particle in 2007 (10-153 nm) and 2018 (10-300 nm). To be consistent, we recalculated CS based on 10-153 nm particles on 2018, and the average of $0.40 \pm 0.15$ s$^{-1}$ was slightly larger than the average CS in 2007 ($0.32 \pm 0.19$). CS should be unrelated to secondary particles via ambient nucleation since the particles cannot grow over 60 nm. Primary emissions of the accumulation mode particles were out of scape of this study, although they may scavenge precursors of PFGE to

some extent. This has been added in revision, lines 371-373.

[Figure]

Figure R1. Particle number size distribution on non-NPF days in each campaign (shaded areas are quarter of the standard deviations).

*2. Minimize, test for, and quantify campaign-specific instrument drift:*

*Achieving consistency in PNSD in long-term measurements is difficult. And it is not sufficient to state that each individual campaign had sufficient quality assurance, referring the reader to the campaign specific papers. There needs to be a presentation of data and discussion of how comparable the instrument responses are from campaign to campaign. What steps were taken to make sure instruments were not drifting. Aging of components can cause variation in flows, sizing accuracy, counting accuracy, particle losses, CPC supersaturations, and in the effective lower size limit of the instrumentation of the particle number spectrometer system. The detection efficiency as a function of size at the lower range of the instrument (5-25 nm), at the upper range of the mobility analyzer, at the lower end of the optical particle devices, and at the upper end of the optical particle analyzer – these are all difficult to maintain at stable levels over long periods of time. The total particle counts, height of the size distribution function, sensitivity at the lower and upper ranges of size distributions – these vary from year to year and require careful intercomparison, quality assurance, and maintenance procedures to deal with. See for example the results of intercomparison studies (Pfeifer, Müller et al. 2016) and papers focusing on quality assurance, calibration, and harmonization (Pitz, Birmili et al. 2008, Wiedensohler, Birmili et al. 2012, Wiedensohler, Wiesner et al. 2018, Gaie-Levrel, Bau et al. 2020). Comparison to other*

*instruments for total particle counts, size distribution functions in overlapping regions, checks with monodisperse particles are some of the techniques that can be used to establish more confidence and quantify campaign-to-campaign comparability.*

*Consistency in inlet dimensions, inversion algorithms (including multiple charge correction), use of impactors to manage multiple charge issues, corrections for inlet transmission efficiency, – these can all be issues in campaign-to-campaign comparability. They need to be discussed.*

*While being able to reproduce time-resolved PM2.5 measurements from the WPS size distribution is not sufficient to show accuracy in the nucleation and Aitken ranges – it is probably necessary. At least showing consistency from campaign to campaign in the volume of particles measured by the WPS and the mass of particles by time resolved mass measurements can help to demonstrate stability in instrumentation and data processing algorithms.*

*The fact that the authors are using an instrument with nominal lower cutoff of 5 nm, but discarding data between 5-10 nm indicates that there may be a problem with sensitivity at the lower size limit, or (more likely) variability in the sensitivity at the lower size limit. There is further evidence in Figures 1 and S6 – of a problem. In all the bursts shown save one, the particle size distribution function slopes down from a peak at about 13 nm to a lower value at 10 nm. If the instrument is biased low in the 10-13 nm range, then the statistics developed in the work will also be biased. If that bias varies from campaign to campaign, then that creates additional interpretation difficulties.*

*At line 180, it is implied that at times the WPS was collocated with instruments with lower limit of 3 nm. Therefore, the actual performance in the 5-15 nm range could (and should) be determined though comparison to such collocated instruments.*

**Response:** Honestly, we rely on the instrument vendor on instrument maintenance and calibration every 1-2 years. Except the problem at size bins around 213 nm sometimes occurring in 2017 and 2018, the measured size distributions were reasonably consistent as mentioned above. We excluded the concentrations of particles below 10 nm for analysis in order to keep the lower limit of PNSD consistent in seven campaigns (lines 110-113 in revision). More details are presented below.

In revision, lines 118-127, we have added: "The WPS instrument was calibrated and/or repaired every 1-2 years by its vendor. The regular maintenance allowed the WPS to perform well, based on the recent comparison results of the WPS and a new scanning mobility particle sizer (SMPS, Grimm) in the summer of 2020, as shown in Fig. S3. The regular calibration parameters included the DMA sample/sheath flow, LPS sample/sheath flow, DMA/CPC pressure, DMA voltage, and DMA/ambient temperature. Polystyrene latex (PSL) spheres (NIST) with mean diameters of 100.7 nm and 269 nm were used for calibration. At the beginning of each campaign, the zero-points of the DMA, CPC, and LPS were checked using a purge filter at the inlet. Sometimes, the WPS operated improperly, and the data were excluded from the analysis (see the Fig. S4 for the occasional unexpected errors in three channels around 213 nm). In addition, we reproduced the $PM_{2.3}$ mass concentration from the WPS data and found that it was reasonably comparable to the measured $PM_{2.5}$, further supporting the accuracy of the WPS data. Details can

be found in Fig. S5 and the supplementary materials."

In addition, the aging of components may lower the detection efficiency of WPS. However, the increased FR and NMINP in 2017-2018 reveal that the signals of nucleation mode particles enhanced in recent years. In lines 260-263, it reads as "During the four campaigns in 2007, 2009, and 2014, the calculated apparent FR varied narrowly in each campaign and the campaign average narrowed to 0.8–1.2 $cm^{-3}$ $s^{-1}$. The apparent FR increased in the three subsequent campaigns, i.e., $2.6 \pm 1.3$ $cm^{-3}$ $s^{-1}$ in 2015, $2.0 \pm 1.7$ $cm^{-3}$ $s^{-1}$ in 2017, and $3.0 \pm 2.7$ $cm^{-3}$ $s^{-1}$ in 2018." Therefore, we convince that the instrument maintenance can effectively reduce the aging impact of instrument components on observational data.

Conductive tubes (TSI 1/4 in.) were used for the WPS sampling, and the length of the tube was kept at approximately 2 m in each campaign. The SWS mode (DMA operating in the voltage-scanning mode) was selected. These have been added in lines 105-106, lines 108-109. The charge correction was calculated by the Boltzmann charge distribution, and the equation has been considered in the instrument algorithm.

The real-time $PM_{2.5}$ mass concentrations were measured in 2007, 2014, 2017 and 2018. Assuming that the particle density as 1.5 g $cm^{-3}$, the mobility diameter can be converted to the aerodynamic diameter with the equation: Aerodynamic diameter=Mobility diameter$\times \sqrt{1.5}$. The particle mass concentration in each size bin could be calculated according to the particle number concentration reported by WPS. The sum of the mass concentrations of particles less than 2.3 μm in aerodynamic diameter (1.9 μm in mobility diameter) was compared with the $PM_{2.5}$ mass concentration reported by TEOM 1400a (2007) or Thermo 5030 SHARP (2014-2018). The relationship of the hourly average data is shown in Figure R2.

For the 2007 data, we calculated the mass concentration of $PM_{0.18}$ from WPS and found that it has a weak correlation with $PM_{2.5}$. A slope of 0.05 indicated that the particles we observed accounted for only a minor fraction of the total mass of $PM_{2.5}$. For the two campaigns in 2014, the mass concentrations of WPS-derived $PM_{2.3}$ and SHARP measured $PM_{2.5}$ showed good correlations, with slopes of 0.69-0.76. For the 2017 and 2018 data, we removed the abnormal data in three bins around 213 nm. A good linear correlation was obtained between the two variables, but the slopes slightly increased to 0.86-0.9. Note that the calculation results depend on the adopted density of the particles. The actual particle density may deviate from the assumed value. In such cases, the sum volume (mass) may suffer from an error to some extent. The difference in the slopes may be partially related to the single value of particle density in different years, but other unknown factors cannot be excluded. Nevertheless, all of the deviations were less than 30%, and the linear correlations were generally good. However, the comparison alone cannot warrant the WPS operating properly. These have been added in the revised supplementary.

[Figure]

Figure R2: The relationship between PM$_{2.5}$ mass concentrations reported by TEOM 1400a (or Thermo 5030 SHARP) (x-axis) and PM mass concentrations derived from WPS (y-axis).

As we mentioned above, the detection limit of the DMA was 10 nm in 2007 and 2009, while it was 5 nm in 2014, 2015, 2017, and 2018. For consistency, only concentrations of >10 nm particles were used for the analysis. On April 7 (Fig. 1b in origin version and Fig. 1c in revised version), the initial peak at about 13 nm. That because the initial nucleation was influence by sporadic spikes, which overwhelmed the nucleation signal. In revision, lines 184-186, we added "Note that a few spikes were occasionally observed with a broader particle number size distribution during the NPF period. These spikes were excluded in the calculation of the FR, GR, D$_{pg}$, NMINP and CCN parameters (described in 2.2.2) because they may reflect primary particles from localized sources (Liu et al., 2014; Man et al., 2015; Zhu et al., 2017)." The revised D$_{pg}$ as shown in Fig. 1c.

The measurements made by NAIS at Mt. Tai were reported by Lv et al. (2018). But we have no confidence on the raw NAIS data. Fortunately, we have conducted a comprehensive comparison between the WPS and a SMPS (GRIMM, Germany) at a costal site in Qingdao, China, during 2-7 July 2020. The SMPS consists of a DMA (55-U,GRIMM) and a CPC (5416,GRIMM). It covers the particle size range of 10 nm-1000 nm, and is set up to 127 channels. The time resolution of SMPS is 4 min. The two instruments of WPS and SMPS were operated side by side for intercomparison.

Figure R3 shows the comparison of particle number concentration in the range of 10-25 nm (nucleation mode) and 10-300 nm (particle size range we used for calculation in this paper) between WPS and SMPS. Particle number concentration in these two size ranges showed good linear correlations, suggesting the measurements of the two instruments are highly consistent. Furthermore, the highly correlated data indicates that the WPS is not experiencing aging problems.

[Figure]

Figure R3 Comparison of particle number concentration in 10-25 nm (a) and 10-300 nm (b) between WPS and SMPS during 2-7 July 2020.

*3. Consistency in subjective data interpretation/classification steps:*

*It is not clear which of the variables used for analysis involve human classification. Sometimes, human classification is used for PFGE types (often using how smooth the growth event is in time); human classification is used sometimes for establishing times (start of event, end of the event). The end time is described. From line 113 of manuscript, "The end time of an NPF event was defined as the time when the particle number concentrations approached the background levels observed before the NPF event. The NPF event duration was defined as the time duration between the start time and end time of an NPF event." This seems like the end of event was a subjective determination of when background was approached. Thus the end time, duration, and any rate that has the duration in the denominator may be subjective.*

*For subjective (human) event classifications, were the events uniformly reclassified for this paper, or were prior classifications adopted from 2007 and 2009 and mixed with new classifications done for the more recent campaigns. See (Dal Maso, Kulmala et al. 2005) for best practices on human classification.*

**Response:** In revision, lines 152-156, we have added the details when classifying NPF events: "In this study, particles with diameters smaller than 25 nm were defined as nucleation mode particles (Kulmala et al., 2012). Following the criteria proposed by Dal Maso et al. (2005) and Kulmala et al. (2012), three features had to be met for an event to qualify as NPF: 1) a distinctly new nucleation mode particles must appear in the size distribution; 2) the new mode should prevail over a time span of hours; and 3) the new mode should show signs of growth. All three features are required for a day (00:00-23:59 LT) to be classified as an NPF day. Otherwise, the day is classified as a non-NPF day."

The definition of "NPF duration" has been added in lines 157-164: "The initial time of an NPF event was defined as when new nucleation mode particles started to be observed (e.g., $t_0$ in Fig. 1b, d). The end time of an NPF event was defined as the new particle signal dropping to a negligible level and the total particle number concentrations approaching the background levels

before the NPF event. In cases with the invasion of other plumes, the end time was determined to be when the new particle signals were suddenly overwhelmed by plumes and could no longer be identified (e.g., the end times in Fig. 1b, d). The NPF event duration was defined as the time duration between the initial time and end time of an NPF event. Note that the detection limit of WPS was 10 nm, but the particles were nucleated at critical cluster sizes of approximately 1-1.5 nm. Therefore, the NPF actually occurred at some time prior to our observation, and the actual duration should be longer than our calculation."

Followed the definition above, we classified the NPF event uniformly during the seven campaigns, i.e., from 2007 to 2018. In addition, the schematic diagram of $t_0$ and end time, as well as other NPF parameters such as NMINP and $\Delta N_{CCN}$ has been added in Fig. 1 (as shown in Figure R4). Moreover, Figure R4b also shows that the choice of $t_0$ may lead to underestimation of $\Delta N_{CCN}$ to some extent in the presence of spatial-temporal heterogeneity of pre-existing particles with diameters larger than 50 nm. In these cases, the mean value of $N_{CCN}$ in the percentiles smaller than $5^{th}$ during the whole NPF event may be more accurate representing the background (see the grey dashed line in Figure R4 b). However, this method may also introduce more subjective factors and therefore was not adopted in this study. These have been added in lines 214-218.

[Figure]

Figure R4 Examples of NPF events in three categories. Black dots in the figures are the fitted $D_{pg}$. (a) Category 1 on December 25, 2017, in which $D_{pgmax}$ was 24 nm (<50 nm), and Category 3 on December 24, 2017, in which $D_{pgmax}$ grew to 163 nm (>80 nm); (c) Category 2 on April 7, 2018, in which $D_{pgmax}$ grew to 53 nm (50–80 nm). (b, d) Schematic diagram of $t_0$, $t_1$, $t'_1$, NMINP and $\Delta N_{CCN100}/\Delta N_{CCN50}$ on December 24, 2017 and April 7, 2018 NPF events (a few spikes have been removed in Figure d).

*4. Statistical methods appropriate to analysis of combined seasonal and interannual variability*

*Statistical procedures for evaluating trends in seasonally varying time series need to be followed in order to state claims that trends exist. These can be found in a number of textbooks, papers, and government reports. See for example Statistical Methods for Environmental Pollution Monitoring by Gilbert https://www.osti.gov/servlets/purl/7037501/. And (Asmi, Coen et al. 2013, Collaud Coen, Andrews et al. 2013, Squizzato, Masiol et al. 2019). Many other good models for seasonally adjusted trend detection can be found in the O3, NOx, PM2.5, and hydrology/climatology literature. Squizzato et al. (2019) for example have the statistical procedures necessary to detect turning points (see line 236 where manuscript discusses turning points)*

*See for example line 290 "the CS still increased in 2018 compared with that in 2007." That implies annual average condensational sink increased, and this is a season or month specific result – and it is not clear there is enough statistical confidence to state this. Many other locations in the paper have broad statements about PFGE behavior in one year vs. another, or imply a long term trend where it has not really been shown.*

*Interpretation of PFGE data from this site seems more complicated than most, because of two issues: (1) it is sometimes influenced by boundary layer and other times by free troposphere; (2) very long PFGE events (see for example Figure 1a, where a 3-d long event is shown) are being compared with shorter (midday + afternoon) growth events. See Figure 7 which has events ranging from 3-h duration to 85-h duration. The flow patterns and chemistry required to sustain a 3-h event and an 80+ h event are likely very different, and would require more thoughtful comparison metrics than used in the paper.*

*The paper acknowledges this difficulty in interpretation (line 295) but more needs to be done than just acknowledge the difficulty. See analysis papers from PFGE studies at other high altitude sites. They do attempt to determine the degree of FT influence and the impact of polluted boundary layer air. And there are many papers that factor in air mass characteristics and/or back trajectory in analysis of PFGE.*

*See for example Figure 1a where on 25-Dec 2017 there were simultaneously occurring a short PFGE (category 1) and evolution of the category 3 event that started on 24-Dec 2017. This raises a number of questions on how such a dataset can be analyzed to determine trends.*

*How much of the variability in data is that some campaigns had more free tropospheric influence and others have less. How much of the conclusions of the paper are driven by switches (during PFGE) in air mass influence to/from FT influence. In other words, PFGE events that have their evolution dynamics controlled by airflows, and not by chemistry – hence the authors observed lack of influence or counterintuitive effects of SO2.*

*As for statistical procedures, I think it would be much more appropriate to put 95% confidence intervals on means rather than standard deviations on the plots. (Most figures have standard deviations)*

*Some of the variables appear to NOT be normally distributed (see figure S4) and thus use of statistical tests designed for normally distributed data are inappropriate.*

*Another weakness of the approaches used are that changes in boundary layer height are not accounted for. This weakness cannot really be addressed without additional measurements, but it*

*should be noted.*

**Response:** We acknowledge that the technical term "trend" is misleading in the origin manuscript. It has been removed in the revision. The "turn point" was inappropriate and we removed this in revision. We are also honest to say that we cannot understand those comments on other statistical issues.

Lines 370-373 (line 290 in origin version) has been revised as "Note that the campaign average PM$_{2.5}$ mass concentration in 2018 indeed decreased. The decrease was apparently determined by the decrease in >153 nm particles, since no significant difference existed in the calculated CS based on <153 nm particles between 2007 (0.32±0.19×10$^{-2}$ s$^{-1}$) and 2018 (0.40±0.15×10$^{-2}$ s$^{-1}$)." We didn't imply the annual trend of CS or other variables, we have checked the full text and revise the ambiguous statements.

Here we comprehensively analyze four cases of NPF events in different categories (category 1 events on 5 April 2007 and 6 April 2018, category 2 event on 8 April 2018), and category 3 event on 23 April 2007) in 2007 and 2018. The meteorological parameters, gases pollutants, PM2.5 mass concentrations and planetary boundary layer height (PBLH, download from https://goldsmr4.gesdisc.eosdis.nasa.gov/data/MERRA2/M2T1NXFLX.5.12.4/) were showed in figure R5 and R6. PBLH shows obvious diurnal variations, and the maximum value are 4110 m, 3000 m, 2316 m, and 2224 m on 6 April 2018, 7 April 2018, 5 April 2007, and 23 April 2007. There was no significant difference in the evolution of PBLH among the three categories. In revised Fig. S12, we also plotted the PBLH in three types of NPF events. In lines 425-427, we added "In addition, the changed boundary layer height had no detectable influence on D$_{pgmax}$ as shown in Fig. S12. However, the change in the late afternoon may largely decrease the observed number concentrations of grown new particles." We argued that PFGE events were controlled by chemistry, since the changes of airflow always associate with the changes of air pollutant, which directly influence NPF as mentioned in the response to the first comment.

As reported in numerous literatures, the growth of newly formed particles is mainly attributed to sulfuric acid, ammonium nitrate, and secondary organic compounds (Wiedensohler et al., 2009; Riipinen et al., 2011; Zhang et al., 2012; Ehn et al., 2014; Man et al., 2015; Wang et al., 2015; Burkart et al., 2017; Lee et al., 2019; Wang et al., 2020b). We therefore explore their respective contributions as follows. First, we calculated the contribution of sulfuric acid to the growth based on the observed mixing ratio of SO$_2$ and equations (5)-(6) in section 2.2.3. Second, we examined whether NH$_4$NO$_3$ freshly formed in PM$_{2.5}$ during the particle growth period. In case of no NH$_4$NO$_3$ formation, its contribution would not be expected. This is because an even higher product of HNO$_3$gas*NH$_3$gas is required to overcome the kelvin effect and form NH$_4$NO$_3$ in nucleation mode and Aitken mode particles. Thus, the growth unexplained by sulfuric acid should be mainly contributed by organics. In case of NH$_4$NO$_3$ formation, we considered the net increase in NH$_4$NO$_3$ may contribute to the particle growth, even though the ratios of increased NH$_4$NO$_3$ in PM$_{2.5}$ may not be the same as the ratios in nucleation mode and Aitken mode particles.

On 6 April 2018 (category 1), the NPF event was first observed at 09:10. D$_{pg}$ was fitted as 13 nm at 09:45, and continuous grow to 30 nm at 18:10. Then both of the particle number

concentration and particle diameter decreased, and the plume overwhelm the new particle signal at 6:00 on 7 April. During the NPF period, sulfuric acid was estimated to contribute about 16% to particle growth. The mass concentration of nitrate in $PM_{2.5}$ was less than 1.0 μg m$^{-3}$, implying that fresh $NH_4NO_3$ formation did not occur. Thus, the growth unexplained should be mainly contributed by organic matter.

On 7 April 2018 (category 2), $D_{pg}$ increased from 13 nm at 10:00 to 43 nm at 18:00, then $D_{pg}$ fluctuate at 41 nm-52 nm in the following 10 hours. Sulfuric acid was estimated to contribute about 11% to particle growth. The mass concentration of nitrate in $PM_{2.5}$ continuously increased from 0.8 μg m$^{-3}$ at 10:00 to 2.7 μg m$^{-3}$ at 20:00, then decreased to 2 μg m$^{-3}$ at 4:00 on 8 April 2018. Formation of ammonium nitrate seems to contribute to the growth of new particles in this case.

Similarly, on 23 April 2007 (category 3), sulfuric acid was estimated to contribute about 23% to particle growth. The mass concentration of nitrate in $PM_{2.5}$ increased from 1 μg m$^{-3}$ to 10 μg m$^{-3}$ during the particle growth period, indicating its important role in the particle growth. On the contrary, the mass concentration of nitrate and sulfate decreased during the NPF period on 5 April 2007 (category 1), and new particles didn't grow to the larger size.

We summarized the mass concentration of $SO_4^{2-}$, $NO_3^-$, $NH_4^+$ and OC during the formation and growth period of NPF events in 2007 and 2018 campaigns (added in revised Table 2). During the growth periods, the contribution of $H_2SO_4$ vapor to particle growth decreased from 36% in 2007 to 11% in 2018. The mass concentration of nitrate in $PM_{2.5}$ was 7.4 ± 4.8 μg m$^{-3}$ in 2007 during the new particle growth period, and it slightly decreased to 6.7± 5.5 μg m$^{-3}$ in 2018. In addition, OC in $PM_{2.5}$ was lower in 2018 (5.5 ± 2.0 μg m$^{-3}$) than in 2007 (6.1 ± 3.0 μg m$^{-3}$). In 2018 campaign, the reduced $H_2SO_4$ vapor, nitrate and OC formation may lead to the decrease in the growth probability of new particles. However, large uncertainties still exist because of a lack of data on the chemical composition of these smaller particles. These have been added in section 4.2.

Two sets of data in Fig. S4 (Fig. S10 in revision) are not linearly correlated, and we removed the fitting equations and changed the figure to scatter plots.

[Figure]

Figure R5 Time series of NPF events on 6 April 2018 and 7 April 2018: (a) contour plot of particle number size distribution using WPS data; (b) SO2 and NO2+O3; (c) mass concentration of SO42-, NO3- and NH4+ in PM2.5; (d) temperature (T) and relative humidity (RH); (e) wind speed and wind direction; (f) Planetary boundary layer height (PBLH).

[Figure]

Figure R6 Time series of NPF events on 5 April 2007 and 23 April 2007: (a) contour plot of

particle number size distribution using WPS data; (b) SO2 and NO2+O3; (c) mass concentration of SO42-, NO3- and NH4+ in PM2.5; (d) temperature (T) and relative humidity (RH); (e) wind speed and wind direction; (f) Planetary boundary layer height (PBLH).

*Other issues:*

*5. The abstract overstates the conclusions of the work. The conclusions have significant caveats, are based on limited number of sampled days, but the abstract makes it seem like the trends are well established, statistically significant, and based on a complete multi-year time series.*

**Response:** Thanks. We have removed the "trend" in revision and revised the abstract.

*6. There are a number problems with Figure 6. It is not appropriate to grey out datasets that are not correlated. Data are data, and data points should not be deemphasized visually just because they do not fit a linear correlation. The datasets should be clearly labeled so that each symbol type can be connected back to its underlying study and land cover type. Having a linear correlation shown and then a change in the tick mark spacing is not a fair way of graphing in my opinion. The size ranges in question should be included in the axis labels and/or the caption. I believe this is the formation rate at 10 nm, and the NMINP at 10-25 nm? Is that consistent for all the datasets? If not, then I don't think this is a fair plot to put in. I don't think having regression equations and correlation coefficients on graphs is effective or appropriate (see additional comments on this later).*

**Response:** Thanks. We have changed the grey markers to black markers, as shown in Figure R7. In this study at Mt. Tai station, all of the FR and NMINP are linearly correlated, and FRs were less than 15 cm$^{-3}$s$^{-1}$ (blue markers in Figure R7). Therefore, the linear relationship was the key point we would like to address, and we change the tick mark space when FR larger than 20 cm$^{-3}$s$^{-1}$ in order to emphasize our data at Mt. Tai.

We confirmed that the FR and NMINP were calculated based on 10-25 nm particles in all cases in this figure.

[Figure]

Figure R7 Relationship between the FR and NMINP in 106 cases of NPF events at Mt. Tai in this study and in urban and marine atmospheres in previous studies (Man et al., 2015; Zhu et al., 2017, 2019; Ma et al., 2020). Semi-solid markers can be fitted linearly in previous studies. Open markers show poor correlations.

*7. If a p-value appears in a figure or in the paper, then the statistical test needs to be discussed. What are the null and alternative hypothesis. And why is each hypothesis test implied by each p value important, scientifically interesting, novel, or useful?*

**Response:** The significance of P value is commonly required for any correlation or regression analysis and has been added in revision. We cannot understand what the reviewer wanted to comment. We are sorry for this.

*8. If a regression equation (e.g., y=12.5x+5.6) appears in a figure or in the paper, then its use – either for scientific or engineering purposes – needs to be discussed. The paper has 9 regression equations in it. Are they of any use?*

**Response:** Yes, the equation has its implication. For example, in type A, the $D_{pgmax}$ and GR can be fitted by the equation: $D_{pgmax} = 11.0 \times GR + 8.2$ with a moderate good *r*. Based on the obtained equation, newly formed particles appear to grow beyond 50 nm only when the GR exceeds 3.8 nm h$^{-1}$ in Type A. In addition, based on the regression equation between the duration and $D_{pgmax}$ obtained in 2007–2015 and 2017–2018, newly formed particles could grow beyond 50 nm only when the NPF duration exceeded 9.9 h in 2007–2015, but the duration in 2017-2018 had to exceed 27.8 h. In type B, following the regression equation of $D_{pgmax}$ against GR, newly formed particles

in Type B could grow beyond 50 nm only when the GR exceeded 5.8 nm h$^{-1}$. These have been added in lines 397-399, lines 400-402, lines 418-420.

*9. I believe all r values can be deleted from the paper without any loss.*

**Response:** Correlation coefficient is a statistical concept, which helps in establishing a relation between predicted and actual values obtained in a statistical experiment. The calculated value of the correlation coefficient explains the exactness between the predicted and actual values. The r values are used to measure the degree of correlation between two variables. A high r value (close to 1) means that the variables are highly correlated, and the fitted equation has its physical meaning. A small r value (close to 0) means the two variables are irrelevant, and the fitted equation is meaningless.

*10. Is the size range covered sufficient for calculating the condensational sink? Or stated differently, how much of the condensational sink is being missed by focusing on 10 to 150 or 250 nm.*

**Response:** We recalculated the CS in different size ranges, i.e., 10 nm-2.5 μm, 10 nm-300 nm and 10 nm-153 nm of particles in the campaign of 2018. The average CS values were $0.80\pm0.37\times10^{-2}$ s$^{-1}$, $0.75\pm0.34\times10^{-2}$ s$^{-1}$, and $0.40\pm0.15\times10^{-2}$ s$^{-1}$ for the three ranges of particles, respectively. The CS calculated by the use of particles in the range of 10 nm-300 nm accounted for 94% of those in the range of 10 nm-2.5 μm. Hence, the size range of 10-300 nm is sufficiently accuracy to estimate the condensational sink for comparing among different campaigns. However, the CS calculated by the use of particles in the range of 10 nm-153 nm accounted for only half of that in the range of 10 nm-2.5 μm. Thus, the CS calculated by the use of particles in the range of 10 nm-153 nm in 2018 was also used to compare with that in 2007. These have been added in lines 170-176, lines 370-373 and the supplementary materials.

*11. Line 138 "can be calculated" or was calculated?*

**Response:** It should be "was calculated".

*12. Are variables that are sensitive to the upper size limit (CCN concentrations that are based on the number of particles greater than size X, condensation sink) consistent given the change in the upper size limit shown in Figure S3, from campaign to campaign.*

**Response:** In this study, 10–300 nm particles were used to calculate $\Delta N_{CCN}$ in all campaigns except for that in spring 2007. In the particular year, the data of >153 nm particles were missing. Alternatively, the data of 10–153 nm particles were used for the calculations. The lack of 153-300 nm particles may have led to the smaller $\Delta N_{CCN}$ in 2007. For example, Figure R8 shows the PNSD at 19:30-20:30 on April 23, 2007, when we calculated the CCN number concentrations, i.e., $N_{CCN}(t'_1)$. On that day, the maximum size of geometric median diameter of the grown new particles ($D_{pgmax}$) was the largest (89 nm) during the spring campaign in 2007. The lognormal fitted curve showed that approximately 15% of the area was missing to gain a complete accumulation mode, suggesting that $N_{CCN}(t'_1)$ was underestimated by ~15%. In other events, the $D_{pgmax}$ was smaller, and the missing areas in the PNSD curve caused even smaller underestimation. Thus, it is safe to say that the lack of data for >153 nm particles had a negligible effect on the calculated $\Delta N_{CCN}$ in 2007. These have been added in lines 219-222 and the supplementary materials.

[Figure]

Figure R8 PNSD during 19:30-20:30 on April 23, 2007, i.e., $N_{CCN}(t'_1)$.

*13. Line 282 – climate change typically requires 30-y averaging. Interannual variability may be much more likely at the time scales studied here.*

**Response:** It has been revised to "Previous studies have reported that the BVOC emissions over the NCP have substantially increased in the last decade because of rapid afforestation and accelerating global warming."

*14. Line 293 – "data size was small" is vague. A more detailed description of what aspects of the dataset are too small is needed.*

**Response:** It has been revised to "the data were obtained in seven independent campaigns, each

lasting in 18~71 days, and the data size did not allow us to extend the conclusion to all the years from 2007 to 2018."

*15. Line 294 – there are two issues: spatial representativeness, and sparsity of the record in time. In my opinion these create two different problems for the work. "the data size was small, and we should be cautious in extending the conclusion to a large spatiotemporal scale"*

**Response:** It has been revised to "However, uncertainties still exist, e.g., 1) the data were obtained in seven independent campaigns, each lasting in 18~71 days, and the data size did not allow us to extend the conclusion to all the years from 2007 to 2018; 2) the observations were conducted only at one site, alternating between the boundary layer and the free troposphere, and the generality of the conclusions on NPF events needs to be examined at more sites.".

*16. Line 299 – this shows the authors are thinking of these events as perfect Lagrangian experiments, where sampling at the mountain site is equivalent to sampling along a 0-D Lagrangian air mass trajectory. Vertical and horizontal mixing are not accounted for in this conceptual model. And the possibility that back trajectories evolve over the course of the PFGE is neglected. In reality, as the event evolves, winds will bring air with a variety of histories (chemical, emissions, radiation, accumulation mode particles, interaction with precipitation and clouds, etc.). The survival probabilities over 100% (Figure 4) are likely a symptom of the fact that reality has complex flows and spatial heterogeneity and does not fit the idealized box model concept.*

**Response:** We agree that the vertical and horizontal mixing play an important role in the observed NPF events. If the ambient nucleation occurs aloft and newly formed particles mixes down to the height to be observed, the observed FR may be determined mainly by the downward moving rate of newly formed particles rather than the true formation rate of newly formed particles. Thus, we change "formation rate" to "apparent formation rate" in revision.

However, the growth rate and $D_{pgmax}$ of newly formed particles were determined by concentrations and cumulative amounts of the condensed vapors, respectively. The condensed vapors are commonly believed to be generated from chemical reactions in air masses regardless of the moving rates of air masses in vertical and horizontal directions.

It does not make sense to calculate the SP beyond 100% because of highly spatial-heterogeneity of NPF in those particular events. In the revision, we added "Note that the spatial-temporal heterogeneity during NPF events may result in high SPs. If the observed $\Delta N_{CCN}$ exceeded NMINP and the calculated SPs exceeded 100%, suggesting equation (3) was not applicable in these cases, and SP was therefore not calculated."

*17. Figure 7 is of low resolution. Difficult to see some of the symbols, and symbols are of different*

*sizes in different plots.*

**Response:** corrected.

*18. The discussions of biogenic and total VOCs throughout the paper are problematic. What species are these? How were they measured? Were the measurements collocated with the PFGE measurements and matched in time? The amount of oxidation needed to grow from 3 to 10 nm or 10 to 20 nm, is quite small, so making broad generalizations about seginficant changes in entire classes of VOCs or in specific compounds, and then connecting them to PFGE is not scientifically valid.*

**Response:** The total VOCs in the June 2006 campaign was cited from Mao et al. (2009), since no data from the spring 2007 campaign were available. As many as 52 VOCs ($C_4$–$C_{12}$) were measured. The analyses method and the species list of VOCs can be found in Mao et al. (2009). In the spring 2018 campaign, a total of 30 whole-air samples were collected on 9 days. The concentrations of VOCs were then quantified by gas chromatography (GC) separation followed by flame ionization detection (FID), mass spectrometry detection (MSD), and electron capture detection (ECD) at the laboratory of the University of California at Irvine (UCI). As many as 75 VOCs ($C_2$–$C_{10}$) non-methane hydrocarbons (NMHCs) were measured. The analyses method can be found in Chen et al. (2020). We acknowledge that the analytical methods were different in the two campaigns, and the difference in the VOCs has been added in revision, lines 365-366.

In addition, a recent study reported that China has experienced rapid afforestation, with provincial forest areas increasing by between 0.04 million and 0.44 million hectares per year over the past 10 to 15 years, leading to a large increase in the $CO_2$ sink based on long-term observations over large spatial scales (Wang et a., 2020a). BVOC emissions in China are reasonably expected to greatly increase over the past 15 years. This was highly consistent with the difference between the VOCs concentration at Mt. Tai in 2006 and 2018. However, a large knowledge gap between the increase in BVOC emissions and the increase in nucleating organics still exists because of a lack of studies. These have been added in revision, lines 67-71 and lines 367-369.

*19. Rather than making the data available "on request", the data should be publicly posted in machine readable formats at the time of publication in order to allow replication.*

**Response:** Thanks, we have provided the website to access the datasets in revision.

**Anonymous Referee #4**

*This paper investigates long-term behavior of new particle formation (NPF) and associated particle growth at an elevated site. This is an important and scientifically very interesting topic, since there are quite limited number of studies about the response of NPF to SO2 emission reductions, and since the obtained results are somewhat mixed between different environments. The fact that the study is based on relatively short-term measurement campaigns made in different seasons, rather than continuous measurements over full years, limits the reliability of the obtained results, and this should be properly acknowledged in the paper. Anyway, I would support publication of this paper, provided that the authors will address the issues raised below.*

**Response:** Thanks for the reviewer's comments. We agree that long-term continuous measurements would allow better investigating NPF trends under changing ambient conditions including air pollutants and meteorological factors. Due to practical difficulties, we tried short-term measurement campaigns made in different seasons of multiple years to characterize the NPF, with particular attention to the response of NPF to $SO_2$ emission. The limitations of the non-continuous measures and the uncertainties to explain the results have been added in the revised discussion. We also try our best to respond the comments and revise our manuscript accordingly.

Please note that after the major revisions of our manuscript, we have also made substantial changes in this response, and the language-editing have been processed in this version.

*The introduction of this paper is generally well written. However, it would benefit from having a more concrete list of scientific questions aimed to addressed (in addition the aim mentioned on lines 75-76) in this paper. Two other, minor issues in this section: 1) the term "functions" on line 34 does not sound correct, perhaps "mechanisms"?, 2) the statement on line 74-75 is unclear. What altitudes are authors referring to here, above the boundary layer in general or upper free troposphere? One should be more careful with this, as elevated NPF can be associated with many different things, including convective uplift, presence of clouds, mixing of different air masses etc.*

**Response:** Thanks. Lines 75-76 (lines 87-95 in this version) was revised as follow: "The main purpose of this study included: 1) to examine the effects of reduced $SO_2$ emissions on regional NPF events at a high altitude (from the upper boundary layer to the lower free troposphere), i.e., the changes in the NPF frequency, intensity and the subsequent growth of new particles; 2) to quantify the potential contribution of new particles to the CCN population and its changes under decreasing $SO_2$ emission; and 3) to rationalize the variation patterns of the NPF characteristics and CCN parameters in terms of observational concentrations of gaseous precursors and their origins and atmospheric behaviors. Note that all these changes in the study area should occur under the background of a rapid increase in BVOCs and their oxidized products as nucleating precursors

over the decades in China, although no studies can confirm the decadal increase in nucleating precursors from biogenic VOCs because of the lack of related analytic technologies in the past."

In the revision, we change ""functions" to "with different health and climate effects". On line 74 (line 86 in this version), we change "at higher elevations" to "above the boundary layer".

We agree that convective uplift, presence of clouds, mixing of different air masses, etc., may affect NPF events in clean remote atmospheres. We revise our discussion on the light of the issue.

*Experimental methods have been described very shortly, and should be expanded a bit. How were the measurement data used in the current paper quality checked, are these data undergoing any quality assurance procedures? Did detection limits etc. cause any issues for data interpretations? Were there any serious gaps in the data during the periods chosen for the current study?*

**Response:** Agree. The information will be added in the revised manuscript as following.

In revision, lines 118-127, we have added: "The WPS instrument was calibrated and/or repaired every 1-2 years by its vendor. The regular maintenance allowed the WPS to perform well, based on the recent comparison results of the WPS and a new scanning mobility particle sizer (SMPS, Grimm) in the summer of 2020, as shown in Fig. S3. The regular calibration parameters included the DMA sample/sheath flow, LPS sample/sheath flow, DMA/CPC pressure, DMA voltage, and DMA/ambient temperature. Polystyrene latex (PSL) spheres (NIST) with mean diameters of 100.7 nm and 269 nm were used for calibration. At the beginning of each campaign, the zero-points of the DMA, CPC, and LPS were checked using a purge filter at the inlet. Sometimes, the WPS operated improperly, and the data were excluded from the analysis (see the Fig. S4 for the occasional unexpected errors in three channels around 213 nm). In addition, we reproduced the $PM_{2.3}$ mass concentration from the WPS data and found that it was reasonably comparable to the measured $PM_{2.5}$, further supporting the accuracy of the WPS data. Details can be found in Fig. S5 and the supplementary materials."

In revision, lines 133-137, we have added: "For the analysers of $SO_2$, $O_3$, NO and $NO_2$, we performed multipoint calibrations every month and changed the filter every two weeks. The detection limits of $SO_2$, $O_3$, NO and $NO_2$ were 1 ppb, 0.4 ppb, 0.04 ppb, and 0.4 ppb, respectively. $PM_{2.5}$ was measured using a TEOM 1400a in 2007 and a Thermo 5030 SHARP after 2014. This device was calibrated by mass foil calibration according to the instrument manual, and the detection limit was 0.5 $\mu g/m^3$."

In revision, lines 140-143, we have added: "A multipoint calibration was performed for the online MARGA before and after the field campaigns to examine the sensitivity of the detectors. The detection limits were determined to be 0.05, 0.05, 0.04 and 0.05 $\mu g\ m^{-3}$ for $Cl^-$, $NO_3^-$, $SO_4^{2-}$ and $NH_4^+$, respectively. More details about the instrument calibration can be found in Wen et al. (2018) and Li et al. (2020)."

*Concerning the calculation methods, the authors should explicitly mention in main text (section*

*2.2.1) at which size particle formation rates were calculated, and what size range the calculated particle growth rates refer to (or if the applied size range for this calculation varied from event to event). Also, definition of "NPF duration" referred to e.g. on line 298 should explicitly described. Is it the time period over which new particles are observed to appear at the smallest sizes, or the time period over which the growth of new particle to larger sizes can be followed.*

**Response:** Thanks. In revision, lines 166-168, we have added "The apparent FR of new particles is calculated based on nucleation mode particles with sizes of 10-25 nm. The GR is quantified by fitting the geometric median diameter of new particles ($D_{pg}$) during the whole particle growth period. The size range of $D_{pg}$ varies from event to event."

The definition of "NPF duration" has been added in lines 157-164: "The initial time of an NPF event was defined as when new nucleation mode particles started to be observed (e.g., $t_0$ in Fig. 1b, d). The end time of an NPF event was defined as the new particle signal dropping to a negligible level and the total particle number concentrations approaching the background levels before the NPF event. In cases with the invasion of other plumes, the end time was determined to be when the new particle signals were suddenly overwhelmed by plumes and could no longer be identified (e.g., the end times in Fig. 1b, d). The NPF event duration was defined as the time duration between the initial time and end time of an NPF event. Note that the detection limit of WPS was 10 nm, but the particles were nucleated at critical cluster sizes of approximately 1-1.5 nm. Therefore, the NPF actually occurred at some time prior to our observation, and the actual duration should be longer than our calculation."

In addition, the schematic diagram of $t_0$ and end time, as well as other NPF parameters such as NMINP and $\Delta N_{CCN}$ has been added in Fig. 1 (as shown in Figure R1).

[Figure]

Figure R1 Examples of NPF events in three categories. Black dots in the figures are the fitted $D_{pg}$. (a) Category 1 on December 25, 2017, in which $D_{pgmax}$ was 24 nm (<50 nm), and Category 3 on December 24, 2017, in which $D_{pgmax}$ grew to 163 nm (>80 nm); (c) Category 2 on April 7, 2018, in which $D_{pgmax}$ grew to 53 nm (50–80 nm). (b, d) Schematic diagram of $t_0$, $t_1$, $t'_1$, NMINP and $\Delta N_{CCN100}/\Delta N_{CCN50}$ on December 24, 2017 and April 7, 2018 NPF events (a few spikes have been removed from figure d).

*Categorizing NPF event based on the maximum size that the formed particle are able to reach by*

*growth is in principle fine. However, doing that has one important issue that should be at least mentioned, and preferably shortly discussed, in the text. Following particle growth over several days, or event over the night, from observations is often difficult because of the typically large diurnal variation of boundary layer properties (e.g. mixed layer height), and because of changes in measured air masses. This can be seen, for example, on Dec 24 NPF event shown in Figure 1a: there are at least two major discontinuities in the particle number size distribution data (apparent in sudden huge changes in particle number concentration in certain size ranges). As a result, it is highly questionable whether the particles observed to reach 217 nm actually initiated from the NPF event that took place much earlier on Dec 24. The same issues concerns the use of the term SP (survival probability). SP works fine when following the particle growth for a few hours, but becomes questionable for larger time periods. The authors should replace the term "survival probability" with something like "apparent survival probability" and discuss shortly this issue in the paper, including when interpreting the results.*

**Response:** In this study, $D_{pgmax}$ and SP were calculated within the NPF duration. The definition of NPF duration has been clarified in the response above and has been added in the revision. In addition, a few spikes were excluded in calculating $D_{pgmax}$ and SP since the spikes were characterized by a sudden change in particle number size distribution (PNSD) and may reflect primary particles from localized sources. These have been clarified in revision, lines 184-186.

As reported in numerous literatures, NPF was a regional phenomenon occurring in a spatial extent varies from tens to thousands of kilometers (Kulmala et al., 2012) However, it is almost impossible to occur identic NPF events over the large spatial range because of different concentration levels of nucleation precursors. In reverse, spatial heterogeneity of regional NPF events is a common phenomenon and would cause the discontinued PNSD to some extent at a fixed observational site.

No criteria have been well-established in the literature to identify spatial heterogeneity of NPF events. In this study, spatial heterogeneity of NPF events was assumed for the discontinued PNSD if no intrusion of primary or aged plume signals were clearly identified. Even though the intrusion of primary or aged plume signals overwhelmed new particle signals for a short period, new particle signals can still be reasonably observed afterwards in the contour plotting. The discontinued PNSD was also assumed as the continuity of the NPF event. The NPF event on Dec. 24 was illustrated as example:

On the Dec. 24 event, (Figure R1a-b), the two discontinuities in PNSD appeared at 7:00-9:00 and 16:10-18:50 on Dec. 25. From 22:48 on Dec. 24 to 7:00 on Dec. 25, $N_{total}$ continuous decreased from $1.1 \times 10^4$ cm$^{-3}$ to $0.8 \times 10^4$ cm$^{-3}$, meanwhile, $D_{pg}$ grew from 30 nm to 63 nm with the growth rate of 4.0 nm/h. Between 7:00 and 9:00, $N_{10-300nm}$ and $D_{pg}$ fluctuate at $0.8 \pm 0.1 \times 10^4$ cm$^{-3}$ and $51 \pm 6$ nm. At 9:00, $N_{10-300nm}$ and $D_{pg}$ were $0.8 \times 10^4$ cm$^{-3}$ and 72 nm, respectively. During the two hours (7:00-9:00), new particles were hypothesized to experience a growth similar to the previous curve. Similarly, $D_{pg}$ was 115 nm at 16:10 and 128 nm at 18:50. The assumed growth rate during the 2.7 hours was about 4.8 nm/h, close to the previous GR. Actually, when we fitted the $D_{pg}$ between 22:48 on Dec. 24 and 19:30 on Dec. 25, the GR was 4.8 nm/h and $R^2=0.97$, suggesting that particles grew in a smooth curve (Figure R2).

In this event, $D_{pg}$ reached the maximum of 163 nm at 8:30 on Dec. 26, and then the new particle signals were overwhelmed by plumes after 9:00 on Dec. 26 (referred as end time). Followed equation (2), $\Delta N_{CCN}$ reached the maximum at 20:03 on 25 Dec. The SP was calculated as SP = $\Delta N_{CCN}$/NMINP, and $SP_{CCN50}$, $SP_{CCN80}$, and $SP_{CCN100}$ was calculated to be 0.2, 0.2 and 0.15, respectively. However, Figure R1b also shows that the choice of $t_0$ may lead to underestimation of $\Delta N_{CCN}$ to some extent in the presence of spatial-temporal heterogeneity of pre-existing particles with diameters larger than 50 nm. In these cases, the mean value of $N_{CCN}$ in the percentiles smaller than 5$^{th}$ during the whole NPF event may be more accurate representing the background (see the grey dashed line in Figure R1b). However, this method may also introduce more subjective factors and therefore was not adopted in this study. These have been clarified in the revision (lines 214-218).

We agree to change "survival probability" to "apparent survival probability". The judgment of spatial heterogeneity in other NPF events followed the similar approach above, and we have add the discussion of spatial heterogeneity in the revised manuscript, lines 209-211: "Note that the spatial-temporal heterogeneity during NPF events may result in high SPs. If the observed $\Delta N_{CCN}$ exceeded NMINP and the calculated SPs exceeded 100%, suggesting equation (3) was not applicable in these cases, and SP was therefore not calculated."

[Figure]

Figure R2 Growth curve of the fitting the geometric median diameter of new particles ($D_{pg}$) from 22:48 on Dec. 24 to 19:30 on Dec. 25.

*The authors should be a bit more careful when using the term "trend". On line 191, for example, should there read "pattern" rather than "trend"? Multi-year trends can be season-dependent, but I suppose this not what the authors mean here. Please check out that "trend" is correctly used throughout the paper.*

**Response:** Thanks. We agree the reviewer's comment that "trend" was inappropriate in this paragraph. Our main purpose was to examine the effects of reduced $SO_2$ emissions on the regional NPF events characteristics and the CCN parameters. We have change "trend" to "pattern" in this sentence and go through the full text and revise the ambiguous statements.

Reference:

[revised manuscript text omitted]
_{2.5}$ ($\mu g\ m^{-3}$) | 56.5 ± 33.0 | 71.1 ± 49.0 | 30.3 ± 21.8 | 29.2 ± 20.4 |
| $SO_4^{2-}$ ($\mu g\ m^{-3}$) | 16.4 ± 11.0 | 18.5 ± 9.7 | 3.3 ± 2.4 | 3.6 ± 2.7 |
| $NO_3^-$ ($\mu g\ m^{-3}$) | 7.4 ± 5.7 | 7.4 ± 4.8 | 6.3 ± 5.1 | 6.7 ± 5.5 |
| $NH_4^+$ ($\mu g\ m^{-3}$) | 5.5 ± 4.2 | 6.1 ± 3.5 | 2.3 ± 1.7 | 2.2 ± 1.6 |
| Calculated $H_2SO_4$ ($10^7$ molecules·$cm^{-3}$) | 8.8 ± 4.9 | 9.4 ± 4.5 | 2.2 ± 1.1 | 2.4 ± 1.0 |
| $[H_2SO_4]_{avg}$/C (%) | 59 ± 23 | 36 ± 18 | 23 ± 10 | 11 ± 7 |
| TVOC (ppb) | 7.0 ± 5.7[a] | | 16.1 ± 6.5 | |
| OC ($\mu g\ m^{-3}$) | 6.1 ± 3.0[b] | | 5.5 ± 2.0[c] | |
| EC ($\mu g\ m^{-3}$) | 1.8 ± 1.6[b] | | 1.3 ± 0.6[c] | |

[a](Mao et al., 2009)

[b](Wang et al., 2011)

780   [c](Dong et al., 2020)

---

## Author Response (AR2)

**Response to comments**

*Suggestions for revision or reasons for rejection (will be published if the paper is accepted for final publication)*
*Thank you for the detailed consideration of the reviewer comments from the first set of reviews at ACPD. I appreciate the checks of key data against independent instruments. I believe this article should be published after minor revisions as outlined below.*

**Response:** The authors appreciate the comments to improve the quality of this study. We will try our best to respond the comments and revise our manuscript accordingly.

*Major issues*

*1. The discussion of increased biogenic VOC and increased HOM, due to afforestataion, as the reason for the observed changes in the apparent formation rate of 10 nm particles is speculative. It should be labeled as a hypothesis. Currently, it is written as if it is a conclusion. It is prominent in the abstract, and in the conclusions. There are qualifications in the paper, but they are not given the prominent placement and strong language that the "conclusion" that BVOC and HOM due to afforestation are a causative agent to the features seen in the manuscript. The BVOC/HOM hypothesis, and its tie to afforestation, should be confined to the discussion and clearly labeled as a hypothesis supported by some suggestive evidence. The evidence presented is insufficient to support it as a firm conclusion.*

**Response:** We agree that no direct observations on long-term trends in BVOC/HOM can be provided to support our analysis at present. We thereby tone down the point throughout the manuscript in the revised version.

The technologies to determine HOM are commercially available recently, therefore, it is practically impossible to gain the data of HOM ten years ago. In addition, BVOCs are highly reactive and have a short life time in ambient air. Therefore, the observed concentration of BVOC always reflects a highly localized feature. On the other hand, NPF events occur in a regional scale. To gain long-term trends in BVOC across regional areas is extremely difficult. To best of our knowledge, no direct observational studies on the trends have been reported in the literature. Thus, our analysis from the rapid afforestation in the literature and those satellite data of leaf area index (LAI) should be given a credit.

*2. The language around the increase in VOC overplays the scientific strength of the underlying data. "large increase" (line 20), "rapid afforestation" (line 21), biogenic emissions "changing significantly" (line 75), "rapid increase in BVOCs and their oxidized products as nucleating precursors" (line 94), "difference was highly consistent with the large increase in forest area" (line 365). The authors think it very important that we believe this is a large, rapid, and statistically significant phenomenon. This requires data, not words. The actual data this is based on is much more modest than the language used to describe the VOC changes – and is not actually quantitatively stated in the paper. The percentage increase in springtime BVOC emissions due to*

*land use change is never stated in the paper. It is difficult to determine from the cited works too (discussed below).*

*The evidence is limited to the land use map (figure S2) where the increase in tree cover is substantial but not overwhelming, and not consistent in all directions. There are the canister measurements (see comment 3). And there are citations; but actually reading Ma et al. (2019), it is difficult to tell how much the BVOC increase was relative to baseline, where it is happening, and during what seasons it is important. Ma et al. is focused on attributing ozone concentrations during a heat wave to BVOC emission changes that are the result of a heat wave, urban forestry, and land use change. The ozone changes are extensively discussed, the relative change in VOC emissions due to land use change are much less discussed. The delta emissions are mapped in Ma et al. (2019), and appear to be minimal at the location of Mt. Tai. Wang et al. (2020) is also not that supportive. The map shows the delta of carbon fluxes most concentrated well to the south of Mt. Tai, and the time series of above ground biomass increases from 40 to about 43 over about 20 years, or about 0.3% increase per year. A large carbon sink, but not support for "rapid increase in BVOCs and their oxidized products as nucleating precursors." Finally, in the introduction to Ma et al. (2019), another work is cited with increase due to climate change and land cover change may be 1-1.5% per year. This is substantial, and over 11 years could lead to a ~15% change in average regional emissions – I think to claim this is "large and rapid" is stretching and distracts from the strength of the paper, which is the data itself, and the careful calculation of the nucleation and growth parameters.*

**Response:** The arguments on the rapid or large increase in BVOCs and their oxidized products are indeed speculated and should be toned down. We thereby delete the type of words in the revision. However, the increase in afforestation is indeed rapid in China as reported by Chen et al. (2019), i.e., a net increase of ~18% in leaf area from 2000 to 2017. More important, they reported that the leaves green percentage in China is much large than that in developed countries and potentially emit more VOC. The BVOC emissions in China are also simulated to increase over the past decades using the MEGAN model (Zhang, et al., 2016; Chen et al., 2017). The references have been added in the revision.

*3. The difference in VOC concentrations between June 2006 (7.0 ± 5.7 ppb) and spring 2018 (16.1 ± 6.5 ppb) is weak and needs further discussion, probably in a supplement – or possibly removal. It is apparently by different research groups, with methods that are not only briefly presented to the reviewers. The time of day of sampling, sampling strategy, and specific compounds are not mentioned and could easily explain the difference. These need to be analyzed properly as an honest species-by-species comparative analysis to tell if there are statistically different concentrations of individual matched biogenic and anthropogenic VOCs – or it needs to be removed. Verification is needed that the samples are appropriately paired (location, time of day, air quality meteorology, season). Taking summer data for comparison to spring data is problematic. In its current form, the discussion of the VOC results seems like data cherry picked to support the predetermined conclusion of an increase in biogenic nucleation precursors, rather than an true hypothesis test.*

**Response:** The sampling methods and analysis of VOCs have added in the supplementary in Test S4, which reads as "VOCs in the spring campaign of 2007 at Mt. Tai were not measured. Mao et al.

(2009) measured the total VOCs at the same sampling site in June 2006, which were used for discussion. Mao et al. (2009) reported the measurements of 52 VOCs ($C_4$–$C_{12}$). The details on the method and the species of VOCs can be found therein. Theoretically, the temperature effect should increase BVOC emission more in June than those in spring. In the spring campaign of 2018, a total of 30 whole-air samples were collected for nine days. The collected VOCs were determined by gas chromatography (GC) separation, followed by flame ionization detection (FID), mass spectrometry detection (MSD), and electron capture detection (ECD) at a laboratory of the University of California at Irvine (UCI). 75 VOCs ($C_2$–$C_{10}$) non-methane hydrocarbons (NMHCs) were analyzed chemically. The analysis method of samples in 2018 has been reported by Chen et al. (2020). Note that a discrepancy on the measured VOCs between the two labs may exist and cause the uncertainty on the comparison to some extent."

*Comment followed by questions 4-7: The exclusion of 3 channels of the SMPS for 30% of hours during the final two campaigns is a big change. I note that it decreased the condensational sink in 2018 by a factor of two. Leaving aside how such a prominent feature could escape the initial QA/QC of the team, the fact that there is now verified instrument error in the 2017 and 2018 campaigns creates an additional requirement for clarity and transparency in the handling of the data. Unfortunately, without knowing more about the root cause for the spurious intermittent counts, it is difficult to trust the remaining 2017 and 2018 dataset. I feel this is important because the 100-300 nm section of the size distribution has a substantial portion of the condensational sink. I note there is what may be a specious peak at 100 nm in Figure S4, which corresponds to the prominent peak at 100 nm in figure S8 and S9 for most of the 2017 and 2018 data. I also note that the PM2.5 vs. WPS reconstructed PM2.5 relationship degrades over time from its best case performance in 2015. The argument that we have increased nucleation rates, lack of growth of nuclei to CCN sizes, an increase in condensational sink, a substantial increase in the size distribution function at 100-300 nm, and a large decrease in PM2.5 mass – this combination is difficult to explain, even by a large increase in HOM that that fuel nucleation but not substantial growth. In my opinion, it requires a source of 100-300 nm particles – either from changes in primary emissions in the region, changes in air pollution meteorology, or instrument error. The authors argue that the time periods are climatologically representative, so that leaves only changes in primary emissions, or instrument error.*

Response: The intermittent count error indeed missed from our screening because of three factors: 1) the instrument itself showed that it operated properly; 2) the error is small relative to the total number concentration; 3) it is our expectation for increased number concentrations in 80-300 nm occurring concurrently with decreasing mass concentrations of $PM_{2.5}$, based on our unpublished independent measurements made in a coastal megacity of China using a fast mobility particle sizer over the last decade. The third factor is out of the scope of this study, and don't detail here.

Figure R1 showed an example when ~100 nm particles prominent in an NPF day at Mt. Tai. In Figure S9, the prominent peak at ~100 nm appeared in 2009, 2014, 2015, 2017 and 2018. We also examined the relationship between $SO_2$ and the particle population between 100-300 nm ($N_{100\text{-}300nm}$). As shown in Figure R2, $SO_2$ and $N_{100\text{-}300nm}$ exhibited the moderate linear relationship in each campaign, although the slope varied on campaigns.

We thank the reviewer very much help us to find the instrument error around 213 nm, which

also reminds us to be more careful. It also indicates that the peer-review process is so important to correct the type of errors. Actually, most of the data in 2017 and 2018 didn't suffer from the problem. We will contact with instrument vendor to check what exactly took place.

[Figure]

Figure R1 Contour plot of the particle size distributions on NPF days on 18 Dec. 2017.

[Figure]

Figure R2 The relationship between $SO_2$ and particle number concentration in 100-300 nm ($N_{100-300nm}$).

*Specific questions related to this:*
*4. Was exclusion of three channels around 213 nm enough? Were all the needed times excluded? What was the test used to determine if a period was to have those channels excluded?*

**Response:** We checked the WPS data in 2017 and 2018. The errors around 213 nm occurred only in approximately 30% of the sampling days and correct accordingly. The errors are almost constant. After knowing the error, it is easy to be excluded.

An example with instrument error on an NPF day (27 Mar. 2018, contour plot in Fig. S4b) was shown in Figure R3. Particle number concentration in three bins of 192 nm, 213 nm and 237 nm

were abnormally high. Thus, the three bins were excluded in the statistical analysis (e.g., the calculation in Fig. S5, S8, S9). For the calculated CS, the error data were smoothed interpolated from 173 nm to 294 nm as shown by the purple dots and lines. This has been added in the revised Figure S4.

[Figure]

Figure R3 Particle number size distribution on 27 Mar. 2018 (shaded areas are the standard deviations, three bins inside blue circle were exclude from analysis, purple dots are smoothed interpolated data from 173 nm to 294 nm).

*5. How is the QA/QC and the replacement of data with smoothed interpolated data marked in the files that are now publicly available? This needs to be clearly explained in the supporting data files.*

**Response:** Sure. It has been added in the revised Figure S4 because the same problem may also occur in previous studies in the literature or future studies in research community.

*6. Show, in addition to the number count vs. number count comparison of instruments (Figure S3), the size distributions functions for hours where both instruments ran collocated. Comparing the number counts rather than the size distributions can hide substantial problems. Do the size distributions match, or does the WPS have some extra peaks? This would be a strong indication of the quality of the size distributions shown in Figures S8 and S9, and the condensational sinks and other parameters calculated from them. The size distributions should be shown with no data removal, and then with the removal/smoothing of suspect malfunction channels/peaks.*

**Response:** The comparison of particle number size distribution between WPS and SMPS was shown in Figure R4 (also added in revised supplementary materials). Both instruments showed unimodal distribution when no NPF events occurred during the instrument comparison. There was small difference in the peak of the particle mode, i.e., 56 nm for WPS and 64 nm for SMPS, which may be due to either the system difference between them or ~1.5 km distance between them. No data were removal in Figure R4, which has added in the caption. However, that the error of 213 nm particles was eliminated.

[Figure]

Figure R4 Particle number size distribution between WPS and SMPS during the summer campaign in 2020 (shaded areas are the standard deviations).

*7. The secondary peak at 100 nm in Figure S12 is quite striking. With a height of dN/dlogDp at 15,000 cm-3, and nearby minima at 5,000 cm-3 at 30 nm and at 250 nm, it really is a very sharp feature. Given problems at 213 nm, what checks were made to make sure this peak is real? The feature in Figure S12 is correlated with the PFGE, but abruptly disappears when instrument trouble strikes at 18:30, as indicated by the blue lines at 213 nm. There is also a faint indication of a "band" at 100 nm in Figure S4 for some 2018 data (and the data in question has acknowledged problems at 213 nm) – and it is appreciable in size, reaching dN/dlogDp at 10,000 cm-3. Such sister peaks appear in some PFGE datasets due to collocated sources of nucleation precursors with Aitken/accumulation mode particles; but such sister peaks seem (at least on average) absent from the Mt. Tai data prior to about 2017. For example, such a sister peak at 100 nm is totally absent in the 2014 figure shown (S12e).*

*My questions: after Figure S12 (or Figure S4) has gone through the exclusion process for the 213 nm 3-channel problem, is all the remaining data considered valid? What checks were made to make sure this 100 nm peak, growing in prominence in 2017 and 2018 (the same time as the 213 nm problem appears), is real? Do the other continuous instruments (for figure S12) support the existence of the 100 nm feature prior to 18:30. If from a primary source for the 100 nm mode, there is often correlation with PM2.5, CO2, EC, NOx, CO, and/or SO2. Is there an abrupt air mass change for S12 at 18:30 needed to support disappearance of virtually all particles? Presumably this is change from upslope (polluted) air to free tropospheric air. But now with the instrumental problems appearing in 2017 and 2018 – every opportunity to show instruments working properly and reporting valid data should be taken advantage of.*

**Response:** As shown in Figure R5, particles with the peak around 100 nm was associated with the low wind speed (<2m/s) and increased concentration of gaseous air pollutants, i.e., $SO_2$ and $NO_x$ (11:30-18:30). It seems that the increased number concentration of ~100 nm mode particles was due to the accumulation of air pollutants. The particle mode jumped to around 200 nm after 19:00, when the wind speed increased to ~6m/s. Meanwhile, the secondary chemical components, i.e., $SO_4^{2-}$, $NO_3^-$ and $NH_4^+$ in $PM_{2.5}$ largely increased by a factor of 5~6. This indicated that heterogeneous reactions may occur and lead the particles grow to the larger sizes. However, the 213 nm bins suffered the error, and the increased particle mass concentration cannot be accurately quantified.

Combined with Figure R1 and R2, we confirm that the particle mode around 100 nm was the true

signal, and they moderately correlated with $SO_2$ concentration. The increase of ~100 nm mode particles was due to either the long-range transport of air pollutants, or accumulation of air pollutants under low wind speeds.

[Figure]

Figure R5 Contour plot of NPF event, planetary boundary layer height (PBLH), time series of gaseous $SO_2$, NOx and particular $SO_4^{2-}$, $NO_3^-$ and $NH_4^+$, and wind direction and wind speed on 21 Mar. 2018.

*Minor issues*
*8. Figure S12 seems to be prior to the data manipulation around 213 nm.*

**Response:** Yes, we didn't modify the contour plots on this issue.

*9. Figures S8 and S9 should have y axis as dN/dlogDp so that they can be more easily compared to the other figures in the paper that use dN/dlogDp and to other publications. Using dN as the y axis makes the height dependent on the bin spacing.*

**Response:** Corrected.

**References**

Chen, C., Park, T., Wang, X. H., Piao, S. L., Xu, B. D., Chaturvedi, R. K., Fuchs, R., Brovkin, V., Ciais, P., Fensholt, R., Tommervik, H., Bala, G., Zhu, Z. C., Nemani, R. R., Myneni, R. B. China and India lead in greening of the world through land-use management, Nat. Sustain., 2, 122-129, 2019.

Chen, W. H., Guenther, A. B., Wang, X. M., Chen, Y. H., Gu, D. S ., Chang, M ., Zhou, S. Z., Wu, L. L., and Zhang, Y. Q.: Regional to Global Biogenic Isoprene Emission Responses to Changes in Vegetation From 2000 to 2015, J. Geophys. Res. Atmos., 123, 7, 3757-3771.

Ma, M., Gao, Y., Wang, Y., Zhang, S., Leung, L. R., Liu, C., Wang, S., Zhao, B., Chang, X., Su, H., Zhang, T., Sheng, L., Yao, X., and Gao, H.: Substantial ozone enhancement over the North China Plain from increased biogenic emissions due to heat waves and land cover in summer 2017, Atmos. Chem. Phys., 19, 12195–12207, https://doi.org/10.5194/acp-19-12195-2019, 2019.

Mao, T., Wang, Y., Xu, H., Jiang, J., Wu, F., and Xu, X.: A study of the atmospheric VOCs of Mount Tai in June 2006, Atoms. Envrion., 43, 2503-2508, https://doi.org/10.1016/j.atmosenv.2009.02.013, 2009.

Wang, J., Feng, L., Palmer, P. I., Liu, Y., Fang, S., Bösch, H., O'Dell, C. W., Tang, X., Yang, D., Liu, L., and Xia, C. Z. Large Chinese land carbon sink estimated from atmospheric carbon dioxide data, Nature, 586, 720-723, https://doi.org/10.1038/s41586-020-2849-9, 2020.

Zhang, X. D., Huang, T., Zhang, L. M., Shen, Y. J., Zhao, Y., Gao, H., Mao, X. X.,  Jia, C. H. and Ma, J. M.: Three-North Shelter Forest Program contributions to long-term increasing trends of biogenic isoprene emission in northern China. Atmos. Chem. Phys., 16(11), 6949–6960, https://doi.org/10.5194/acp-16-6949-2016, 2016.